# High-resolution profile of neoantigen-specific TCR activation links moderate stimulation to increased resilience of engineered TCR-T cells

Franziska Füchsl [1,12], Johannes Untch [1,12], Vladyslav Kavaka[2,3], Gabriela Zuleger [1], Sarah Braun [4], Antonia Schwanzer[1], Sebastian Jarosch [4], Carolin Vogelsang[1], Niklas de Andrade Krätzig [5,6], Dario Gosmann[1], Rupert Öllinger [5], Piero Giansanti [6,7], Michael Hiltensperger [1,8], Roland Rad [5,6], Dirk H. Busch [4,9], Eduardo Beltrán [2,3,10], Eva Bräunlein [1,8,13] & Angela M. Krackhardt [1,6,8,11,13] ✉

Neoantigen-specific T cell receptors (neoTCRs) promise safe, personalized anti-tumor immunotherapy. However, detailed assessment of neoTCR-characteristics affecting therapeutic efficacy is mostly missing. Previously, we identified diverse neoTCRs restricted to different neoantigens in a melanoma patient. In this work, we now combine single-cell TCR-sequencing and RNA-sequencing after neoantigen-specific restimulation of peripheral blood-derived CD8+ T cells of this patient. We detect neoTCRs with specificity for the previously detected neoantigens and perform fine-characterization of neoTCR-transgenic (tg) T cells in vitro and in vivo. We describe a heterogeneous spectrum of TCR-intrinsic activation patterns in response to a shared neoepitope ranging from previously detected more highly frequent neoTCRs with moderate activation to rare ones with initially stronger activation. Experimental restimulation of adoptively transferred neoTCR-tg T cells in a xenogeneic rechallenge tumor model demonstrates superior anti-tumor responses of moderate neoTCR-tg T cells upon repeated tumor contact. These insights have significant implications for the selection of TCRs for therapeutic engineering of TCR-tg T cells.

Immunotherapeutic regimens have revolutionized anti-tumor therapy of multiple malignancies, especially advanced by the efficacy of immune-checkpoint inhibition (ICI)[1]. ICI treatment is especially based on unleashing T cells, specifically recognizing tumor cells. However, the exact cellular interplay is often multi-faceted and requires deeper understanding to improve therapeutic response[2]. Besides ICI, T cell-based adoptive cellular therapy (ACT) approaches using tumor-infiltrating lymphocytes (TILs) or T cells genetically engineered to express T cell receptors (TCRs) or chimeric antigen receptors (CARs) have shown promising results[3–5]. Since an overall challenge of adoptive cellular transfer lies in attacking mutant cells without targeting healthy tissues[6,7], neoantigens arising from somatic, tumor-restricted mutations promise a safe, precise, and highly personalized target structure. In fact, tumor mutations correlate with response to ICI treatment[8] and represent prognostic biomarkers for successful immunotherapy, emphasizing the importance of neoantigens and neoantigen-specific

T cells for anti-tumor response[9]. Moreover, targeting neoantigens with TILs or TCR-transgenic (tg) ACT has been shown to confer deep, durable responses in various cancer entities[10–12]. However, the discovery of neoantigen-specific TCRs (neoTCRs) relied on labor-intensive functional T cell assays or sorting of reactive T cells in the past[13,14]. Recently, single-cell sequencing-based identification approaches led to a less biased discovery to some extent[15–20]. Yet, low frequency in peripheral blood and dysfunction of TILs still pose major challenges for neoTCR identification[21].

Recently, a clinical trial showed feasibility of adoptive transfer of individual neoTCRs in a small number of patients with diverse malignancies, although clinical efficacy was limited[22]. Optimization of this approach, including neoTCR selection, will require in-depth characterization of neoantigen-specific T cell phenotype and neoTCR functionality as well as understanding of mechanisms affecting sustained TCR-reactivity versus T cell dysfunction[23,24]. Generally, the heterogeneous functional states of tumor-specific T cells are known to range from strong effector to dysfunctional phenotypes, yet their effects on short- and long-term tumor control remain largely unclear[25–27]. So far, a small number of approaches combined neoTCR identification in tumor-derived TILs across different entities with transcriptomic characterization of the whole neoTCR-population, although limited focus has been put on individual TCR-clonotype properties and functional patterns[16–20]. Meanwhile, few preclinical models – tumor- or infection-based – aimed at deciphering the impact of TCR-stimulation strength on T cells and within the TCR repertoire of oligo-/polyclonal T cell responses, so far, with limited translational significance for engineering T cells[28–30]. Despite attempts to transfer such results into patient datasets, translational assessment of persistence of different patient-neoTCRs in tumor settings with chronic antigen presence is missing[31].

In this case study, we build on previous work, where we identified mutated peptide ligands by mass spectrometry (MS) and in-silico prediction in melanoma patient Mel15. Subsequently, we investigated TILs and PBMCs from Mel15 and discovered six neoTCRs targeting the two neoantigens KIF2C$^{P13L}$ and SYTL4$^{S363F}$ [14,31]. We now combine single-cell transcriptome sequencing (scRNA-seq) and single-cell TCR sequencing (scTCR-seq) and thereby identify two further KIF2C$^{P13L}$-specific neoTCRs. These two neoTCRs differ substantially in their precursor frequency in the patient and transcriptomic activation profile from the previously known TCRs with identical KIF2C$^{P13L}$-specificity and thereby reveal a broad functional repertoire of neoTCRs recognizing a common neoantigen. We show that diverse activation patterns detected in scRNA- and scTCR-seq of primary T cells are reinforced by in vitro and in vivo functionality of neoTCR-tg T cells. Moreover, including an in vivo xenograft model for repeated tumor rechallenge, we also provide evidence for substantial differences in maintaining the functional capacity of engineered T cells expressing defined neoTCRs depending on their stimulation signatures. We here demonstrate that upon repeated antigen encounter in vivo, neoTCR-tg T cells harboring an initially moderate activation pattern outperform initially more strongly activated TCR-T cells. These data correlate with different TCR frequencies in the patient and suggest the inclusion of such TCRs in the selection and modification of tumor-reactive TCRs for their application in ACT.

## Results

### Sensitive identification of neoTCRs via scRNA-seq
We previously reported about the neoantigens SYTL4$^{S363F}$ and KIF2C$^{P13L}$ identified in a melanoma patient (Mel15) using our proteogenomic approach as published before[14,32]. We subsequently detected reactive T cell clones and neoTCRs derived from peripheral blood mononuclear cells (PBMCs) or TILs of the patient with specificity for SYTL4$^{S363F}$ or KIF2C$^{P13L}$. Functional characterization of these neoTCRs revealed that TCR clonotypes of comparably lower avidity (KIF2C$^{P13L}$-reactive) in

comparison to those with higher functional avidity (SYTL4$^{S363F}$-reactive) showed high frequencies within tumor, lymph node and blood of the patient and surprisingly demonstrated equal reactivity upon initial tumor encounter in vivo within a xenogeneic murine tumor model[14,31].

To further understand qualitative transcriptomic differences between these previously described neoTCRs and potentially identify additional clonotypes, we performed scTCR- and scRNA-seq on a peripheral blood sample of stage IV melanoma patient Mel15 at the time he was treated with Pembrolizumab in a setting of no further evidence of disease[14]. By enriching for CD137$^+$ activated T cells following specific stimulation with the neoantigens SYTL4$^{S363F}$ and KIF2C$^{P13L}$ and employing scTCR-seq (Fig. 1a), we aimed to increase the sensitivity for detection of less frequent neoTCRs. As a reference for expansion rates, we compared the specifically stimulated and enriched blood sample to a freshly thawed, unstimulated sample from the same time point. Indeed, upon antigen-specific stimulation of enriched and expanded T cells, we observed an increase in peptide-specific T cells in the enriched population with significantly upregulated CD137 expression (Fig. S1a) and increased Interferon-γ (IFN-γ) secretion (Fig. S1b).

The diversity of TCR clonotypes with one defined alpha and one defined beta chain in our samples decreased throughout stimulation, from 1832 different clonotypes in the unstimulated to 279 in the restimulated sample. When including clonotypes with only one defined alpha or beta chain, the numbers decreased from 2657 different clonotypes in the unstimulated to 362 clonotypes in the restimulated sample (Supplementary Data 1). All six previously known reactive receptors ranged amongst the most expanded TCRs, suggesting high efficacy of the CD137$^+$-selection step for these TCRs, KIF-P1, KIF-P2, SYT-T1, SYT-T2, SYT-P1 and SYT-P2 (Fig. S1c–e and Supplementary Table 1). KIF-P1 accounted for 69.0% of the restimulated clonotypes, with a high baseline frequency of 3.2% before enrichment and thereby greatly exceeded all other receptors in total frequency (Fig. S1c, d). Besides, we sought to identify additional clonotypes with defined specificity by comparing abundance in the unstimulated and restimulated conditions (Fig. 1b and Supplementary Data 1). Two previously unknown clonotypes were identified—KIF-sc1 and KIF-sc2—demonstrating specific reactivity towards the mutated epitope KIF2C$^{P13L}$ (Fig. 1c). Surprisingly, their binding motifs were significantly different from the previously identified KIF2C$^{P13L}$-reactive TCRs with no matching human protein hits containing the recognition motifs (Fig. S1f and Supplementary Table 2). In the patient, the two additionally identified KIF2C$^{P13L}$-reactive TCRs could furthermore be detected at different frequencies in several compartments: both were detectable below the previously described high frequencies of KIF-P1 and KIF-P2 in lung and intestinal metastases as well as the respective lymph nodes (Fig. 1d and Supplementary Table 3). While KIF-sc1 was less frequent in the intestinal metastasis as well as the draining lymph nodes than KIF-sc2, the opposite was true for the lung metastasis and its draining lymph node (Fig. 1d and Supplementary Table 3). In conclusion, by combining CD137$^+$-enrichment and frequency comparison of clonotypes identified from scTCR-seq we could enrich all previously discovered neoTCRs and identify two further neoTCRs from peripheral blood. In comparison to the previously identified, the additional receptors showed significantly different binding motifs as well as varying frequencies in metastases and lymph nodes.

### Negatively regulated and proliferative transcriptomic signatures in ex vivo restimulated, patient-derived T cells
In order to understand if qualitative differences in T cell activation and proliferative capacity determine clonotype frequency as previously hypothesized[31], we combined scTCR-seq with transcriptome analysis via scRNA-seq upon neoantigen-specific stimulation using the described ex vivo TCR-centered restimulation model detached from the tumor microenvironment (TME). To specifically focus on stimulation-

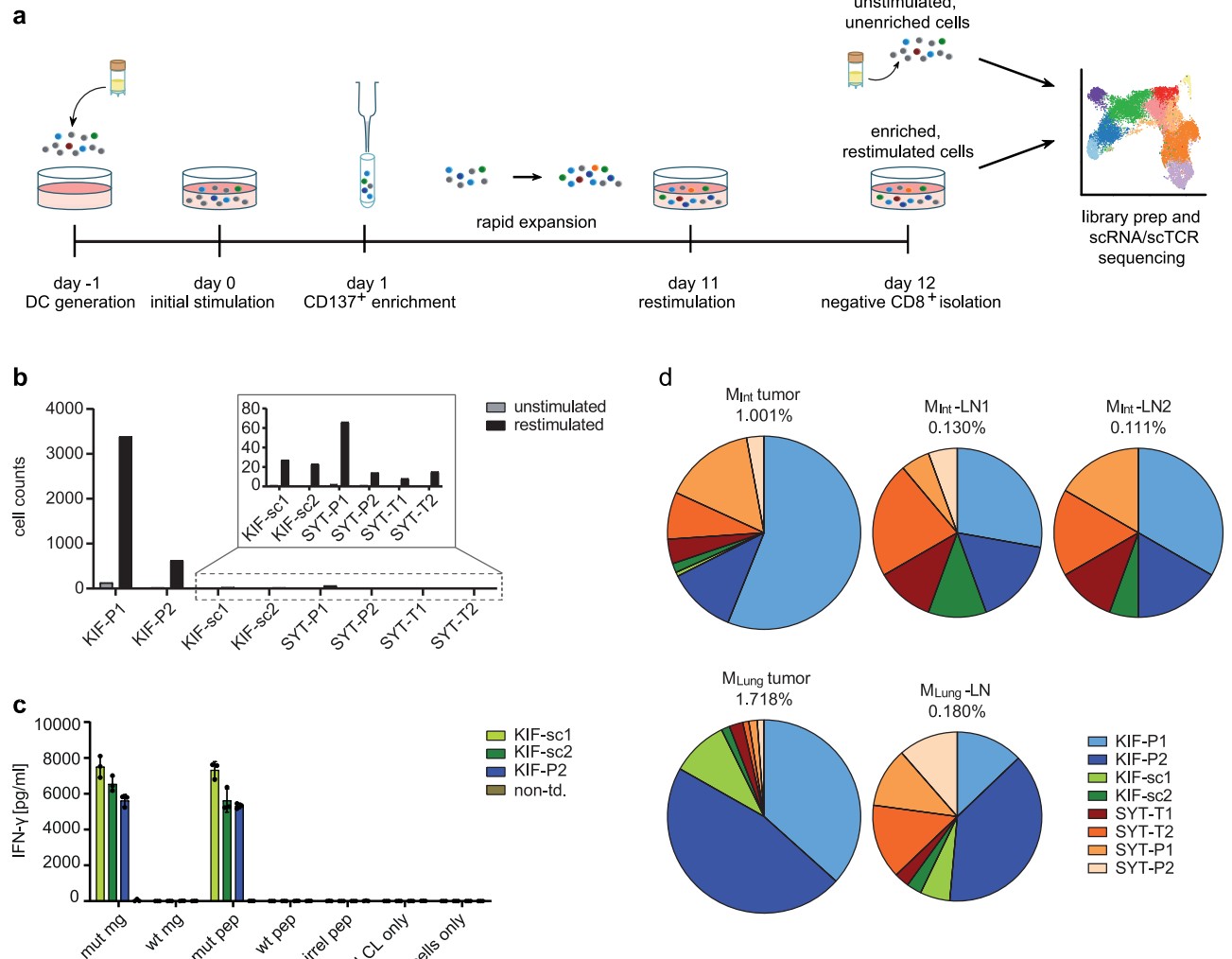

**Fig. 1 | Identification of neoTCRs via scTCR-seq of CD8⁺ T cells from melanoma patient Mel15. a** Schematic experiment setting. **b** Increase in total number of cells of known (KIF-P1 and -P2, SYT-T1, -T2, -P1, -P2) and additionally identified (KIF-sc1 and -sc2) TCRs upon antigen-specific stimulation and CD137-enrichment with dominance of KIF-P1 and -P2 harboring high precursor frequency. The two additionally identified KIF2C-TCRs were selected based on fold change of TCR frequency and highest absolute frequency in the stimulated sample. **c** Assessment of antigen-specific IFN-γ-secretion for the two identified TCRs KIF-sc1 and -sc2 in comparison to the known TCR KIF-P2. Cytokine secretion was measured by IFN-γ-ELISA upon 24 h of co-culture of TCR-tg T cells from one representative donor with

Mel15-LCL transgenic for the mutated KIF2C^P13L minigene (mut mg) and the wild-type KIF2C minigene (wt mg) as well as pulsed for 2 h at 37 °C with the mutated and wildtype peptide (mut pep and wt pep). An irrelevant peptide (irr peptide), target cells (LCL only) or T cells alone (T cell only) served as negative controls. Mean and SD of technical triplicates depicted. Data representative for two different donors. **d** Frequency of KIF-sc1 and -sc2 in relation to the previously identified TCR-sequences[31] identified by deep sequencing of the TCR-β-chain in intestinal (M_Int) and lung metastases (M_Lung) as well as corresponding non-malignant draining lymph nodes (M_Int-LN1, M_Int-LN2 and M_Lung-LN) of patient Mel15. Non-td non-transduced.

dependent effects, we performed unbiased clustering using differential gene expression of the CD137⁺ enriched, repeatedly neoantigen-stimulated cells compared to freshly thawed CD8⁺ T cells reflecting the mainly native TCR repertoire of patient Mel15 (Fig. 2a). These analyses revealed eleven clusters according to our experimental setting (Fig. 2a, b). As expected, unstimulated (mainly clusters 1–6, partly 7–9) and restimulated cells (mainly clusters 7–9 and partly also 5 and 6) clustered differently (Fig. 2a).

Within this approach, we focused on the overall distribution of T cell phenotypes across these defined clusters irrespective of their clonotype (Fig. S2a–f). A naïve-like, antigen-inexperienced transcriptional state (expressing *CCR7, LEF1, NELL*) could be identified in the unstimulated sample, mainly within clusters 1, 2 and 3 (Fig. S2a, b). A smaller fraction of the unstimulated as well as parts of the restimulated cells mainly clustering in 5 and 6 (partly also 8) could be assigned to an effector-like phenotype (expressing *CX3CR1, GNLY, GZMH, FGFBP2, FCGR3A, PLEK, ADGRG1, PRF1*), however, missing expression of

proliferative genes (Fig. S2b, c). Meanwhile, the upregulation of inhibitory surface receptors (most dominantly *LAG3*, but also *TIGIT* and *HAVCR2*) was a particular feature of clusters 6 and 7 mostly comprising stimulated cells (Fig. S2d, e). In contrast, stimulated T cells in clusters 8 and 9 had vastly initiated proliferative processes upregulating typical genes such as *MKI67, HIST1H4C, HSPD1, NME1, SP90AB1, ENO1, EIF4A1* (Fig. S2e, f). Within the mainly negatively regulated, inhibitory cluster 7 pathways indicating TCR signaling and cytokine-mediated response to the cognate antigen were highly upregulated, yet proliferative processes and cell cycle G2/M-phase transition were negatively regulated (Fig. S2g). Clusters 8 and 9 are prominently reflected in cell cycle phases (Fig. S2h) and high numbers of features and expanded cells of these clusters (Fig. S2i, j). Overall, as projected by trajectory analyses, the dynamic evolution of the differentiation state starting at cluster 1 (most naïve) with the lowest and ending at cluster 9 with the highest pseudotime score (Fig. S2k, l) also confirmed successful initiation of T cell stimulation within our experimental setup.

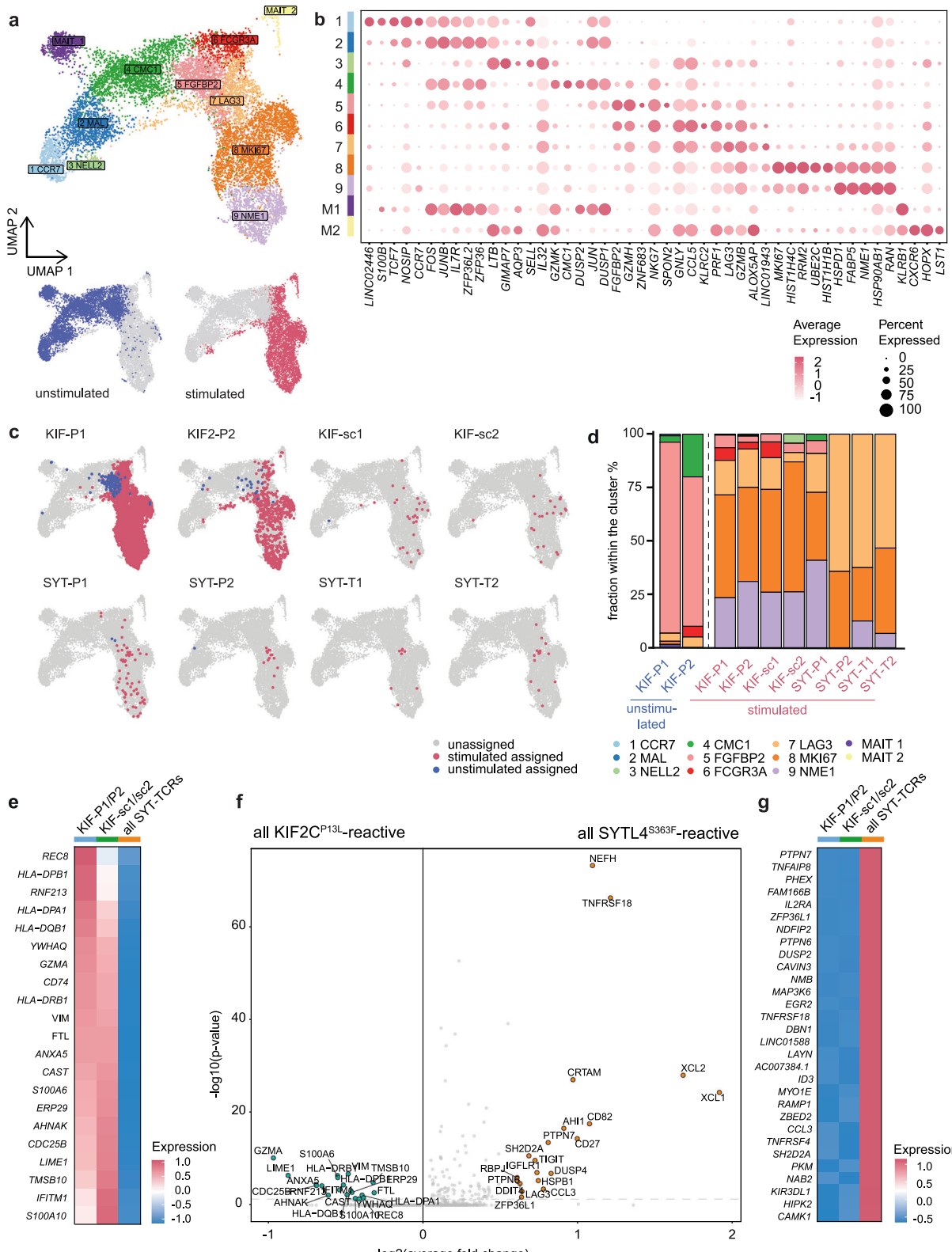

## Heterogeneous gene expression patterns in neoTCRs with shared and divergent specificities

Regarding stimulation patterns of each individual neoTCR, we then analyzed the distribution of all known clonotypes within these clusters (Fig. 2a, c). Regarding the cluster-related composition of different neoTCR-clonotypes, neoTCRs showed marked differences in their effector, inhibitory or proliferative state (Fig. 2d). From all unstimulated cells, only KIF-P1 and -P2 could be included in the comparison of

cluster distribution surpassing the subset-analysis threshold of 25 cells after quality control. Both clonotypes were mostly present in the FGFBP2-effector cluster 5 (Fig. 2a–d). Considering only the stimulated T cell population, all four KIF2C$^{P13L}$-specific TCRs were mostly present in a proliferative state (clusters 8 and 9), whereas SYT-P2, -T1 and -T2 were distinguished by a high percentage of cells from the inhibitory LAG3-cluster (cluster 7). SYT-P1 clustered more similarly to the KIF2C$^{P13L}$-specific pattern, raising the question about further

**Fig. 2 | Heterogeneous spectrum of transcriptomic activation patterns for different neoTCRs between cytotoxic, less inhibitory (KIF-P1 and -P2) and inflammation-related, negatively regulated activation (SYTL4$^{S363F}$-specific TCRs). a** UMAP of 5764 unstimulated and 6007 restimulated CD8$^+$, sorted T cells after QC with color code indicating 11 different clusters named after one of the most differentially expressed genes each (except for MAIT1 and MAIT2). UMAPs of unstimulated (blue, lower left graph) and restimulated (red, lower right graph), enriched (single alive) CD8$^+$ T cells next to the UMAP with all identified clusters. **b** Dot plot showing the five most differentially expressed genes per cluster. Size of each dot indicates percentage of gene-expressing cells per cluster; color indicates scaled average fold expression of the corresponding gene within the cluster. **c** UMAPs of unstimulated and restimulated CD8$^+$ cells showing distribution of single known TCR-specificities from both, the stimulated (red) and unstimulated (blue) sample. Non-assigned cells are depicted in gray for stimulated and unstimulated sample. **d** Bar-plot indicating percentual distribution of neoTCR clonotypes per cluster. Only conditions surpassing the threshold for minimal cell numbers (>25) were included. **e** Heatmap showing scaled average differential transcriptomic gene expression comparing all KIF2C$^{P13L}$-specific neoTCRs (separating KIF-P1 and -P2 from the identified KIF-sc1 and -sc2) and all SYTL4$^{S363F}$-specific neoTCRs (all SYTL4-TCRs). **f** Volcano plot indicating fold changes and *p* values of differential transcriptomic gene expression comparing all KIF2C$^{P13L}$- (all previously and all additionally identified TCRs) and SYTL4$^{S363F}$-specific TCRs. Wilcoxon rank sum test with Bonferroni correction for *p* value adjustment was used for statistical testing. **g** Heatmap showing scaled average differential transcriptomic gene expression comparing the same neoTCR-groups as in (**e**), ranked with focus on highest expressed genes in SYTL4$^{S363F}$-specific TCRs.

heterogeneity within the SYTL4$^{S363F}$-TCRs. It has to be noted, however, that the absolute number of cells compared per TCR differed substantially, likely associated with TCR frequencies before stimulation, among other factors (Fig. 1b and Supplementary Table 1). In conclusion, these transcriptome analyses supported the notion of heterogeneity within activation patterns of KIF2C$^{P13L}$- and SYTL4$^{S363F}$-specific TCRs.

An unbiased look at the differentially expressed genes of KIF2C$^{P13L}$- versus SYTL4$^{S363F}$-specific TCRs within the stimulated population further reflected the patterns described. On the one hand, KIF2C$^{P13L}$-specific TCRs upregulated genes of cytotoxic effector functions (*GZMA*), antigen presentation (MHC class II-genes and *CD74*) and TCR-signaling (*ANXA5, AHNAK, S100A6, S100A10, LIME1*) among which many are involved in calcium-dependent processes (Fig. 2e, f). On the other hand, SYTL4$^{S363F}$-specific TCRs diverged from this activation pattern upon stimulation. SYTL4$^{S363F}$-TCRs highly expressed genes correlated with chemokine profiles and proinflammatory pathways (e.g., *XCL1, XCL2, CD27, CCR3, CCL3*; Figs. 2f, g and S3a, b). At the same time, inhibitory receptors like *LAG3* and *TIGIT* (potentially also *TNFRSF18*), but also *DUSP4* and *PTPN7*, two MAP-Kinase inhibitors, were upregulated in SYTL4$^{S363F}$-TCRs implicating simultaneous inhibitory regulation. In contrast, the significantly upregulated genes for KIF2C$^{P13L}$-TCRs did not include such indicators of inhibitory signaling (Fig. 2e, f).

Differential gene expression revealed further insights on KIF2C$^{P13L}$-specific TCRs, indicating a distinct state for KIF-sc1 and -sc2 differing from both previously described qualitatively contrasting activation signatures of KIF2C$^{P13L}$- versus SYTL4$^{S363F}$-TCRs. This is displayed by a gradient detectable in the expression level of MHC class II genes, *CD74* and *GZMA* from KIF-P1 and -P2 over -sc1 and -sc2 towards SYTL4$^{S363F}$-TCRs (Fig. 2e). A heterogeneity between KIF2C$^{P13L}$-specific profiles was further supported by the direct comparison of KIF-sc1/-sc2 versus SYTL4$^{S363F}$-TCRs showing only the upregulation of genes associated with TCR signaling like *LIME1* and *S100A10* in KIF-sc1/-sc2 in contrast to upregulation of negative regulators like *PTPN7* and *DUSP4* only in SYTL4$^{S363F}$-TCRs (Fig. S3c). MHC class II genes, however, were not differentially upregulated between KIF-sc1/-sc2 and SYTL4$^{S363F}$-TCRs (Fig. S3c). Regarding unbiased analysis of unstimulated neoTCRs, again, only KIF-P1 and -P2 transcriptomes comprising sufficient cell counts could be analyzed. Comparing both TCR clonotypes with all other unstimulated T cell clones, cytotoxic markers, including *FGFBP2, GZMB, GZMH, GNLY* and *NKG7*, were predominantly upregulated (Fig. S3d).

Overall, we describe a spectrum of TCR-dependent T cell activation patterns in this scRNA-seq dataset from an ex vivo restimulation setting from patient-derived neoantigen-specific CD8$^+$ T cells. We detected cytotoxic, proliferative and less inhibitory T cell activation patterns, especially for KIF-P1 and -P2 and comparatively higher expression of inflammation- and chemokine-related as well as inhibitory genes for SYTL4$^{S363F}$-specific TCRs. KIF-sc1 and -sc2 shared features of both patterns.

## TCR-construct-inherent differences in surface expression of Mel15's neoTCRs in retrovirally and orthotopically TCR-engineered T cells

All cells originating from patient PBMCs possess a certain differentiation state at the time of blood collection due to multiple variables, including potential previous encounters with their cognate antigen as well as therapeutic regimens. To circumvent potential bias between T cell populations with different previous fates in the patient, we further compared the different neoTCRs after genetic transfer via retroviral transduction into activated CD8$^+$ T cells of several healthy donors in independent experiments. This in vitro analysis enabled antigen dose-titrated T cell stimulation and, moreover, helped to decouple TCR-intrinsic features from patient-specific cellular differentiation within the narrow spectrum of determined functional avidities[31]. Expressing the neoTCRs under the retroviral promotor (Fig. S4a), we observed notable differences in extra- and intracellular TCR expression (Fig. S4b–d), with KIF-sc1 showing the highest TCRmu$^+$ expression rates but also the highest TCR density as determined by gMFI (Fig. S4c, d). Of note, the relative differences in TCR surface expression between constructs were neither entirely reflected by the absolute quantity of TCR transcripts (Fig. S4e) nor insertions (Fig. S4f). High surface expression of KIF-sc1, however, was associated with the highest number of RNA transcripts detected (3-fold higher than the endogenous human TCR-β chain; Fig. S4e).

To rule out expression differences only based on the retroviral CMV-promotor, we further employed CRISPR/Cas9 for orthotopic TCR replacement (OTR) of the endogenous TCR-α chain by our TCR-constructs in the TRAC locus (Fig. S4g–k)[33,34]. Comparing the retroviral (RV) with the OTR system, we detected a similar level of TCRmu-surface expression per cell within the TCR-tg population in both systems for KIF-P2 and -sc1 after enrichment by FACS-sorting and in vitro expansion. This indicated similar construct-inherent surface levels under both promotors for these two TCRs, with KIF-P2 showing overall lower surface expression likewise in both engineering systems (Fig. S4k). In contrast, surface expression of KIF-P1 and KIF-sc2 markedly increased in the OTR system (Fig. S4k), suggesting altered expression characteristics of both TCRs under retroviral gene expression and potentially higher dependence on the expression system.

## Inflammation-related, inhibitory neoTCR-transcriptome signatures correspond to overall stronger activation of virally engineered TCR-tg T cells in vitro

In line with several current clinical ACT protocols, we further focused our analyses on retrovirally engineered T cells with the highest expression of neoTCRs: KIF2C$^{P13L}$-specific TCRs KIF-P2, -sc1 and -sc2 in direct comparison to SYT-T1. We investigated the effect of different stimulation strengths by cytokine secretion and expression of activation and inhibitory markers in response to target cells pulsed with ascending peptide concentrations (Figs. 3a–d and S5a–d). This illustrated stronger, more sensitive activation of SYT-T1-tg T cells after

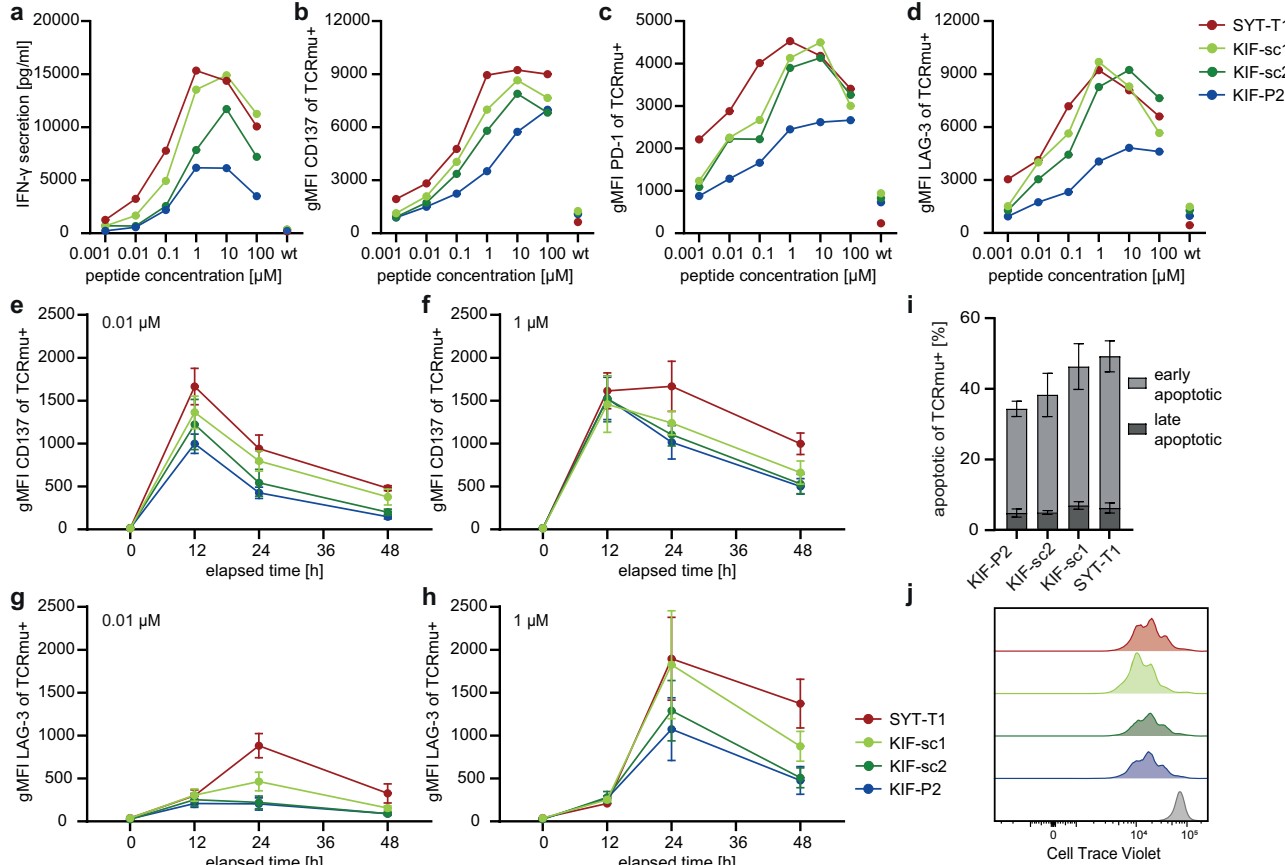

**Fig. 3 | Moderate versus strong activation patterns are transferable to CD8 neoTCR-tg T cells from healthy donors during in vitro co-cultures with different cell lines.** Color code for neoTCRs in **a–f** indicated next to **h**: SYT-T1 red, KIF-sc1 light green, KIF-sc2 dark green, KIF-P2 blue. **a–d** Mel15 LCL were pulsed with titrated peptide concentrations (2 h, 37 °C) and co-incubated with TCR-tg T cells with subsequent ELISA-based assessment of IFN-γ-secretion within 24 h of co-culture (**a**). The cellular activation level was determined after 24 h by FACS staining of the extracellular level of CD137 (**b**), PD-1 (**c**) and LAG-3 (**d**) expression (reflected by geometric mean of all CD3+CD8+/TCRmu+ cells). Wildtype control depicts only the highest peptide concentration (100 μM wt peptide). The mean for ELISA data is depicted for technical triplicates of one representative of four donors; triplicates from the same donor have been pooled prior to EC FACS-staining. E:T = 1:1 (15,000 tg T cells:15,000 tumor cells). EC FACS staining at different time points after co-culture setup displays temporal dynamics of T cell activation marker CD137 (**e**, **f**) and inhibitory receptor LAG-3 (**g**, **h**) for TCR-tg T cells upon co-culture with JJN3-

B27 peptide-pulsed target cells. A weak (0.01 μM for peptide pulsing; **e**, **g**) versus a strong (1 μM for peptide pulsing; **f**, **h**) stimulus were compared. E:T = 1:1 (10,000 tg T cells:10,000 tumor cells). gMFI-values of all TCRmu+ cells are shown. **i** Annexin-V/PI-staining was employed for detection of activation induced cell death (AICD) after 20 h of co-culture upon strong stimulation with 1 μM mut-peptide pulsed Mel15 LCL (early apoptotic = AnnexinV+PI−, late apoptotic = AnnexinV+PI+). E:T = 1:1 (30,000 tg T cells:30,000 tumor cells). **j** Representative FACS plot of a healthy donor of CTV-analysis for all TCRmu+ cells depicted after 4 days of co-culture with 1 μM mut-peptide pulsed Mel15 LCL (colors were chosen according to (**b–e**); representative wt mg-control depicted in gray). E:T = 1:1 (30,000 tg T cells:30,000 tumor cells). For all co-cultures, TCRmu+ rates were adjusted by addition of non-transduced T cells to equalize TCRmu+ cell frequencies for all neoTCRs. For all co-cultures in **e–j** technical triplicates per donor were pooled prior to staining; mean and SD for biological replicates from three different human donors are shown.

24 h, reflected by quantitative IFN-γ secretion compared to KIF-P2 as previously described[31]. KIF-sc1 and -sc2 showed intermediate responses between those two diverse reactivity patterns (Fig. 3a–d), as also confirmed by EC50-value measurement in the viral expression system (Fig. S5e and Supplementary Table 4). These differences in activation patterns were similarly reflected by the expression of the activation marker CD137 (Figs. 4b and S5b) as well as, during early activation, the inhibitory receptors PD-1 and LAG-3 (Figs. 3c, d and S5c, d); the latter being the most prevalent gene in the inhibitory signature of the transcriptome analysis (Fig. 2a, f). Besides functional avidity, structural avidity was only recently described as an important feature for TCR-functionality and T cell tumor tropism[35]. Comparing KIF-P2, -sc1 and -sc2, we did not detect any significant differences in k_off rates (Fig. S5f).

While the detected spectrum of activation strengths for different neoTCRs remained stable across different effector-to-target (E:T)-ratios (Fig. S6), we also analyzed temporal dynamics of activation kinetics at different stimulation strengths (Figs. 3e–h and S7). Surface

staining of CD137 over the course of 48 h on different TCR-tg populations upon co-culture with two different peptide concentrations (0.01 μM and 1 μM) consistently showed maximal expression after 12 h with a similar spectrum of activation patterns over time as observed before (Figs. 3e, f and S7a–c). Higher peptide concentration, nevertheless, prolonged the time of CD137 expression on a population level for all TCRs and, moreover, increased CD137 levels, particularly for the KIF2C^P13L-reactive TCRs. The same pattern was detected when stimulating TCR-tg T cells with another tumor cell line (Fig. S7d–i). We detected maximal upregulation of LAG-3 after 24 h, especially on SYT-T1- and KIF-sc1-tg T cells for both stimuli, while KIF-P2 and -sc2 upregulated LAG-3 only upon the strong stimulus (Figs. 3g, h and S7j–l). Similar trends could also be shown for PD-1 levels despite overall lower expression as compared to LAG-3 (Fig. S7m–o). In addition, the stronger TCR activation patterns of SYT-T1 and KIF-sc1 were linked to a slightly increased percentage of apoptotic (Annexin V+) cells in co-culture with diverse cell lines pulsed with 1 μM peptide

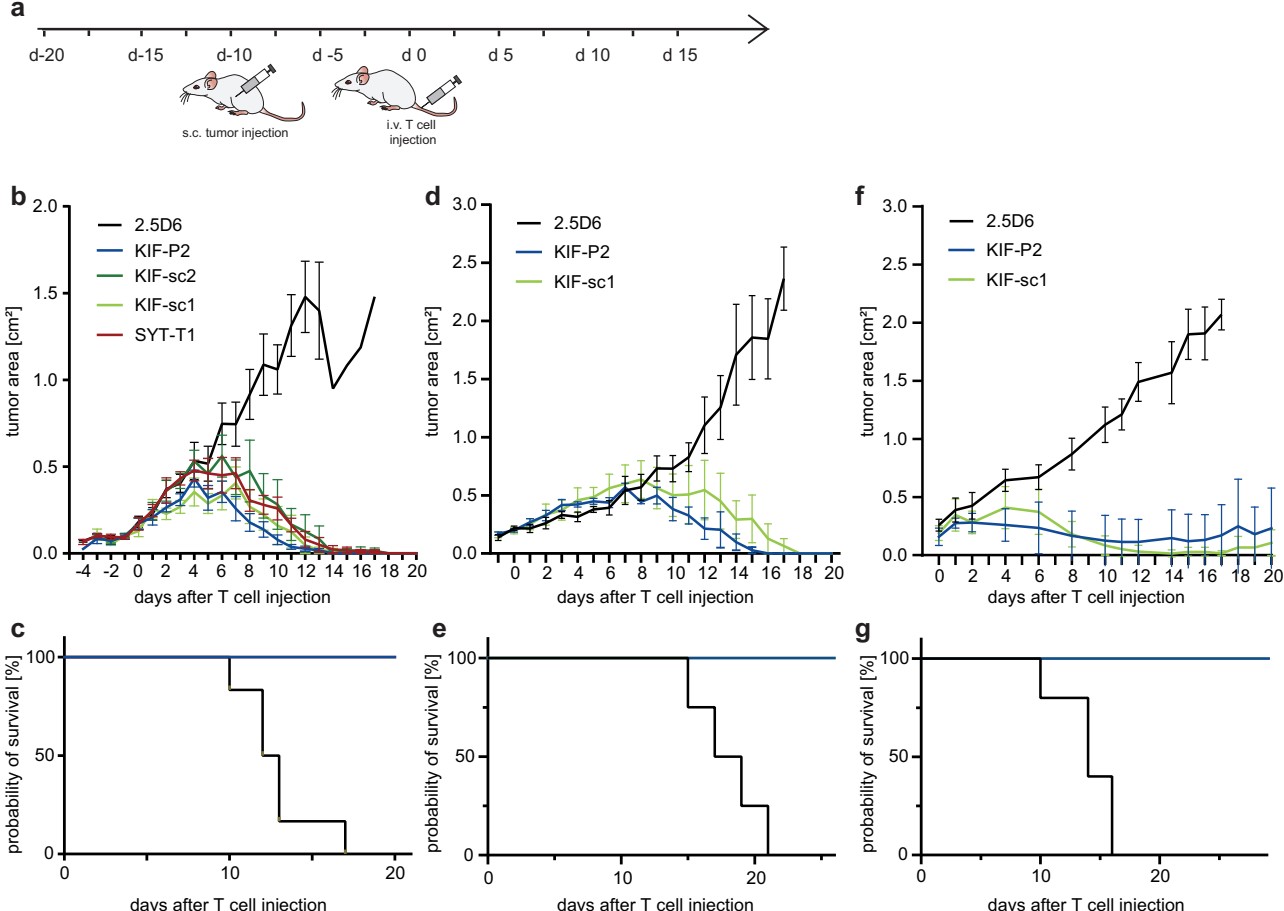

**Fig. 4 | Despite distinct activation patterns neoTCR-tg T cells demonstrate entity-independent, comparable in vivo tumor rejection upon first tumor encounter. a** Schematic setting of xenograft first-encounter tumor rejection experiment with newly transduced neoTCR-tg T cells. **b** Tumor growth kinetics in a lymphoma model are displayed as tumor area (in cm²) for U698M-mut-mg-tumor-bearing NSG-mice comparing neoTCR-tg T cells to irrelevant TCR 2.5D6 until day 20. 30 × 10⁶ TCR-tg T cells were injected per mouse. Mean and SD display rejection dynamics of tumor-bearing mice as biological replicates (n = 6). Parts of this dataset were published before[31]. Here, tumor rejection kinetics of KIF-sc1 and -sc2 analyzed in the same experiment are additionally shown. **c** Kaplan–Meier-survival curve is displayed for U698M-tumor-bearing mice injected with 30 × 10⁶ neoTCR-tg T cells

(Mantel–Cox test). **d** Tumor growth kinetics for U698M-mut-mg-tumor-bearing NSG-mice for KIF-P2, -sc1 and 2.5D6 after injection of 5 × 10⁶ TCR-tg T cells per mouse (n = 4 biological replicates). **e** Kaplan–Meier-survival curve displayed for U698M-tumor-bearing mice injected with 5 × 10⁶ neoTCR-tg T cells (Mantel–Cox test; KIF-P2 and KIF-sc1 p = 0.0067). **f** Tumor growth kinetics in a melanoma model for mut-mg-A2058-tumor-bearing NSG-mice after injection of 5 × 10⁶ TCR-tg T cells per mouse (n = 5 biological replicates). **g** Kaplan–Meier-survival curve displayed for A2058-tumor-bearing mice injected with 5 × 10⁶ neoTCR-tg T cells (Mantel–Cox test, KIF-P2 and KIF-sc1 p = 0.0026). For **d**–**g** mean and SEM display rejection dynamics.

(Figs. 3i and S8a–f). Meanwhile, no significant proliferative differences could be detected for these neoTCRs in vitro throughout the first four days after stimulation (Figs. 3j and S8g–i). Thus, despite higher levels of inhibitory receptors, proliferative dysregulation in vitro did not appear to be a key feature of TCR-tg T cells with strong activation patterns upon this first in vitro stimulation.

In summary, KIF-sc1-tg T cells show patterns of stronger activation more similar to SYTL4ˢ³⁶³ᶠ-specific TCRs, while KIF-sc2 showed comparably more moderate activation closer to the KIF-P2-pattern. This indicates a level of heterogeneity in activation patterns of TCRs with identical neoantigen/peptide-HLA-specificity. Overall, T cells transduced with TCRs associated with proinflammatory, negatively regulated transcriptomic signatures performed more sensitively in the applied viral expression system. They reached higher overall levels of cytokine secretion and activation markers but also increased inhibitory receptor expression upon first antigen encounter in vitro. This pattern of stronger activation could be described for SYT-T1 and partly KIF-sc1. In comparison, KIF-P2 and KIF-sc2 appeared with a more moderate activation signature.

## NeoTCR-tg T cells demonstrate comparable tumor rejection upon first in vivo encounter despite different activation patterns

To assess functionality in vivo, we investigated the anti-tumor reactivity of neoTCR-tg T cell populations, including the additional clonotypes in a previously established in vivo xenograft tumor model with the HLA-matched B cell lymphoma cell line U698M expressing minigenes encoding KIF2Cᴾ¹³ᴸ and SYTL4ˢ³⁶³ᶠ (mut mg). We initially used a model designed for highest efficacy in tumor rejection (Fig. 4a–c), revealing comparably potent rejection kinetics for neoTCRs—previously known and now additionally identified—compared to the irrelevant, MPO-specific TCR-control (2.5D6)[31]. We have already published part of the data from this experiment, including KIF-P2, SYT-T1 and 2.5D6[31], and now additionally show data for KIF-sc1 and -sc2, which were included in the same experiment. In this setting, the two additionally identified neoTCRs KIF-sc1 and -sc2 performed equally well compared to those previously known and reached complete tumor rejection in all mice with significantly prolonged survival (Fig. 4b, c).

Subsequently, to investigate differences within our observed spectrum of activation, we focused on two TCRs with shared

neoantigen-specificity, HLA-restriction as well as similar behavior of surface expression in different engineering systems, yet different activation patterns: moderate (KIF-P2) versus strong (KIF-sc1). Lowering effector cell numbers, both neoTCRs still demonstrated equally potent in vivo rejection in the lymphoma model (Fig. 4d, e). However, to return from our entity agnostic-approach to patient Mel15's entity, we also tested in vivo anti-tumor response against the endogenously HLA-A03-expressing melanoma cell line A2058 transgenic for the same neoantigen-encoding minigene (mut mg). Again, we observed potent anti-tumor response in all tumor-bearing hosts for both neoTCRs (Fig. 4f, g). Both cell lines selected for our in vivo model covered different levels of neoantigen surface expression: Notably, U698M expresses a 4.1- to 4.8-fold lower level of HLA-A03 compared to other tumor cell lines (Fig. S9a) and its surface level of KIF2C$^{P13L}$, measured by MS, ranked overall lowest compared to A2058 and Mel15 LCL (Fig. S9b). Thereby, MS analysis revealed comparable antigen levels resulting from minigene expression and in vitro peptide pulsing (0.1 μM and 1 μM) for all three different cell lines (Fig. S9b). In fact, the level of T cell activation after co-culture with minigene-expressing or peptide-pulsed targets correlated with the level of antigen since much higher concentrations of peptide were needed to achieve comparable activation between the mut mg and pulsed conditions for U698M compared to A2058. However, regarding Mel15 LCL in comparison to the other cell lines, it becomes evident that this response also seemed dependent on other determinants of the tumor entity (Fig. S9c–e).

Tumor-infiltrating T cells were characterized in both models at day 5 after T cell injection (Fig. S10) for their composition (Fig. S10b–e, l–o), activation status (Fig. S10f–i, p–s) and phenotype (Fig. S10j, k, t, u). Compatible with increased TCRmu⁺ T cell enrichment of KIF-P2 at the tumor site, we observed significantly higher percentages of TCRmu⁺ KIF-P2 than KIF-sc1 T cells in both models (Fig. S10c–e, m–o). However, despite clear signs of activation of T cells at the tumor site compared to those residing in the spleen (Fig. S10f–i, p–s), no further significant differences comparing the two neoTCRs could be observed (Fig. S10f–k, p–u).

Overall, the observed in vitro differences in neoTCR activation patterns did neither translate into significant differences in killing capacity nor TIL activation status upon first in vivo encounter of the tumor.

### Moderate TCR-signal associates with superior tumor control upon repeated neoantigen challenge in vivo

Aiming to understand the impact of the detected slight differences in activation between neoTCRs on long-term T cell functionality, we next challenged our setting by investigating repeated in vivo tumor challenge. Therefore, we generated TIL products (TIL-P) from tumor-bearing TIL-P-treated mice and after ex vivo expansion reinjected these cells (TIL-P-KIF-sc1-tg or TIL-P-KIF-P2-tg) into other tumor-bearing recipients. In parallel, we compared the performance of these TIL-P with a new transduction of the same two TCRs on freshly isolated CD8⁺ T cells from the same donor (NEW) as control groups (Fig. 5a).

While the newly transduced TCR-tg T cells conferred complete tumor rejection with both TCRs in all mice until day 18 as previously described, we observed clear dysfunction of TIL-P-KIF-sc1 upon rechallenge in vivo (Figs. 5b and S11). Tumors in TIL-P-KIF-sc1 mice could not be controlled by the T cell product applied as observed for animals receiving the non-specific T cell product (TCR 2.5D6). TIL-P-KIF-P2, meanwhile, reached potent tumor rejection in all mice and performed equally efficiently compared to the newly transduced (NEW) T cells (Figs. 5b, c and S11). These distinct response patterns upon repeated antigen challenge were observed albeit with incomplete tumor rejection in two additional independent experiments using T cells from different healthy human donors (Fig. S11).

During in vitro expansion of TIL-P of the individual mice, we again detected differences in TCRmu⁺ frequencies (Fig. S12a–c) and overall superior expansion of TCRmu⁺ KIF-P2 conditions despite overall comparable growth of the CD8⁺ fraction (Fig. S12d–i). This suggests superior preservation of proliferative capacity and TCR-expression for the moderate TCR KIF-P2. Prior to reinjection of TIL-P, we performed ex vivo co-cultures with the U698M-mut mg tumor cell line to compare T cell functionality. Multiplex analysis of several traditional CD8/natural killer (NK)-cytokines on a protein level did not reveal differences between both neoTCRs in the NEW conditions (Fig. 5d). Meanwhile, the comparison of both TIL-P revealed that secretion of classical CD8 effector cytokines (IFN-γ, IL-2 and TNF) was highly heterogeneous between donors and, therefore, not causative for the shared in vivo phenotype (Fig. 5e). While furthermore, no differences for cytokines linked to killing capacity (GzmA, GzmB and Perforin) were detected between both TCRs, it was interesting that secretion of the inhibitory, anti-inflammatory cytokine IL-10 was significantly upregulated in TIL-P KIF-P2 of all three donors (Figs. 5e and S13a–c) potentially linked to a protective role of this cytokine for these T cells. For one donor, we moreover investigated the transcriptome in a CD8⁺-purified fraction after in vitro stimulation via bulk-RNA-sequencing, which clearly separated stimulated from unstimulated cells as seen by principal component analysis (PCA) (Fig. S13d). Thus, we could also confirm an upregulation of *IL-10* transcripts in the KIF-P2 TIL-P CD8⁺ cells (Fig. 5f).

Rechallenging the selected neoTCRs in the melanoma model, we detected the same patterns of tumor rejection albeit with a smaller survival advantage for KIF-P2 (Figs. 5g, h and S14a–c). Moreover, among the secreted cytokines for TIL-P generated in response to the melanoma cell line, IL-10 secretion was among the cytokines increased for TIL-P KIF-P2 compared to KIF-sc1 (Fig. S14d, e). Altogether, these findings strengthen entity-independence of the resilience patterns described and underline their dependence on the TCR in our model. We furthermore lowered the effector cell number of TIL-P to 5×10⁵ TCRmu⁺ cells per mouse and included the two other KIF2C$^{P13L}$-specific TCRs into our rechallenge setting to cover the whole spectrum of identified KIF2C$^{P13L}$-neoTCRs. The previously detected activation spectrum in vitro ranging from moderately to strongly activated was translated into tumor growth dynamics in the rechallenge model during the first 14 days after T cell injection: KIF-P1 and -P2 significantly slowed down tumor growth, while KIF-sc2 and particularly KIF-sc1 did not (Fig. S15a, b). Thus, only KIF-P1 and -P2 were able to significantly improve survival despite the low effector cell dose upon tumor rechallenge (Fig. S15b).

The potent in vivo capacity of TIL-P KIF-P1 upon rechallenge was particularly surprising as significantly lower effector cell numbers were used for TIL-generation due to the inferior surface expression capacity of KIF-P1 in the RV system. In vitro, this TCR showed low activation levels and functional avidity in the RV system (Fig. S15c–f) and was not directly comparable to all other neoTCRs due to very low TCRmu⁺ frequencies in this system (Fig. S4b–d). Compared to the very similar structural avidities of the other three KIF2C$^{P13L}$-specific TCRs, KIF-P1 exhibited a 4.6-fold increase in structural avidity (Fig. S15g), suggesting a potential compensation mechanism for low TCR surface expression. KIF-P1 demonstrated a strong in vitro killing capacity comparable with all other TCRs (Fig. S15h), indicating distinct qualities of this neoTCR compared to the other neoTCRs.

### Orthotopically engineered T cells confirm increased resilience of KIF-P2 upon rechallenge

As described earlier, KIF-P1 profited substantially from the OTR-based expression (Fig. S4k). To investigate functional differences of Mel15's neoTCRs based on engineering systems in vivo, we continued to functionally test all KIF2C$^{P13L}$-reactive neoTCRs after orthotopic insertion into the TRAC locus via CRISPR/Cas9 (Fig. S4g–k). Due to

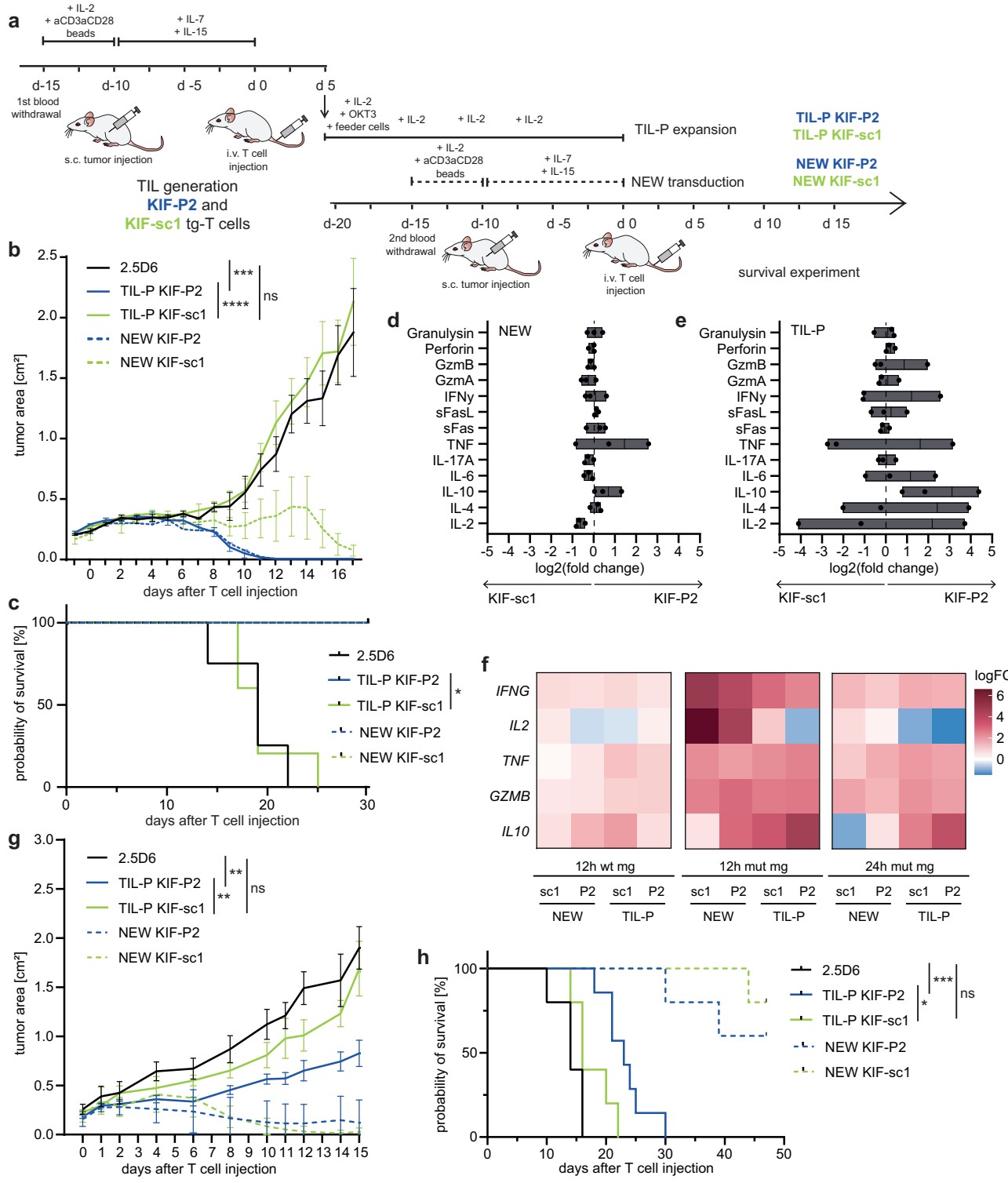

substantially lower TCR-knock-in frequencies compared to viral expression (Fig. S4g–k), an intensive expansion protocol (Fig. S4g) was applied to OTR- as well as RV-engineered cells for all following experiments (Fig. S16). Interestingly, higher surface expression of OTR-KIF-P1 led to increased activation levels but no functional advantage compared to RV-KIF-P1 (Fig. S16a–f), while increased structural avidity (9-fold increase in $k_{off}$ rate) remained similarly detectable (Fig. S16e). Meanwhile, OTR-KIF-P2 still demonstrated moderate T cell activation (Fig. S16a–d) linked to slightly increased killing and higher residual numbers of OTR-KIF-P2 neoTCR-T cells already upon first co-culture with neoantigen-expressing tumor cells (Fig. S16f–h). OTR and RV

engineered T cells expressing KIF-P2 or -sc1 were injected into tumor-bearing mice, which were sacrificed on day 5 for TIL-P generation (Fig. S16i–m). KIF-P2 showed slightly improved in vivo tumor control upon first tumor encounter until day 5 (Fig. S16i, j). Following TIL-P generation of the OTR-engineered T cells according to the previously established protocol (Fig. 5a), in vitro (Fig. S16n–u) and particularly in vivo rechallenge (Fig. S16v, w) substantiated significantly improved tumor control of the more moderate neoTCR KIF-P2 upon rechallenge compared to OTR-KIF-sc1 (in vitro: Fig. S16n, r; in vivo: Fig. S16v, w) and resulted in significantly prolonged survival of tumor-bearing mice (Fig. S16w).

**Fig. 5 | Moderate activation pattern of KIF-P2 T cells associates with sustained anti-tumor response upon in vivo rechallenge in contrast to strongly activated KIF-sc1. a** Setting of xenograft neoTCR-TIL-P rechallenge experiment. **b** Tumor growth kinetics of U698M-mut mg displayed as tumor area (in cm²) on day 17 after second injection of in total $5 \times 10^6$ neoTCR-tg T cells (55% TCRmu⁺ for all groups). For TIL-P-groups, TIL-P from two mice per TCR were pooled. Mean and SEMs for each group display rejection dynamics ($n = 5$ experimental groups, $n = 3$ 2.5D6; one 2.5D6-control sacrificed earlier). Statistical significance calculated for tumor area on day 17 with one-way ANOVA and Tukey's multiple comparison test (adjusted $p$ values of TIL-P-KIF-P2 versus 2.5D6: ***$p = 0.0001$ and TIL-P-KIF-P2 versus -KIF-sc1: ****$p < 0.0001$). **c** Kaplan–Meier-survival curve displayed for U698M-tumor-bearing mice injected with TCR-tg T cells ($n = 5$ for experimental groups, $n = 4$ for 2.5D6-control; Mantel–Cox test, $p = 0.0019$). **d**, **e** Log2(fold change) is depicted for the ratio KIF-P2:KIF-sc1 for cytokines secreted within 20 h of in vitro co-culture on the day of (re)injection of TIL-P or NEW T cell conditions (d0 of survival experiment) with U698M-mut-mg (E:T = 50,000:50,000). Ratios depicted for NEW (**d**) and TIL-P

(**e**) cells from three human donors (A, B and C). Mean and min-to-max-range depicted. **f** Heatmap showing transcriptional expression for selected cytokines detected in bulk-RNAseq on CD8⁺-enriched neoTCR-tg T cells of donor B normalized to 2.5D6-control (12 h or 24 h stimulation with U698M mut mg or wt mg). Technical triplicates pooled prior to CD8⁺-enrichment. **g** Tumor growth kinetics of A2058-mut mg displayed as tumor area (in cm²) until day 15 after injection of $5 \times 10^6$ neoTCR-tg T cells (55% TCRmu⁺ for all groups). Mean values and SEMs display rejection dynamics ($n = 5$ for experimental groups, $n = 7$ for TIL-P KIF-P2 and $n = 4$ for 2.5D6; one 2.5D6-control sacrificed earlier). Statistical significance was calculated for tumor area on day 14 with one-way ANOVA and Tukey's multiple comparison test (adjusted $p$ values of TIL-P-KIF-P2 versus 2.5D6: **$p = 0.0018$ and TIL-P-KIF-P2 versus -KIF-sc1: **$p = 0.0075$). **h** Kaplan–Meier-survival curve for A2058-tumor-bearing mice injected with TCR-tg T cells ($n = 5$, $n = 7$ for TIL-P KIF-P2; Mantel–Cox test, $p$ values of TIL-P-KIF-P2 versus 2.5D6: ***$p = 0.0005$ and TIL-P-KIF-P2 versus -KIF-sc1: *$p = 0.0138$).

---

Overall, based on our results we conclude more preserved functional activity and resilience upon tumor rechallenge in TIL-P-KIF-P2-tg T cells with primarily more moderate activation. In addition, we observed an enhanced expression of the anti-inflammatory cytokine IL-10 potentially associated to this reactivity pattern. In contrast, we show functional impairment for initially more strongly activated TIL-P-KIF-sc1-tg T cells upon antigen-specific T cell rechallenge and subsequently ineffective anti-tumor activity upon restimulation in vivo. Data from an orthotopic non-viral expression system strengthened these findings by highlighting the improved persistence of KIF-P2-engineered T cells. This oligoclonal neoantigen-defined TCR-repertoire highlights the complexity of TCR-intrinsic structural features influencing long-term anti-tumor functionality of TCR-engineered T cells.

## Discussion

To date, first clinical studies for adoptive transfer of highly personalized neoTCR-T cells prove clinical feasibility although therapeutic efficiency is still limited[22]. We are convinced that better understanding of neoTCR-inherent qualities is required for an optimal benefit from this promising approach. Neoantigen-reactive T cell clones typically represent minor fractions among TILs and comprise scarce populations in human blood[16,21,31]. Therefore, identification and characterization of neoTCRs still pose a major bottleneck for selecting T cells and TCRs with favorable characteristics for effective ACT. Several approaches already aimed at enrichment of tumor-reactive T cells, exemplarily by sorting for CD137⁺ or PD-1⁺ T cells[13,36]. In contrast to other recent studies on TIL-derived neoTCRs[16,18,37], we used peripheral blood-derived T cells of a metastatic melanoma patient under ICI treatment with known neoantigen-specific T cell reactivity. We present a restimulation-dependent single-cell sequencing approach detached from the TME for identification of neoTCRs and subsequent in-depth fine-characterization of these TCRs in vitro and in vivo.

The sequential approach of specific stimulation of blood-derived CD8⁺ T cells with MS-approved epitopes[14,32], magnetic enrichment of CD137⁺ cells and in vitro restimulation enabled sensitive detection of T cell clones specific for the two known neoantigens SYTL4^S363F and KIF2C^P13L despite partially very low precursor frequencies. Direct comparison with the native TCR repertoire thereby enabled a ranked quantification of T cell expansion rates after neoantigen-specific stimulation. Beyond detection of all six previously described neoTCRs[14,31], two additional neoantigen-reactive TCRs with specificity for KIF2C^P13L were identified. This suggests peripheral blood as a valuable, easily accessible source for detection of potent neoTCRs independent from the TME providing potential advantages compared to neoantigen-specific TILs which are often either not present or in an exhausted and dysfunctional state[16,38,39]. In fact, markers for dysfunction, such as CXCL13, CD39 or CD69, have been proposed as bio- or

selection markers for neoantigen-specific TILs with potential for diagnostic or therapeutic exploitation[16–20]. Of note, we did not observe notable transcriptomic upregulation of such markers among patient-derived, non-restimulated T cells. In contrast, we aimed at the dissection of neoantigen-specific T cells upon early (re)activation with focus on a head-to-head comparison between clonotypes with known specificity.

Since T cell effector functions are defined by distinct activation properties associated to intrinsic TCR-associated determinants, we went beyond a static signature of patient-derived neoantigen-specific TILs[16–20] by restimulating peripheral blood-derived T cells of Mel15 with defined mutated peptide ligands. Of note, upon specific in vitro restimulation we observed a heterogeneous pattern in neoTCR-dependent transcriptomics of these patient-derived T cells revealing qualitative differences between the identified neoTCRs. In synopsis with analyses on TCR-tg cells mainly in a retroviral expression system, we identified on one end of the spectrum more strongly activated but simultaneously inhibitory activation patterns especially within SYTL4^S363F-specific T cells, which harbored slightly higher functional avidities[31]. These cells were characterized by strong transcriptomic upregulation of proinflammatory markers and chemokines, e.g., the inflammatory chemokines XCL1 and XCL2, both regularly expressed by natural killer cells and activated CD8⁺ T cells[40,41]. Simultaneously, these cells also significantly upregulated inhibitory receptors (LAG3, TIGIT, HAVCR2) throughout the first 24 h of stimulation. Furthermore, SYTL4^S363F-specific T cells showed upregulation of DUSP4 and PTPN7, negative regulators of the mitogen activated protein kinase (MAPK)[42–44]. Interestingly, these neoantigen-specific T cells were found at comparably low frequencies in the patient potentially associated to defects in MAPK phosphorylation and subsequent proliferation as previously described for TCRs with higher signaling strength[45].

On the other end, KIF-P1 and -P2, neoTCRs with notably higher frequencies in the patient, demonstrated a distinct, in conjunction with functional data later defined moderate activation pattern with lower negative regulation. The marked transcriptomic upregulation of GZMA suggested cytotoxic capacity[46], while the presentation of HLA-class II molecules and CD74 may be associated with T cell-mediated antigen-presentation and proliferation[47–49]. The expression of genes related to calcium-dependent TCR-signaling, such as ANXA5[50], AHNAK[51], S100A6, S100A10 (S100 calcium binding proteins)[52] and with a lesser extent of Ca²⁺-dependency LIME1[53], further supported qualitative differences in signaling cascades. Both of these identified TCRs, KIF-sc1 and -sc2, seem to be in between those opposite transcriptional patterns.

To distinguish TCR-intrinsic features from those potentially imprinted by previous antigen encounter or other patient-specific properties, we retrovirally transduced T cells from healthy donors with defined neoTCRs and investigated functional patterns of these

TCRs. Our analyses in neoTCR-tg T cells largely reflected the activation spectrum determined by transcriptomic signatures and added further in vitro distinction between KIF-sc2 (closer to KIF-P1 and -P2) and KIF-sc1 (closer to SYTL4^S363F-reactive TCRs). These in vitro findings strengthen TCR-inherence of the heterogenous activation patterns of the patient-derived neoTCRs rather than patient-imprinted differentiation.

Despite general accordance between activation signatures of patient and TCR-engineered T cells, bias introduced through the artificial expression system cannot be fully excluded. Slightly higher functional avidity and activation were associated with higher TCR surface expression under a CMV promoter. However, similarly to the RV system, expression of KIF-P2 remained on a lower level compared to KIF-sc2 and KIF-sc1 under the endogenous TCR-α-chain-promotor in the OTR setting[33,34]. Thus, this difference in KIF-P2 TCR surface expression can be considered construct-inherent. The expression differences between OTR and RV, particularly highlighting a special role for KIF-P1, suggest that each engineering system likely may contribute to the performance patterns of each TCR individually. Substantially more aggressive in vitro expansion in the OTR-system might account for significant differences between TCR-T cells of different engineering modalities. However, the repeatedly shown more moderate activation and eventually superior tumor control of KIF-P2 upon rechallenge in both systems, further substantiates our findings. At the same time, it remains important to keep in mind that features of TCR-engineered T cells cannot be inferred directly on natural, patient-inherent neoTCR-expressing T cell clones.

Based on the strong clonotype distinction in patient-derived neoantigen-specific T cells (cluster 7 in our scRNAseq), we focused more closely on inhibitory regulation. Strong expression of LAG-3 and PD-1 upon specific stimulation in vitro upon early T cell stimulation corroborated the distinction between more strongly activated but simultaneously inhibitory patterns (SYT-T1 and KIF-sc1) and otherwise moderate expression of activation markers (KIF-sc2 and -P2) with limited inhibitory marker expression. Our observations are in line with previous reports accounting for a threshold of stimulation for the initiation of inhibitory programs as a protective rheostat mechanism during early T cell activation[54,55]. Currently, upregulation of inhibitory receptors is mostly understood as dysfunction and exhaustion[16,18,24], most likely resulting from chronic antigen encounter or over-stimulation early during tumorigenesis[56]. In our approach, simultaneously high levels of canonical activation as well as inhibitory receptors upon early activation could be observed alongside stronger induction of AICD as part of the strong activation pattern. This suggested TCR-driven dysfunction associated to hyperresponsivity, which, however, could not be observed functionally upon first tumor challenge in vivo for retrovirally engineered T cells.

To understand potential TCR-dysfunction in the context of chronic stimulation, we investigated the persistence and resilience of T cells tg for KIF-sc1 and KIF-P2 representative for the two opposing response patterns recognizing the identical antigen in a rechallenge model. To mimic repeated antigen encounter we adapted our in vivo model and restimulated TIL-P from xenograft tumor explants in vitro as well as in vivo. In this setting, we detected significant functional impairment of KIF-sc1, the TCR with stronger initial activation. Meanwhile, KIF-P2, the more moderate neoTCR with higher frequencies in the patient and lower inhibitory regulation, revealed potent in vivo tumor rejection, especially upon repeated stimulation independently from the tumor entity. Of note, the engineered antigen expression in our model cell lines does not reflect the heterogeneous neoepitope presentation expected in a primary tumor. Thus, further investigation of T cell resilience in response to varying antigen densities will be crucial to elucidate effects in different tumors with distinct mutational burden and intra-tumoral heterogeneity. Functionally, no traditional effector cytokine such as IFN-y, IL-2, TNF or GzmB was significantly

differently regulated between TIL-P KIF-P2 and -sc1 across healthy donors. Instead, the superior TIL-P from moderate TCR KIF-P2 upregulated secretion of the anti-inflammatory cytokine IL-10[57]. IL-10 secretion from CD8^+ T cells is known to have a protective function in acute viral infection[58]. At the same time, IL-10-receptor (IL10R) signaling plays an important role in sustaining non-exhausted T cell phenotypes in anti-tumor immunity[59]. In fact, co-expression of IL-10 and CARs has recently been demonstrated to increase preservation of T cell functionality and improve tumor control[60]. While more elaborated models beyond the highly artificial TME in NSG mice will be necessary to elucidate the role of CD8^+-driven IL-10 secretion after moderate, but not strong initial T cell stimulation, this finding complements the picture of a more stable, persistent functionality of TCRs with an initially moderate activation profile.

Consequently, we hypothesize from our findings, that (1) TCR-intrinsic features qualitatively determining activation have an enduring impact on the functional state, and (2) more strongly activated T cell reactivity patterns are associated with functional impairment upon repeated stimulation.

Other reports recently associated moderate rather than overly strong T cell stimulation to beneficial proliferation and longevity of T cells[45,61,62], lower TCR avidity to a more effector-like, and less exhausted phenotype with increased persistence upon murine chronic viral infection[63] and intermediate levels of TCR signal strength to superior anti-tumor efficacy in a murine model system[28]. Our results complement these findings with reverse translation of human data from a tumor patient and question the current understanding to exploit mainly high avidity T cells and TCRs for ACT[29,64–67]. Since factors like antigen density and tumor burden crucially affect T cell activation, it will be important to further test such response patterns in other tumor models with diverse tumor microenvironments.

The herein proposed picture of diverse neoantigen reactivity bases on the immune repertoire of one single patient and thus, in-depth analysis of other cases will contribute essentially to the definition of factors rendering T cell responses and respective neoTCRs as significant for individualized therapeutic approaches. The TCRs identified in Mel15 covered an only small range of functional avidity rather at the lower end of functional avidity scales in other recent publications for human neoTCRs[29,35]. Yet, we want to stress the substantial differences in maintained anti-tumor reactivity upon rechallenge suggesting further influences on T cell persistence beyond the slight differences in functional avidity. Recently, structural avidity was highlighted to improve prediction on tumor tropism of tumor-specific T cells[35], which could account for the surprising phenotype of KIF-P1 across all experiments despite low surface expression. However, very similar values between KIF-P2, -sc1 and -sc2 cannot explain the differences seen in their rechallenge response. This suggests association of individual neoTCR activation patterns to other structural determinants, binding properties or inherent signaling differences of TCR-peptide-MHC-interaction. NeoTCR activation patterns here appear as a complex equation of different variables which might compensate for each other and in their diversity be fittest within different settings.

In a synopsis, experimental outcomes like these and ours, have implications for T cell engineering and vaccination strategies[68,69] currently mainly focusing on enhancing co-stimulatory receptor interactions[70,71], reducing inhibitory signals[66] or TCR affinity maturation[64,72]. We show that individual TCR-intrinsic characteristics play a major role in determining T cell activation and sustained functionality in addition to peptide-HLA-complex density, antigen expression, co-signaling interactions and immunosuppressive factors in the TME. The question arising for future investigations therefore is, whether TCRs with qualitatively distinct activation profiles are necessary to complement each other in ACT. Whereas strongly activating TCRs might play a role in initial tumor debulking (under adequate ICI modulation), we hypothesize a substantial role for TCRs exhibiting

more moderate stimulation patterns in sustained and resilient long-term tumor control.

## Methods

This research and all experiments align with the regulations and approval of the institutional review board (Ethics Commission, Faculty of Medicine, project nr. 5722/13, 193/17S and 521/18S) of Technical University Munich and are in accordance with principles put forth in the Declaration of Helsinki. Informed consent of all participants in this study was granted in written form. All animal studies were approved by the Regierung von Oberbayern (Government of Upper Bavaria; ROB-55.2-2532.Vet_02-19-125).

### Primary patient material and cell lines

The clinical course of melanoma patient Mel15 was previously described in detail[31]. The identification of neoantigens resulting from somatic mutations (SYTL4[S363F] and KIF2C[P13L]) by MS and in silico-prediction were previously reported[14,31,32]. The PBMC sample of Mel15 used for single-cell-sequencing was selected based on previously confirmed reactivities within tested primary material at the specified time point[31], i.e., 966 days after first Ipilimumab application and 41 days after start of therapy with Pembrolizumab in a stage IV without evidence of disease.

PBMCs were isolated using density-gradient centrifugation (Ficoll-Paque) from either EDTA-anticoagulated blood of patient Mel15[31], EDTA-anticoagulated blood or leukapheresis products from healthy donors. PBMCs were either immediately included in further downstream assays or stored in freezing medium (90% FCS and 10% DMSO) in liquid nitrogen. Feeder cells used in this study included pools of irradiated healthy-donor PBMCs.

Cell lines used in this study included as target cell lines: Mel15 lymphoblastoid cell line (LCL) generated from Mel15 B cells by infection with Epstein-Barr Virus (EBV)-containing supernatant, T2 somatic cell hybrid (American Type Culture Collection - ATCC cat. CRL-1992; purchased from ATCC in 2005; RRID:CVCL_2211), U698M B cell lymphoma cell line (DSMZ cat. ACC-4, RRID:CVCL_0017) endogenously HLA-A03:01[+] and HLA-B27:05[+] as well as stably transduced with the mutated (mut mg) or wildtype (wt mg) tandem-minigene[31] and a fluorescent marker (Discosoma red fluorescent protein (dsRed) or green fluorescent protein (GFP)), JJN3-B27 multiple myeloma cell line (DSMZ cat. ACC-541, RRID:CVCL_2078), endogenously HLA-A03[+] and stably retrovirally transduced with HLA-B27 as described earlier and A2058 melanoma cell line (ATCC cat. CRL-3601, RRID: CVCL_1059), endogenously HLA-A03[+] and stably retrovirally transduced with mutated (mut mg) or wildtype (wt mg) tandem-minigene and a fluorescent marker (Discosoma red fluorescent protein (dsRed))[31]. For retroviral transduction the embryonal kidney cell line 293Vec-RD114 (BioVec Pharma, Québec, Canada) stably expressing gag/pol and env was employed. For mouse experiments NS0-IL15 cells, kindly provided by S. R. Riddell in 2011, were used.

T cells were cultivated as previously reported[14]. Target cell lines were cultivated in RPMI 1640 supplemented with 10% FCS, glutamine, non-essential amino acids, sodium pyruvate, and Penicillin/Streptomycin (Mel15 LCL, U698M, T2), DMEM supplemented with 10% FCS and Penicillin/Streptomycin (A2058) or 40% DMEM + 40% IMDM supplemented with 20% FCS and Penicillin/Streptomycin (JJN3 B27). RD114 cells were cultivated in DMEM supplemented with 10% FCS and Penicillin/Streptomycin. Growth and morphology of cultivated cells were checked routinely. Absence of mycoplasma infection in cell lines and media was regularly confirmed by PCR or a cellular-based detection assay (PlasmoTest, Mycoplasma Detection Kit, cat. rep-pt1).

### CD137 enrichment, rapid expansion and restimulation

To enrich PBMCs from patient Mel15 for KIF2C[P13L]- and SYTL4[S363F]-specific TCRs, we adapted our previously described method for

identification of neoantigen-specific TCRs[14,31]. Both neoepitopes arose from a non-synonymous point mutation, resulting in naturally presented ligands on HLA-A03:01 for peptide KIF2C[P13L] (amino acid sequence RLFLGLAIK) and HLA-B27:05 for SYTL4[S363F] (GRIAFFLKY). Briefly summarized, PBMCs from Mel15 were cultivated in AIM-V supplemented with cytokines. After 24 h, both neoepitope peptide ligands, KIF2C[P13L] and SYTL4[S363F] (0.1 μM) were added to the culture. Another 24 h later, reactive T cells were separated using magnetic labeling and positive selection with the CD137 MicroBead Kit (Miltenyi, cat. 130-093-476). CD137[+] enriched cells were then co-incubated with irradiated feeder cells in T cell medium (TCM) with supplements and expanded for eleven days.

After expansion, T cells were stimulated again with mutated KIF2C[P13L] and SYTL4[S363F] peptides using autologous antigen-presenting cells. Therefore, Mel15 LCL were pulsed either with 0.1 μM KIF2C[P13L] or SYTL4[S363F] and irradiated with 30 Gy. Expanded T cells and irradiated LCL were co-cultured at a ratio of 10:1 (T cells:LCL) for 24 h before preparing cells for single cell sequencing.

IFN-γ release of T cells was assessed before and after enrichment using ELISpot assay as described before[14]. Briefly, ELISpot plates were coated with IFN-γ capture antibody 1-DK1 (Mabtech, cat. 3420-3-250) and incubated with cells. After removal of cells, anti-IFN-γ 7-B6-1 (biotinylated, Mabtech, cat. 3420-6-250) as well as streptavidin-horseradish complex was added for visualization.

### CD8 isolation, scRNA-seq and scTCR-seq

CD8[+] T cells were negatively isolated from the enriched, restimulated as well as an unstimulated Mel15-PBMC sample from the same time point using the Dynabeads™ Untouched™ Human CD8 T Cells Kit (Invitrogen, cat. 11348D). Single, alive (Propidium Iodide (PI)-negative) cells were sorted, $25 \times 10^3$ cells from each sample were loaded onto one lane of a Chromium Next GEM Chip G (10x Genomics, cat. 1000263) and used for library prep using the Chromium next GEM Single Cell VDJ V1.1, Rev D) workflow (10x Genomics) as per company protocols. A high-sensitivity dsDNA was used for quality control and analyzed on a Bioanalyzer 2100. Quantity of dsDNA was measured using a Qubit dsDNA HS kit (Life Technologies, cat. Q32854). Libraries were sequenced on an Illumina NovaSeq 6000 using read lengths of 26 + 8 + 0 + 91 for combined assessment of single cell RNA sequencing (scRNA-seq) and TCR sequencing (TCR-seq) information.

### Single-cell sequencing data bioinformatic analyses

Samples were converted from BCL to FASTQ using bcl2fastq (demultiplexed).

Raw paired-end sequencing files of the GEX and VDJ libraries were aligned to the human reference genome (refdata-gex-GRCh38-2020-A) and VDJ reference (refdata-cellranger-vdj-GRCh38-alts-ensembl-4.0.0) respectively, using 10x Genomics Cell Ranger (v4.0.0). Subsequently, we used the R package Seurat (v. 4.1.0)[73] to further analyze the transcriptome- and TCR-based data. Only the genes detected in at least three cells were included in the raw counts matrix of the object. We retrieved only cells containing at least 200 genes and fewer than 6000 genes. To avoid possible dead cells contamination, we excluded cells with a fraction of mitochondrial genes higher than 18%. In the next step, the corresponding TCR data was added to the meta.data slot of the Seurat object. Raw gene counts were log-normalized, and variable features were detected with the vst method. Subsequently, canonical correlation analysis (CCA) integration was used to leverage the batch effects between two experimental setups combined in one Seurat object. After that, we newly determined the variable features using the integrated assay and scaled the expression matrix with regression on the number of UMIs and fraction of mitochondrial genes per cell. Unbiased calculation of k-nearest neighbors was done, and using UMAP, neighborhood graph and embedding were generated. After the UMAP construction, we retrieved only cells containing the TCR

information and clonotypes expressing more than one alpha or beta chain were removed. Previously identified neoTCRs from our index patient[31] were detected using their CDR3 region, and corresponding clonotypes in our samples were assigned to the respective TCR group. The final cell numbers in our linked TCR-transcriptome data set were 5764 cells in the unstimulated and 6007 cells in the restimulated sample. Cell cycle stage was determined with the CellCycleScoring function of the Seurat R package. The FindAllMarkers function was used to calculate differentially expressed genes in each cluster and the corresponding upregulated genes were retrieved for the subsequent pathway enrichment analysis using the enrichR (v. 3.0) R package. Seurat clusters were annotated manually by analyzing the expression of upregulated genes on the UMAP. The gathered signature expression score was generated by using AddModuleScore function. Subsequently, the Seurat object was converted into.h5ad format, and the pseudotime score with corresponding diffusion maps was generated using the scanpy library implemented in Python[74]. For pseudotime score calculation, cluster 1_CCR7 (most naïve) was set as a starting point.

For the differential gene expression analysis between the TCR groups, we used the FindMarkers function of the Seurat package by plotting the results using the ggplot2 (v. 3.3.5) R package.

### V(D)J analysis and selection of TCRs for TCR transduction
For subsequent TCR selection a meta data.csv was exported after initial QC (see above). Only clonotypes expressing exactly one productive alpha and one beta chain were considered to allow for precise identification of TCRs. The total number of this refined TCR set was 4182 in the unstimulated and 4913 in the restimulated sample. To select new neoTCRs, clonotypes that had previously been identified were excluded and the frequencies of remaining clonotypes were compared. We considered two metrics: highest fold change of TCR frequency before and after stimulation as well as greatest absolute frequency of clonotypes in the restimulated sample. We selected four new TCRs for investigation of specificity and functionality, two of them demonstrated specificity for KIF2C[P13L], later termed TCR KIF-sc1 and -sc2.

### Engineering of KIF-sc1 and -sc2 TCRs
α- and β-chain-sequences of clonotypes identified as potential neoantigen-reactive TCRs were submitted to IMGT to obtain comprehensive information on respective V-(D-)J sequences (https://www.imgt.org/IMGT_vquest/vquest). Full-length TCR sequences were reconstructed using Ensembl database and subsequently in silico optimized throughout insertion of a cysteine bridge, murinization of the constant region and codon optimization[75–77]. β- and α-chain were linked by a P2A element and tandem gene products were synthesized (BioCat). Each TCR candidate was cloned into MP71 retroviral vector and subsequently used for transduction into healthy donor T cells.

### Retroviral transduction of healthy donor CD8+ T cells with neoTCRs
CD8+ T cells used for transduction were obtained by magnetic negative selection from healthy donor-derived PBMCs (EasySep™ Human CD8+ T Cell Isolation Kit, Stemcell, cat. 17953) and activated for 48 h with 30 U/ml human IL-2 and anti-CD3-anti-CD28-beads (Dynabeads™ human T-Activator CD3/CD28, Thermo Fisher, cat. 11131D). Retroviral packaging cells RD114 were seeded to reach a confluency of 60% on the day of transfection and subsequently transfected with plasmids containing the neoTCR-α- and -β-chain-sequences using TransIT®-293 (MirusBio, cat. MIR 2700). Transfected cells were incubated for 48 h and supernatants subsequently filtered and used for spin infection of activated CD8+ T cells. Transduced T cells were cultivated with IL-7 and IL-15 for 10 days as described before[14]. Transduction efficacies were determined via fluorescence activated cell sorting (FACS) staining with TCRmu+

antibody (anti-mouse TCR-β-chain, FITC, BD Biosciences, RRID:AB_394683) against the murine-β-chain of engineered TCR-constructs in comparison to non-transduced T cell populations.

### Orthotopic T cell receptor replacement via CRISPR/Cas9 Knock-in
CRISPR/Cas9-mediated TCR engineering was done as described before[33,34]. In brief, isolated PBMCs were activated at a density of $1 \times 10^6$ cells/ml for 48 h with CD3/CD28 Expamer (Juno Therapeutics), 300 IU/ml IL-2, 5 ng/ml IL-7 and 5 ng/ml IL-15 in RPMI. Expamer stimulation was stopped by 20 min incubation with 1 mM D-biotin. Single guide RNAs (sgRNAs) were generated by annealing CRISPR RNA (crRNA) (80 μM; Integrated DNA Technologies) with trans-activating crRNA (tracrRNA) (80 μM; Integrated DNA Technologies) for 5 min at 95 °C. Ribonucleoproteins (RNPs) were then assembled by incubating sgRNAs with high-fidelity Cas9 (24 μM; Integrated DNA Technologies) at final concentrations of 12 μM Cas9 and 20 μM gRNA for 15 min at room temperature. Fifteen million cells were electroporated with 15 μl RNPs per target, 15 μg HDR-DNA template and 20 μM electroporation enhancer (Integrated DNA Technologies) in P3 buffer (Lonza) using the 4D Nucleofector X unit, pulse code EH-100 (Lonza) and the corresponding electroporation cuvettes.

After five days of cultivation, OTR and RV modified cells were enriched for TCRmu+ cells on an Astrios cell sorter (Beckman Coulter). Cells were then expanded with irradiated feeder cells in RPMI supplemented with 5% human serum, 180 IU/mL IL-2 and 1 μg/mL phytohaemagglutinin (PHA). Latest five days before experiments no more PHA was added and IL-2 reduced to 50 IU/ml. The following crRNA sequences were used: TRAC 5′-AGAGTCTCTCAGCTGGTACA-3′; TRBC 5′-GGAGAATGACGAGTGGACCC-3′.

### $K_{off}$ rates of TCRs using pMHC-multimers
TCR:pMHC $k_{off}$-rates were determined as previously described[78]. Atto488-conjugated monomeric pMHCs for StrepTamer staining were generated by in vitro refolding of the peptide of interest with HLA-A*03:01 heavy chain and β2 microglobulin as previously described[79]. pMHC-StrepTamer were generated by incubating 1 μl *Strep*Tactin-APC backbone (IBA, cat. 6-5010-001) with 1 μg Atto488-conjugated pMHC in a final volume of 50 μl FACS buffer (PBS 1x, 0.5 % (w/v) BSA, pH 7.45) for 30 min on ice in the dark. Up to $5 \times 10^6$ cells were stained with 50 μl multimer for 45 min on ice, in the dark. 20 min before the end of the StrepTamer staining, additional surface antibody staining was added. Cells were stained with PI for live/dead discrimination just before the acquisition. The final volume was adjusted to 1 ml with FACS buffer to allow an acquisition for up to 20 min. Acquisition was performed at 4 °C on a Cytoflex S (Beckman Coulter). Upon 30 s initial acquisition, 1 ml cold 2 mM D-biotin was added to the cell suspension whilst monitoring the dissociation kinetics. Analysis of the $k_{off}$-rates was performed with FlowJo and GraphPad Prism. $t_{1/2}$ were calculated by fitting of a one-phase exponential decay curve. FACS antibodies used for analyses: aCD45-PO (Exbio, clone HI30, RRID:AB_10952114), aCD45-PB (DAKO / Agilent, T29/33, RRID:AB_579532), aCD45-ECD (Beckman Coulter, J33, RRID:AB_130855), aCD45-PerCP (Thermo-Fisher, MEM-28, RRID:AB_11152976), amTRBC-APCeF780 (biolegend, H57-597, RRID: AB_2629697), aCD8a-PE (eBioscience, OKT8, RRID:AB_10732344).

### In vitro assessment of reactivity and activation patterns in TCR-tg or OTR engineered T cells
The subsequently described functional and phenotypic aspects were assessed within co-culture settings using retrovirally tg or OTR engineered CD8+ T cells from different healthy donors and different target cells. Cell lines were either transgenic for the tandem minigene (mutated minigene (mut mg) versus wildtype minigene (wt mg)) or pulsed with different concentrations of peptides KIF2C[P13L] and

SYTL4$^{S363F}$, their wildtype form or peptide derivates containing single amino acid substitutions with alanine and threonine at all possible positions as described before[31]. FACS- as well as ELISA-based readout was performed at different timepoints after co-culture setup as indicated. In selected experiments, varying transduction efficiencies (between donors and transductions) were equalized by diluting to the lowest rate per assay with a minimum at 10% of TCRmu$^+$ cells with non-transduced CD8$^+$ T cells obtained from the same donor. TCR-tg or OTR engineered TCRmu$^+$ T cells were considered effector cells for all E:T-ratios unless indicated otherwise.

## Extra- and intracellular FACS staining

FACS staining was performed in FACS buffer (PBS with 1% FCS and 2 mM EDTA) in 96well-u-bottom plates. Cells from in vitro co-cultures or tumor lysates were washed in FACS buffer and for in vitro co-cultures experimental triplicates were pooled prior to staining. Unspecific binding sites were blocked with 30% human serum in FACS buffer for 20 min at 4 °C before extracellular (EC) staining with diverse antibodies diluted in FACS buffer at 4 °C for 30 min. Live/dead stains were either directly added to the EC-antibody mix (Hoechst, Thermo Fisher) or added directly prior to measurement (PI, 7-AAD).

In addition to EC-staining subsequent intracellular (IC)-staining was performed for several analyses. Prior to EC-staining, a fixable live-dead stain (Zombie UV or Zombie NIR, biolegend) was stained in PBS. After EC-staining, cells were washed and fixed (fixation buffer, biolegend) for 20 min at room temperature (RT) (protected from light). Afterwards, perm buffer (biolegend) diluted in deionized water was used for permeabilization according to manufacturer's protocol. IC-staining antibody-mix in perm buffer was added afterwards for 40 min at RT (protected from light), followed by further washing steps.

FACS antibodies used for analyses: aCD137-APC (RRID:AB_830671) and aCD137-APC-Cy7 (RRID: AB_2629645, all biolegend, clone 4B4-1), aCD137-PE (RRID:AB_314782), anti-murine TCR-β-FITC (BD Biosciences, H57-597, RRID:AB_394683), anti-murine TCR-β-PE (BD Biosciences, H57-597, RRID:AB_10563767), aCD3-AF700 (biolegend, UCHT1, RRID:AB_493740), aCD3-PerCP/Cy5.5 (biolegend, UCHT1, RRID: AB_893301), aCD2-BV785 (biolegend, RPA-2.10, RRID: AB_2800717), aCD45RA (biolegend, H100, RRID:AB_10708880), aCD45RO-APC (biolegend, UCHL1, RRID:AB_314426), aCD8-PE-Cy7 (BD, clone RPA-T8, RRID: AB_396852), aCD8-PerCP (biolegend, SK1, RRID:AB_2890877), aHLA-A03-APC (Miltenyi, REA950, RRID:AB_2727171), aIFN-γ-APC (biolegend, 4S.B3, RRID:AB_315237), aIL-2-BV785 (biolegend, MQ1-17H12, RRID:AB_2566471), aTNF-PE-Cy7 (biolegend, Mab11, RRID:AB_2204079), aLAG-3-BV605 (biolegend, 11C3C65, RRID: AB_2721541), aLAG-3-BV650 (biolegend, 11C3C65, RRID: AB_2632951), aPD-1-BV785 (biolegend, EH12.2H7, RRID:AB_11218984), aPD-1-APC-Cy7 (biolegend, EH12.2H7, RRID: AB 10900982). Sample analysis was performed at an LSRII (BD Biosciences, RRID:SCR_002159) or LSR Fortessa (BD Biosciences, RRID:SCR_019601). FACS data was analyzed using Flow Jo_v10.8.1.

## Activation induced cell death (AICD) assessment: Annexin-V staining.
Cells stained extracellularly with FACS antibodies were stained in Annexin-V binding buffer diluted in water (Thermo Fisher) with AnnexinV (APC, biolegend) and PI for 20 min at RT prior to analysis.

## Proliferation assessment: cell trace violet (CTV)-staining.
TCR-tg CD8$^+$ T cells from three healthy donors were labeled with CTV Dye (Thermo Fisher) according to manufacturer's guidelines. On day 4 of co-culture, cells were stained extracellularly with FACS antibodies and afterwards the percentage of TCRmu$^+$ T cells per division was determined via flow cytometric readout.

## Quantitative analysis of the murine TCR-β-chain on RNA and DNA level

**RNA and gDNA isolation.** For the isolation of RNA, snap-frozen cell pellets were first homogenized using the QIAshredder Homogenizer (Qiagen, cat. 79656). Then, RNA was isolated with the RNeasy mini kit (Qiagen, cat. 74104) including an on-column DNA digestion step using the RNase-free DNase set (Qiagen, cat. 79254) according to the manufacturer's instructions. RNA was eluted with 25 μl DEPC-H$_2$O and quantified with a NanoDrop 1000 spectrophotometer (Thermo Fisher). Genomic DNA (gDNA) was extracted from snap-frozen cell pellets with the DNA blood and tissue kit (Qiagen, cat. 69504) and eluted using 30 μL DEPC-H$_2$O. To prevent co-purification of RNA, the RNA was removed using 4 μL Monarch RNAse A (20 mg/mL, NEB).

**Reverse transcription of RNA.** 1 μg RNA was reversely transcribed to cDNA using the AffinityScript Multiple Temperature cDNA Synthesis Kit (Agilent Technologies, cat. 200436) following the manufacturer's protocol.

**Real time (RT) PCR.** The RT-PCR was performed in a QuantStudioTM 5 Real-Time-PCR-System (Applied Biosystems). The assay was carried out in a 20 μL reaction volume using 2 μL of 1:10 diluted cDNA or 50 ng gDNA, 0.6 μM of each, forward and reverse primer, and 10 μL PowerUp SYBR Green Master Mix (Applied Biosystems). The cycling conditions used were the following: 50 °C for 2 min, 95 °C for 10 min, and then 40 cycles of 95 °C for 15 s and 65 °C for 1 min. After the run, a melt curve analysis was performed to determine the specificity of the primers. For the absolute quantification of the TCRs on the RNA and the gDNA level, standard curves were generated using serial dilutions of the respective vector that was used for the retroviral transduction ranging from 10$^6$ to 10 copies. For normalization, additionally a control vector encoding the constant region of the human TCRβ chain was used. Primers used for RT-PCR:

KIF-P1-fwd AGCAAAGAGACTCCGCAATG, -rev CTTTGTACGCCTG TGGATCC;

KIF-P2-fwd CGGACAAGGGTGAGGTATCT, -rev GAATCCTCGGGC CAAACAAA;

KIF-sc1-fwd TCAATAACAACGTGCCTATCGA, -rev AGGTGTCACA TTCCTCAGGT;

KIF-sc2-fwd TACAGACAGTTCCCCAAGCA, -rev TTCTCAGATC CTCCACCACG;

2.5D6-fwd CTGATGGCTACAACGTGTCC, -rev CACCAAGACAGTT CCACGTG;

huTCRb-fwd GAAGCAGAGATCTCCCACAC, -rev CCCGTAGAAC TGGACTTGAC.

## In vitro real-time monitoring of TCR-mediated cytotoxicity

Killing of adherent target cells by T cells was measured with the xCEL-Ligence® RTCA eSight-System of using the technique of impedance-based real-time cell analysis as described before. Briefly, culture media measured in 96 well E-Plates (OLS) for background impedance assessment. A2058 were seeded as target cells (30,000/well) and were incubated for 24 h to reach a growth plateau. Impedance was measured every 15 min with the xCELLigence® system. Measurement was paused for addition of TCR-tg T cells in a 1:1 E:T-ratio and the analysis was continued every 15 min for further 30 h. The number of effector cells was equalized according to their retroviral transduction efficiency.

To allow direct comparison of killing mediated by different neoTCR-tg T cells, cytolysis was calculated with normalized Cell Indexes (CI) by using the following formula (1):

$$specific\ cytolysis\,[\%] = 100 - \left( \frac{CI_x}{CI_{non-transduced}} x100 \right) \qquad (1)$$

## Mass spectrometry (MS)-based measurement of neoepitope KIF2C$^{P13L}$ abundance

**Preparation of cells.** The cell lines U698M, A2058, Mel15 LCL, U698M-mut mg, A2058-mut mg and Mel15 LCL-mut mg were expanded to reach sufficient cell numbers. The cell lines U698M, A2058 and Mel15 LCL were pulsed with 0.1 or 1 µM of KIF2C$^{P13L}$ in AIM-V at a cell concentration of $5 \times 10^6$ cells/ml while rotating for 2 h at 37 °C. The cells were washed two times with cold PBS before $1.5 \times 10^8$ cells of each condition were snap-frozen.

## Immunoprecipitation of HLA peptide complexes and purification of HLA peptides

For the purification of pan-HLA class I peptides from cell lines, the cells were first lysed in 4 mL lysis buffer (PBS 1x, 1% (w/v) Ocytl-β-D-Glucopyranoside, 0.25% (w/v) Na-Deoxycholate, 1 mM EDTA, 1 mM phenylmethylsulfonyl fluoride (PMSF), pH 8.0) for 2 h at 4 °C. Meanwhile, 1 mg mouse IgG2a W6/32 per condition was coupled to 0.5 mL Pierce Protein G Agarose beads (ThermoFisher) through incubation for 2 h at 4 °C while rotating. The lysates were cleared by centrifugation at 20,000 g for 20 min at 4 °C. Subsequently, the antibody-coupled beads were transferred to the cleared lysates and immunoprecipitation was performed O.N. at 4 °C while rotating. The beads were washed sequentially with 5 mL of 20 mM Tris-HCl buffer, pH 8, that contained varying concentrations of NaCl (150 mM, 400 mM, 150 mM, 0 mM). HLA peptides were then eluted from the beads together with the IP antibody and the MHC complex in three subsequent elutions with 1 mL 200 mM Glycine buffer, pH 2.5. Between elutions, the beads were incubated in elution buffer for 5 min at RT while rotating. The proteins were separated from the HLA peptides by using 10 kDa molecular weight cut-off columns (Millipore). The volume of the <10 kDa fraction was then reduced to 200 µL using vacuum centrifugation in order to purify the HLA peptides using C18 SPE-StageTips (3 M). The elution of HLA peptides was performed using 50 µL 60% acetonitrile (ACN) in 0.1% formic acid (FA). The peptides were finally dried using vacuum centrifugation before they were used for mass spectrometry analysis.

## Mass spectrometry analysis

HLA peptide samples were resuspended in 0.1% formic acid (FA) and analyzed by LC-MS/MS (liquid chromatography tandem mass spectrometry). Peptides were chromatographically separated using a Dionex Ultimate 3000 RSLCnano system (Thermo Fisher Scientific) coupled to an Orbitrap Eclipse mass spectrometer (Thermo Fisher Scientific). Peptides were loaded to a trap column (75 µm i.d. × 2 cm, packed in-house with 5 µm of ReproSil-Pur 120 ODS-3 beads, Dr. Maisch) using 0.1% FA at a flow rate of 5 µL/min for 10 minutes. Subsequently, peptides were transferred to an analytical column (75 µm i.d. × 40 cm, packed in-house with 1.9 µm ReproSil-Pur C18-AQ beads, Dr. Maisch) at a flow rate of 300 nL/min and chromatographically separated using an 80 min linear gradient from 4% to 32% of solvent B (0.1% FA, 5% DMSO in ACN) and solvent A (0.1% FA, 5% DMSO in water). The total measurement time for each sample was 90 min. The Orbitrap Eclipse was operated in data dependent mode, automatically switching between MS1 and MS2 spectra. MS1 survey spectra were recorded in the Orbitrap from 360 to 1800 m/z at a resolution of 120 K (automatic gain control (AGC) target value of 100%, maximum injection time (maxIT) of 50 ms). Peptide fragmentation was performed via higher energy collisional dissociation (normalized collision energy of 30%), and MS2 spectra were recorded in the Orbitrap at 30 K resolution via sequential isolation of the 15 most abundant precursors (isolation window 1.3 m/z, AGC target value of 400%, maxIT of 54 ms, and dynamic exclusion of 35 s). To enhance coverage, mass ranges were specified for each charge state as follows: 360–1800 m/z for charge 2–4+, and 700–1800 m/z for charge 1+. The acquisition method also integrated an "inclusion list" containing the theoretical mass of the doubly charged HLA peptide RLFLGLAIK, 515.8422 m/z.

Raw mass spectrometry data were processed using the FragPipe software (version 21.1) with its built-in search engine MSFragger version 4.0[80]. Spectra were searched against the human UniProtKB database UP000005640 (82,507 entries downloaded on 04.2024), supplemented with the translated open reading frame of the KIF2C gene, containing the mutated sequence RLFLGLAIK. Default parameters for a nonspecific-HLA search were employed, with a defined precursor tolerance of 20 ppm and no enzyme specificity for database digest. After peptide-to-spectrum matches (PSM) rescoring via MSBoster and percolator[81], identifications were adjusted to 1% false discovery rate (FDR) at the peptide and PSM levels, whereas protein FDR was not applied. IonQuant[82] was used to perform MS1-based quantification of the detected peptide features, with the match-in-between runs option enabled.

The mass spectrometry proteomics data have been deposited in the ProteomeXchange Consortium via the PRIDE partner repository[83] with the dataset identifier PXD051734.

## In vivo tumor rejection potential in a xenograft model

NOD.CG-Prkdcscid IL2rgtm1Wjl/SzJ (NSG; The Jackson Laboratory, RRID: IMSR_JAX:021885) were maintained according to the institutional guidelines and approval of local authorities (Regierung von Oberbayern; ROB-55.2-2532.Vet_02-19-125). A xenograft murine model was established as previously described[84,85]. Animal well-being was assessed daily and tumor growth was monitored in vivo by external measurements with digital caliper until endpoint criteria as regulated in ROB-55.2-2532.Vet_02-19-125 were achieved. Mice were euthanized by isoflurane and cervical dislocation upon achievement of endpoint criteria or end of experiment.

## Tumor rejection potential of TCR-tg T cells

The capacity of primary tumor control was assessed as described before[31]. Briefly, male and female NSG mice at the age of six to nineteen weeks were subcutaneously injected with U698M-mut mg cells or A2058-mut mg cells ($10 \times 10^6$ cells/flank). As tumors reached an area of ca. 20 mm$^2$, T cells transduced with neoTCRs (KIF-P2, KIF-sc1, KIF-sc2, SYT-T1) or T cells transduced with an irrelevant TCR (2.5D6 targeting MPO[84]) were injected intravenously. For the initial setting, a total of $2 \times 10^7$ neoTCR-tg T cells ($3.2 \times 10^7$ absolute T cells including non-transduced cells) administered to each individual of 6 mice per group ($n = 6$). Injection was split on two subsequent days. In this initial setting, IL-15-producing NSO cells were injected intraperitoneally after T cell administration two times per week (irradiated with 80 Gy). To challenge our setting in subsequent experiments, a total of $5 \times 10^6$ neoTCR-tg T cells (KIF-P2 versus KIF-sc1) was administered intravenously into either U698M-mut mg- or A2058-mut mg-tumor bearing mice. Male and female animals as well as animals of different age were distributed evenly across all treatment groups. Tumor growth kinetics were monitored daily for up to 12 weeks with digital caliper until experiment endpoint criteria were reached.

## Ex vivo analysis of TILs on day 5 after T cell injection

On day 5 after T cell injection, animals were sacrificed and tumors as well as spleen explanted. Minced tumors were enzymatically digested for 90 min at 37 °C (Human tumor dissociation kit, Miltenyi Biotec, cat. 130-095-929) and passed through a cell strainer (100 µm) afterwards in parallel to the spleen. Both lysates from tumors as well as spleen were directly used for further analysis.

## Rechallenge model: tumor rejection potential of TIL products generated from transgenic T cells

For the generation of TIL products, male and female NSG mice at the age of seven to twelve weeks were subcutaneously injected with U698M-mut mg cells or A2058-mut mg cells ($10 \times 10^6$ cells). As tumors reached an area of 20 mm$^2$, T cells of different healthy donors (A, B and

 

C) transduced with neoTCRs (mostly KIF-P2 and KIF-sc1 or in later experiments also KIF-P1 and -sc2) were injected intravenously. $8 \times 10^6$ transduced T cells (in total 11 to $32 \times 10^6$ including non-transduced cells) were administered to five to six mice per group at equalized transduction rates for both groups (KIF-P2 versus KIF-sc1 in U698M: 70% for donor A, 60% for donor B, 62% for donor C, KIF-P2 versus KIF-sc1 in A2058-model: 25% for donor A, all KIF-TCRs in U698M-model: 25% for donor A). Tumor growth kinetics were monitored daily for 5 days with digital caliper. On day 5, before tumor regression was measurable, animals were sacrificed and tumors as well as spleens explanted. Minced tumors were enzymatically digested for 90 min at 37 °C (Human tumor dissociation kit, Miltenyi Biotec, cat. 130-095-929) and passed through a cell strainer (100 μm) afterwards in parallel to the spleen. Partly, tumor lysate was used for immediate downstream applications as described above (FACS staining or co-culture). Further parts of the tumor material were cultivated with irradiated (70 Gy) feeder cells, 1000 U/ml human IL-2 and 30 ng/ml anti-CD3 antibody (OKT3) in TCM for 21 days. IL-2 was supplemented on days 7, 11 and 15 (300 U/ml). Efficacy of TIL generation was assessed by FACS staining (CD8 and TCRmu) and TILs of those mice with the highest rate and count of TCRmu$^+$ CD8-TILs per TCR pooled to reach equal transduction rates for all subsequent experiments wherever possible.

For the initial experiments comparing KIF-P2 and -sc1 in U698M, we injected $5 \times 10^6$ transgenic T cells (transduction rate of 55% (donor A), 45% (donor B) or 50% (donor C)) into U698M-tumor bearing mice (equal distribution of male and female, age between 7 and 15 weeks per group). For those experiments, we applied KIF-P2 and KIF-sc1 TCR-tg T cells each for the TIL-P-conditions (injected on day 21 after tumor explant; 43 days after blood donation) as well as a new batch of transgenic T cells (NEW) from the same donor (17 days after blood donation). For the transfer of this model into the melanoma cell line A2058, we generated TIL-P in the described fashion from A2058-tumors and injected $4 \times 10^6$ KIF-P2 and KIF-sc1 TIL-P (transduction rates 22% for KIF-P2 and 15% for KIF-sc1) in parallel to NEW conditions into A2058-tumor bearing hosts. For the comparison of all four KIF2C$^{P13L}$-specific neoTCRs in the U698M-model, we injected $5 \times 10^5$ transgenic T cells of TIL-P from all four TCRs (transduction rates: 2% KIF-sc1, 4.5% KIF-sc2, 8% KIF-P2, 1% KIF-P1) into U698M-tumor bearing mice. In both settings, due to very different TCRmu$^+$ rates of TIL-P conditions no equalized transduction rate was possible; therefore we injected the same amount of TCR-tg, but different absolute amounts of T cells. In all rechallenge experiments, we compared neoTCRs with 2.5D6-tg T cells as negative control. Tumor growth kinetics were monitored daily by blinded measurement regarding T cell condition for up to 12 weeks with digital caliper until experiment endpoint criteria were reached.

**Rechallenge model: in vitro co-cultures on the day of reinjection Multiplex analysis.** In parallel to the injection of TIL-P (and NEW) T cells into tumor-bearing hosts, in vitro co-cultures of TIL-P cells were set up with U698M-mut mg tumor cells. TCR-tg T cells from either the TIL-P or the NEW condition were co-cultured for 24 h with U698M-mut mg tumor cells. Supernatant was collected after 24 h and 13-plex legendplex (biolegend, cat. 741186) analysis of human CD8/NK-cytokines (Granulysin, Perforin, GzmB, GzmA, IFN-γ, sFasL, sFas, TNF, IL-17A, IL-6, IL-10, IL-4 and IL-2) was performed for comparison of KIF-P2 versus -sc1.

**BulkRNA-sequencing and data analysis.** For transcriptomic analysis of TIL-P (KIF-P2, KIF-sc1) and NEW (KIF-P2 and KIF-sc1, 2.5D6) of donor B on the day of reinjection, in vitro co-cultures of $1 \times 10^6$ TIL-P cells (50% TCRmu$^+$) were set up with $1 \times 10^6$ U698M-mut mg or U698M-wt mg tumor cells for 12 h and 24 h in triplicates. At each timepoint, triplicates were pooled and CD8$^+$ T cells purified by CD8-MACS (Miltenyi Biotec). CD8$^+$ T cells were immediately snap frozen and total RNA was

isolated for all co-culture conditions as well as unstimulated T cells using the RNeasy Mini Kit (Quiagen, cat. 74104) according to the manufacturer's instructions. Library preparation for bulk-sequencing of poly(A)-RNA was performed as described previously[86]. In brief, barcoded cDNA was generated with a Maxima RT polymerase (Thermo Fisher) using oligo-dT primer containing barcodes, unique molecular identifiers (UMIs) and an adapter. 5′-Ends of the cDNAs were extended by a template switch oligo (TSO) and full-length cDNA was amplified with primers binding to the TSO site and the adapter. The NEB Ultra II FS kit (NEB, cat. E6177) was used to fragment cDNA. After end repair and A-tailing, a TruSeq adapter was ligated, and 3′-end-fragments were amplified using primers with Illumina P5 and P7 overhangs. In comparison with Parekh et al. the P5 and P7 sites were exchanged to allow sequencing of the cDNA in read1 and barcodes and UMIs in read2 to achieve a better cluster recognition. The library was sequenced on a NextSeq 500 (Illumina) with 57 cycles for the cDNA in read1 and 16 cycles for the barcodes and UMIs in read2. Data were processed using the published Drop-seq pipeline (v1.0) to generate sample- and gene-wise UMI tables[87]. Reference genome (GRCh38) was used for alignment. Transcript and gene definitions were used according to the GENCODE version 38.

The raw gene counts were normalized through the vst method of the DESeq2[88] (v. 1.30.1) R package for the subsequent principal component analysis (PCA) calculation. The first two PCA dimensions were used for the visualization. For the expression comparison, the raw counts values were normalized through the median of ratios method of the DESeq2 package by dividing the counts by sample-specific size factors determined by the median ratio of gene counts relative to the geometric mean per gene. Subsequently, the normalized values were divided by the expression values of the 2.5D6 T cell clones per group by adding the pseudo-counts values, thus eliminating unspecific signaling signatures due to cell culture-specific or processing issues. All plots were generated with the help of the ggplot2 (v. 3.4.2) R package.

**Statistics**
Significance of differences between TCRmu$^+$ frequencies of transgenic T cells, TCRmu transcripts, EC$_{50}$-values, half-life of $k_{off}$-rates as well as TCR-TIL-P products on the day of TIL reinjection (FACS analysis and IL-10 secretion) were investigated by one-way analysis of variance (ANOVA) and Tukey's multiple comparison test. Significance of differences between TCRmu$^+$ frequencies and count in the tumor or spleen on day 5 of sacrifice was calculated by Student's t-test. Regarding the rechallenge model, differences in tumor growth were calculated for the tumor area on day 14 to 17 with ordinary one-way ANOVA and Tukey's multiple comparison test or for the rechallenge of OTR-TIL-P on day 15 with two-tailed, unpaired t-test. Statistical comparison of survival was performed using the Mantel-Cox test. Statistical analyses were performed with GraphPad Prism V.9.3.1 software.

**Reporting summary**
Further information on research design is available in the Nature Portfolio Reporting Summary linked to this article.

## Data availability
The scRNA and TCR-seq have been deposited to the European Genome-phenome Archive (EGA) with the study identifier EGAS50000000600 and are available upon request from the associated Data Access Committee (EGAC00001000546) due to the patient information contained under controlled access. Access will be granted to commercial and non-commercial parties according to patient consent forms and data transfer agreements. The mass spectrometry proteomics data generated in this study have been deposited in the ProteomeXchange Consortium via the PRIDE partner repository with the dataset identifier PXD051734. All further data generated in this

study are provided in the Supplementary Information or Source Data file or from the corresponding author upon request. Source data are provided with this paper.

## Code availability

Scripts required for the transcriptomic analysis replication with the corresponding clonotype information are deposited upon the link to a Github repository: https://github.com/beltranLab/NeoTCR_Beltran_Krackhardt.

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

## Acknowledgements

The authors thank the patient for participating in the study and his continuous support. We also thank Stefanie Stein for excellent technical support. This project has been supported by the European Institute of Innovation & Technology (EIT) Health 19638 (AMK), Deutsche Forschungsgemeinschaft (DFG, German Research Foundation)—SFB824 (C10 AMK), SFB- TRR 338/1 2021—452881907 (A03 AMK/A01 DHB) and INST 95/1650-1 (PG)—as well as Else Kröner-Fresenius-Stiftung, doctoral program "Translationale Medizin" (JU).

## Author contributions

Conceptualization: F.F., J.U., E.Br. and A.M.K.; Methodology: F.F., J.U., E.Br., V.K., S.B., S.J., C.V., N.d.A.K., D.G., R.Ö., P.G., Ed.B. and A.M.K.; Experimental work: F.F., J.U., E.Br., S.J., G.Z., S.B., C.V., A.S., D.G., R.Ö., P.G. and M.H.; Data analysis: F.F., J.U., E.Br., V.K., G.Z., S.B., A.S., N.d.A.K., P.G., M.H., Ed.B. and A.M.K.; Visualization: F.F., J.U., E.Br., V.K., G.Z., A.S. and A.M.K.; Funding acquisition: D.H.B., A.M.K. and P.G.; Project administration: E.Br. and A.M.K.; Supervision: R.R., D.H.B., Ed.B., E.Br. and A.M.K.; Writing—original draft: F.F., J.U., E.Br. and A.M.K.; Writing—review & editing: all authors.

## Funding

## Competing interests

The authors declare no competing interests.

## Additional information

[1]Technical University of Munich, School of Medicine and Health, III Medical Department, TUM University Hospital, Ismaninger Str. 22, 81675 Munich, Germany. [2]Ludwig-Maximilians-Universität München, Institute of Clinical Neuroimmunology, University Hospital, Marchioninistr. 15, 81377 Munich, Germany. [3]Ludwig-Maximilians-Universität München, Faculty of Medicine, Biomedical Center (BMC), Großhaderner Str. 9, 82152 Martinsried, Germany. [4]Technische Universität München, Institute for Medical Microbiology, Immunology and Hygiene, Trogerstr. 30, 81675 Munich, Germany. [5]Technische Universität München, Institute of Molecular Oncology and Functional Genomics, TUM School of Medicine and Health, Ismaninger Str. 22, 81675 Munich, Germany. [6]Technical University of Munich, Center for Translational Cancer Research (TranslaTUM), TUM School of Medicine and Health, Einsteinstr. 25, 81675 Munich, Germany. [7]Bavarian Center for Biomolecular Mass Spectrometry at Klinikum rechts der Isar, Technical University of Munich, Einsteinstr. 25, 81675 Munich, Germany. [8]German Cancer Consortium (DKTK), Partner-Site Munich and German Cancer Research Center (DKFZ), Im Neuenheimer Feld 280, 69120 Heidelberg, Germany. [9]German Center for Infection Research (Deutsches Zentrum für Infektionsforschung, DZIF), Partner Site Munich, Munich, Germany. [10]Munich Cluster of Systems Neurology (SyNergy), Feodor-Lynen-Str. 17, 81377 Munich, Germany. [11]Malteser Krankenhaus St. Franziskus-Hospital, Flensburg, Germany. [12]These authors contributed equally: Franziska Füchsl, Johannes Untch. [13]These authors jointly supervised this work: Eva Bräunlein, Angela M. Krackhardt. ✉e-mail: angela.krackhardt@tum.de

