## [Peer Review File · Nature Communications]

High-resolution profile of neoantigen-specific TCR activation links moderate stimulation to increased resilience of engineered TCR-T cellsEditorial note: parts of this Peer Review File have been redacted as indicated to maintain patient confidentiality.

REVIEWER COMMENTS

Reviewer #1 (Remarks to the Author):

The identification of the properties of optimal antitumor TCRs is important for the development of TCR gene therapy. Fuchs et al provide deep characterization of TCRs targeting private neoantigens from a previously reported patient with melanoma in an attempt to address this issue. The authors demonstrate that TCRs with high functional avidity against neoantigen lead to strong T-cell activation, upregulation of activation/exhaustion markers and induction of an inhibitory transcriptomic program. The high functional avidity TCRs appear to display inferior antitumor activity in an in vivo mouse tumor model compared to TCRs with moderate functional avidity. While these observations are interesting, it is difficult to make strong conclusions about the impact of TCR functional avidity and antitumor responses due to the size of the study and the models employed. The study might be strengthened by addressing the following points.

1. The identification of optimal TCRs for use in therapy is complex and there are contradictory findings in the literature. Some studies suggest the highest functional avidity TCRs are best in a leukemia mouse model (e.g., Vincent et al, Biol Blood Marrow Transplant, PMID: 24161924), others believe the best TCRs in solid tumor mouse models follow a “Goldilocks” story (e.g., Shakiba et al, JEM, PMID: 34935874), and yet others believe that TCR affinity/avidity play little role, and that epitope density on the tumor (mouse lymphoma cells) is a more important factor (e.g., Segal et al, JI, PMID: 27036915). In the clinic, the transfer of very high functional avidity TILs against mutant KRAS led to durable tumor regression (e.g., Tran et al, NEJM, PMID: 27959684; and Sim et al, PNAS, PMID: 32461371). It seems likely that the impact of TCR avidity/affinity on antitumor responses may be model specific, and thus the conclusions of the current manuscript may be limited.

a. What is the neoepitope density on the U698M tumor, and is this within similar physiologic densities of neoepitopes presented by human solid cancers? Does epitope density impact therapeutic efficacy in this model?

b. The U698M is a B-cell lymphoma which likely can act as an APC and provide costimulatory signals to T cells. Why did the authors choose malignant B cells rather than a solid tumor model? Would the efficacy of the TCRs be different in the setting of a solid cancer model

that doesn't provide costimulatory signals? E.g., would the more highly functionally avid TCR perform better in this (more relevant) setting?

c. The authors evaluate 2 TCRs in their rechallenge model and make the conclusion that moderate avidity is better than higher avidity. But with only 2 TCRs it's difficult to make a strong point that avidity is important. Why didn't the authors use all 4 KIF-reactive TCRs and see if the functional avidity directly correlates with in vivo response?

d. The tumor rechallenge model is a bit peculiar. The transduced T cells are injected into tumor bearing mice, and then 5 days later, tumors are taken out and TIL undergo rapid expansion. Could re-stimulating the "TIL" so soon after antigen/TCR triggering in vivo preferentially negatively impact the higher functional avidity TCR Tg cells (e.g., too soon or too much/strong TCR stimulation)? TIL in humans likely have been in the tumor for much longer periods of time, and importantly in the setting of tumor progression, rather than tumor regression as in the mouse model, which could impact their phenotype.

e. A major utility of the authors' work is in the setting of TCR gene therapy rather than TIL therapy; therefore, have the authors developed a model that would allow for the evaluation of efficacy of different functionally avid TCRs against a tumor (e.g., Fig. 5A, but in a situation where not all the TCRs mediate complete tumor rejection). For example, have the authors titrated the T cell doses, treated larger tumors, or as alluded to in (b) above, tested this in a solid cancer model?

2. Do all the TCR engineered T-cell products use in the study express the same level of Tg-TCR (e.g., the same MFI)? This may impact functional avidity.

3. Related to the above, while most TCR gene therapy studies use retroviral vectors to express the TCR, do the authors know if transgenic overexpression of TCRs using a viral promoter behave similarly (signaling, regulation, transcriptome, exhaustion, etc) to endogenous TCRs? E.g., how would this compare to a Crispr knock-in of TCRs to the endogenous TCR locus? Are the authors' overall findings relevant only in the context of retroviral insertion of TCRs?

4. Did the authors evaluate affinity of TCR against pMHC? How does functional avidity compare to affinity?

5. How do the authors define moderate avidity? All of the KIF-reactive TCRs are within ~3 fold EC50. What would be low or high?

6. While the single-cell RNA-seq experiments are interesting, there are some limitations:

a. Given the above points, it is unclear that the RNA-seq signatures after in vitro stimulation will be helpful in predicting the optimal TCRs to use for therapy, which was a major goal of the study. Prospective use of the signatures to identify TCRs and show they provide better antitumor immunity than TCRs derived from other signatures would be needed to validate this.

b. Are there peptide concentration effects on the transcriptome of the T cells?

c. Results, line 162: the negative control are “freshly thawed CD8+ T cells”, wouldn't the best negative control be the same T cells, just unstimulated?

7. Introduction, line 68: reference 7 and 8 both refer to TIL studies. Leidner et al., PMID: 35648703 refers the TCR gene therapy targeting KRAS.

8. Results, line 317: it would be helpful to put details related to the transduction efficiency of the different TIL used for infusion in the Results or figure legend (this is in the methods, but it would help the readers for it to be in Results or legend).

Reviewer #2 (Remarks to the Author):

The manuscript by Fuchs et al describes a study where the investigators isolate neoantigen T cells from a disease-free melanoma patient undergoing checkpoint blockade using pembrolizumab. PBL-derived T cells were neoantigen stimulated and the cultures were enriched for CD137+ cells, then rapidly expanded prior to use. Neoantigen reactive T cell cultures were subjected to scRNAseq and scTCRseq to examine the relationship between the TCR expressed and biologic function. New neoantigen reactive TCR's were cloned and their function was in vitro and in vivo was evaluated against neoantigen expressing targets.

The authors conclude that T cells expressing different neoantigen TCR's have different transcriptional signatures and different antigen recognition potentials, even when targeting

the same neoantigen peptide. Furthermore, T cells expressing neoantigen T cells with moderate signaling have superior control of tumor growth in their xenograft models. The authors argue that their analysis will help guide the field in improving adoptive T cell transfer for patients with advanced cancer.

While the authors data generally supports their conclusions, there are concerns with this study.

1) Isolating T cells from a patient undergoing checkpoint blockade may not be representative of the T cells in a typical cancer patient. Since the patient was disease free, it is also not clear that T cells are representative of a cancer patient.

2) Since the authors used in vitro priming, CD137 selection, and a rapid expansion prior to scRNAseq, the repertoire is not likely to be reflective of the TCR repertoire and the T cell physiology is not likely to be reflective of the native T cells.

3) Most of the analysis was using peptide loaded targets. Furthermore, the tumor cells used were transfected with a mini-gene. None of these are physiologically normal. There should be at least some experiments to show that the T cells, whether native or TCR Tg, recognize physiologically real levels of antigen. Otherwise, the message is overall relevance less convincing.

4) With the TCR Tg T cells, the authors should consider how false pairing and culture conditions impact their results. Also, how is the level of TCR transgene expression impact the results?

Reviewer #3 (Remarks to the Author):

The authors in this manuscript investigated the different properties of neoantigen-specific TCRs and claimed that moderate stimulation of TCR is linked to resilience and sustained tumor control. In this manuscript however many controls and additional data are missing to conclude this.

Major:

This is a follow up to previous publication; maybe better to transform it to letter format?

Figure 1, 2, 3: I miss what is the point of putting this data in this manuscript. Why 3 figures?

It is difficult to extract the distinct messages of figures 1, 2 and 3.

Figure 4:

Controls missing in figure 4a, 4b, 4c, 4h: no target and with target but no peptide. What about endogenous antigen presentation? This last point is highly relevant

What is the expression of retrovirally introduced TCRs, are they all efficiently and comparably expressed? Percentage wise and also level of expression (gMFI).

Please show LAG3 MFI in addition to percentage

The conclusion of figure 4 is that sc1 is more activated and more inhibited; however, I do not see evidence for higher level of inhibition. (no functional inhibition shown, only more activation)

Only one E:T ratio shown; choosing more E:T ratios would have provided more insight

Figure 5:

It is impossible to draw any conclusions without showing a detailed analysis of the TIL product before infusion.

What is the phenotype of TIL products? Major thing not considered: differentiation. What is the differentiation state, as in what is the frequency of memory/effector cells in the TILs before expansion as well as after ex vivo expansion?

What is the percentage of mTCR+ cells after the expansion of the TILs? What is the HLA typing of the donor T cells and the tumor cells.

Why did the authors choose for this specific TIL expansion protocol (OKT3)? Would the authors expect different results when choosing for a CD3CD28 based expansion protocol?

What would have been interesting: scRNA of TIL T cells...

In figure 4 the authors chose to show percentage rather than MFI; in figure 5 the authors showed MFI rather than percentage. For both the data of figure 4 and figure 5, please show either percentage or MFI in the main figure and show the other respective type in the supplement.

Positive control is missing for figure 5E 5F and 5G; (for example anti-CD3/CD28 beads,

OKT3).

Why do the authors chose for a B-cell lymphoma cell line rather than a melanoma cell line? If I understood correctly from materials and methods, the authors generated TIL products for all 5 mice per group and pooled TIL products from those two mice per group that showed highest mTCR frequency. Why no mTCR enrichment step? What were the frequencies of all TIL products generated from all mice? What were the expansion curves of TIL products? All these data would be important to make an informed judgement about what is happening between tumor explantation and TIL generation until reinfusion of the product.

The conclusions made in this manuscript are based on one single in vivo experiment with one T cell donor and two TCRs (figure 5). To make such a strong conclusions these experiments have to be repeated with additional T cell donors and with other TCRs also exhibiting differences in stimulation levels.

Minor:

Some sentences should be simplified to improve readability, for example line 100: Thereby, we are not only able to show that diverse activation patterns detected in scRNA- and scTCR-seq of primary T cells are mirrored by in vitro and in vivo data of T cells transgenic for defined neoTCRs indicating significant stability in structural functionality.

Material and methods:

What do the authors mean by

2×10^7 transduced T cells (3.2×10^7 absolute T cells including non-transduced cells) were administered to 6 mice per group ($n = 6$) in two injections on two subsequent days.

Point-to-point reply to reviewer comments (NCOMMS-23-01734)

Reviewer #1 (Remarks to the Author):

The identification of the properties of optimal antitumor TCRs is important for the development of TCR gene therapy. Fuchs et al provide deep characterization of TCRs targeting private neoantigens from a previously reported patient with melanoma in an attempt to address this issue. The authors demonstrate that TCRs with high functional avidity against neoantigen lead to strong T-cell activation, upregulation of activation/exhaustion markers and induction of an inhibitory transcriptomic program. The high functional avidity TCRs appear to display inferior antitumor activity in an in vivo mouse tumor model compared to TCRs with moderate functional avidity. While these observations are interesting, it is difficult to make strong conclusions about the impact of TCR functional avidity and antitumor responses due to the size of the study and the models employed. The study might be strengthened by addressing the following points.

1. The identification of optimal TCRs for use in therapy is complex and there are contradictory findings in the literature. Some studies suggest the highest functional avidity TCRs are best in a leukemia mouse model (e.g., Vincent et al, Biol Blood Marrow Transplant, PMID: 24161924), others believe the best TCRs in solid tumor mouse models follow a “goldilocks” story (e.g., Shakiba et al, JEM, PMID: 34935874), and yet others believe that TCR affinity/avidity play little role, and that epitope density on the tumor (mouse lymphoma cells) is a more important factor (e.g., Segal et al, JI, PMID: 27036915). In the clinic, the transfer of very high functional avidity TILs against mutant KRAS led to durable tumor regression (e.g., Tran et al, NEJM, PMID: 27959684; and Sim et al, PNAS, PMID: 32461371). It seems likely that the impact of TCR avidity/affinity on antitumor responses may be model specific, and thus the conclusions of the current manuscript may be limited.

We thank the reviewer for this important comment and agree on the complex interactions of TCR avidity (and affinity) with antitumor response, which is a central focus of our manuscript. As one of the first studies, we compare different neoTCRs recognizing the same neoepitope in the identical HLA context on the same target cells in such depth. Thereby, we provide detailed TCR-centered functional analyses on unexpectedly heterogenous TCRs including avidity as one aspect of TCR-characterization. In line with the inconsistencies between different model systems outlined by the reviewer, we are convinced that our understanding of natural human TCR-dependent neoantigen reactivity will significantly improve by such detailed analyses on model TCRs in diverse murine models providing crucial impact on the TCR-engineering field.

While the majority of single-cell sequencing approaches employ TIL signatures to improve identification pipelines for tumor-specific TCRs¹⁻⁶, we aimed at a neoantigen-centric approach using previously identified naturally presented neoepitopes as stimulus. We confirmed robust detection of previously known T cell clonotypes with defined neoTCR and additionally identified novel TCR clonotypes. As most striking clinically related finding, all known neoTCRs from this patient show noticeably different frequencies in various tissue compartments. The comparatively minor avidity differences between these TCRs thus suggest that other structural aspects of TCRs and factors associated to defined TCR signaling might influence sustained T cell fitness. We are aware that the neoTCRs we focused on rank on the lower end of ranges of functional avidity defined in other recent publications^{7,8}, although comparability between functional avidity measurements in different studies might be limited. By now incorporating an orthotopic TCR replacement model into our analyses, we show that this setting generally follows the trends outlined within our previous analyses using retroviral gene transfer (see answer to question 2). We also include epitope density and a variety of stimulation modes in our model confirming the general reactivity patterns previously observed.

We believe, that by incorporating this additional information in our manuscript we can address and control many variables driving model-specific conclusions on TCR-affinity/avidity and antitumor response. As a result, the picture of neoTCR reactivity demonstrates even more complex layers of specific responses emphasizing the importance of individualized, patient-centric approaches for selection of most suitable neoTCR candidates.

a. What is the neoepitope density on the U698M tumor, and is this within similar physiologic densities of neoepitopes presented by human solid cancers? Does epitope density impact therapeutic efficacy in this model?

We thank the reviewer for this valuable comment. Indeed, (neo)epitope density plays a crucial role in the capability of tumor recognition and reactivity^{9,10}. The density and dynamics of neoepitope surface presentation depend on many aspects including, but not limited to, the antigen presentation machinery, MHC turnover or peptide-MHC stability. Consequently, epitope presentation varies from tumor to tumor and even within intra-tumoral sub-clonal populations. Thus, in vitro models will always only partially reflect the complexity of antigen presentation within human tumors and this question cannot be answered generally.

By pulsing with different doses of mutated peptide we took neoantigen density into account knowing, that this method can neither reflect the clonal heterogeneity of a patient's tumor not on the cell-intrinsic presentation machinery. However, we detected a consistent dose-dependent pattern of activation strength for the neoTCRs (Fig 3A-D, S7, S8). Furthermore, we have strengthened our approach by testing T cell response to different cell lines with varying characteristics of antigen presentation. Of note, the lymphoma cell line U698M serves as a rather "hard to treat" model, since we observed comparably low MHC expression in relation to the other cell lines used (Figure S10).

To exclude overexpression by artificial gene transfer, we aimed at a single nucleotide variant (SNV) substitution for endogenous epitope presentation under the natural KIF2C promotor. We employed the CRISPR/Cas9-based strategy of prime editing^{11,12} to modify a nucleotide from C to T at the orthotopic KIF2C locus of our target cell lines with confirmed high KIF2C expression level (Figure R1). Despite several protocol adaptations, we have not succeeded in generating SNV-modified cell lines so far. However, also this approach has several limitations. Next to potential off-target reactivity¹³, primary human tumor cells may inherit complex aberration in their antigen processing machinery impacting presentation of defined neoantigens, which cannot be adequately mirrored by cell lines. In addition, the reproduction of intra- and intertumoral heterogeneity of primary cancer manifestations for experimental validation still poses a challenge to be solved in the future. We therefore think, that directed orthotopic nucleotide modification of KIF2C in our cell lines will provide only limited additional information with respect to the focus of this manuscript and therefore think that we need more complex tumor models which is, however, beyond the scope of this work.

b. The U698M is a B-cell lymphoma which likely can act as an APC and provide costimulatory signals to T cells. Why did the authors choose malignant B cells rather than a solid tumor model? Would the efficacy of the TCRs be different in the setting of a solid cancer model that doesn't provide costimulatory signals? E.g., would the more highly functionally avid TCR perform better in this (more relevant) setting?

We agree that any model comes along with certain limitations. The U698M lymphoma cell line was selected to investigate KIF2C^{P13L}- as well as SYTL4^{S363F}-specific immune responses head-to-head and, since the two epitopes are restricted to different HLAs (A*03:01 and B*27:05). U698M was the only available cell line carrying both desired alleles. We thereby followed an entity-agnostic approach focused on efficient neoantigen processing and expression. In addition, the U698M cell line poses a rather difficult to treat and challenging model with comparably low MHC expression, which we

included in the supplementary data of this manuscript (Figure S10). We also tested different cell lines to investigate the principle of target recognition, such as peptide pulsed LCL (Mel15 LCL) and the myeloma cell line JJN3.

To exclude additional entity- or tumor-specific aspects as recommended by the reviewer, we now also included data on reactivity against the neoantigen-expressing melanoma cell line A2058, which carries the A*03:01 allele and therefore serves as a suitable model for all KIF2C^{P13L}-specific TCRs. We integrated this data in Figures 4 and 5 and observed efficient tumor killing upon first tumor encounter for both, KIF-P2 as well as KIF-sc1 in vitro (Figure S16I) and in vivo (Figure 4F, G). Mirroring our lymphoma model, we also detected more pronounced impairment of tumor killing for the TCR KIF-sc1 upon restimulation (Figure 5G, H). Similarly, the functional differences we detected for the reinjected TIL-P cells were reflected in the melanoma model (Figure S15D, E). Overall, we can confirm that the differences between both neoTCRs were reflected in different models emphasizing that the effect is not dependent on one tumor entity or cell line.

While it is essential to understand factors influencing tumor reactivity and escape mechanisms, especially those associated with the individual tumor microenvironment (TME), a detailed TME analysis is beyond the focus of the current manuscript.

c. The authors evaluate 2 TCRs in their rechallenge model and make the conclusion that moderate avidity is better than higher avidity. But with only 2 TCRs it's difficult to make a strong point that avidity is important. Why didn't the authors use all 4 KIF-reactive TCRs and see if the functional avidity directly correlates with in vivo response?

We strongly concur with the reviewer's view that beyond avidity, numerous aspects may play a decisive role in the reactivity of defined TCRs. Based on the observed differences of neoTCR prevalence in the patient's primary tissues as well as in vitro reactivity, we focused on those two TCRs, KIF-P2 and KIF-sc1, demonstrating moderate versus strong activation patterns. We identified an unexpected inverse correlation of stimulation strength (and thereby also functional avidity) with reactivity after repeated antigen stimulations in vivo.

This observation is further supported by more recent data, where we investigated, as recommended by the reviewer, the potency of all KIF2C^{P13L}-reactive neoTCRs (KIF-P1, -P2, -sc1 and -sc2) side-by-side upon in vivo rechallenge with a lower number of effector T cells (0.5×10^6 transduced cells, Figure S16A, B). As observed before, the neoTCR-dependent spectrum of T cell activation translated directly into patterns of tumor rejection within the first two weeks after T cell injection with smaller tumor size on day 14 for KIF-P1 and -P2 compared to KIF-sc1 and -sc2. The neoTCRs with a moderate activation profile, headed by KIF-P1 and -P2, demonstrated a statistically significant survival benefit in comparison to control TCR 2.5D6 (Figure S16B). Thus, we propose to link robust in vivo persistence and superior long-term tumor control after repeated antigen challenge to a moderate activation profile rather than a defined functional avidity. One strength of this work is that in this novel model of T cell resilience the investigated spectrum of patient derived TCRs targeting the same neoantigen opens new, currently neglected important aspects. Yet, emphasizing the extraordinary features of KIF-P1 even draws a more complex picture and confirms that reliable biomarkers for streamline neoTCR identification and engineering for clinical applications need further investigation.

D. The tumor rechallenge model is a bit peculiar. The transduced T cells are injected into tumor bearing mice, and then 5 days later, tumors are taken out and TIL undergo rapid expansion. Could re-stimulating the "TIL" so soon after antigen/TCR triggering in vivo preferentially negatively impact the higher functional avidity TCR Tg cells (e.g., too soon or too much/strong TCR stimulation)? TIL in humans likely have been in the tumor for much longer periods of time, and importantly in the setting of tumor progression, rather than tumor regression as in the mouse model, which could impact their phenotype.

We agree with the reviewer's opinion that our *in vivo* experiments model early restimulation of TILs and depict a likely shorter timeframe compared to endogenous (human) TILs in the classical sense. Yet, dwell time and recirculation of TILs in and from a tumor remain to be further elucidated and (human) tumor infiltration by lymphocytes comprises heterogeneous interactions, which cannot be reflected by such a xenograft model. Instead, we aimed to create an *in vivo* model which challenges the identified TCRs repeatedly to see whether we can observe significant differences invisible upon initial stimulation. In fact, we found that neoTCRs with moderate activation pattern mediate superior tumor control which may help explain different frequencies in the primary patient tissue (Figure 1D). Indeed, we demonstrated with the model chosen herein that upon repeated antigen stimulation neoTCRs significantly differ in their subsequent *in vivo* tumor rejection capacity. In this regard, our model primarily depicts a setting to repeatedly challenge TCR-tg T cells and to detect TCR-inherent qualitative differences even after such short time of TCR-tg T cells in the tumor. At the same time, we do not intend to make a statement about the functionality or dysfunction of heterogeneous, endogenous human TIL populations developing over the course of years in a complex TME.

e. A major utility of the authors' work is in the setting of TCR gene therapy rather than TIL therapy; therefore, have the authors developed a model that would allow for the evaluation of efficacy of different functionally avid TCRs against a tumor (e.g., Fig. 5A, but in a situation where not all the TCRs mediate complete tumor rejection). For example, have the authors titrated the T cell doses, treated larger tumors, or as alluded to in (b) above, tested this in a solid cancer model?

We thank the reviewer for this important remark. To further emphasize this approach, we now included additional data showing a side-by-side comparison of a reduction of effector T cells from 30×10^6 (Figure 4B, C) to 5×10^6 TCR μ^+ T cells (Figure 4D, E). Even upon lower doses of TCR-tg T cells we observed similar tumor killing patterns for different TCRs (KIF-P2 and KIF-sc1).

As suggested by the reviewer, we further compared the lymphoma model to a melanoma model in NSG mice (see comment 1b). Both, KIF-P2 and KIF-sc1, similarly mediated tumor regression upon first encounter in all melanoma-bearing hosts (5×10^6 TCR-tg T cells, Figure 4F, G). It stresses the importance of our findings, that upon second tumor encounter in the melanoma model, the moderate TCR KIF-P2 again outperforms KIF-sc1 (Figure 5G, H).

2. Do all the TCR engineered T-cell products use in the study express the same level of Tg-TCR (e.g., the same MFI)? This may impact functional avidity.

We thank the reviewer for making this valuable point. Indeed, the TCRs differ in their extra- and intracellular expression after retroviral transduction as captured by TCR μ -staining (Figure S4A-C) which most likely impacts on functional avidity. These differences between TCRs remained highly construct-specific and stable over the course of many transductions. Interestingly, neither the absolute quantity of TCR-transcripts (Figure S4D) nor by the TCR-insertion frequency (Figure S4E) entirely explained those stable differences in surface expression in the retroviral (RV) system. Of note, particularly the TCR KIF-P1 displayed surprisingly convincing killing capacity (Figure 4A-C, Figure S16A, B, I) despite markedly lower surface expression in the RV system. To avoid bias in TCR expression by the CMV-promotor of the RV system, we therefore now additionally expressed the neoTCRs under the natural TCR alpha promotor and performed orthotopic T cell receptor replacement (OTR)^{14,15} for all four KIF2C^{P13L}-reactive TCRs (compared to a control TCR) and investigated their function side-by-side with retrovirally transduced TCR and integrated these data into the revised manuscript (Figure S4F). Our analyses detected a similar level of TCR μ -surface expression per cell within the TCR-tg population in both systems for KIF-P2 and KIF-

sc1 (Figure S4G). Thus, expression differences between KIF-P2 and -sc1 seem construct-inherent and comparable under both promoters. Differences in TCR expression may contribute to the differences observed, however, this is rather independent of the mode of genetic modification. In contrast, other TCRs – KIF-P1 in particular – clearly profited from the OTR system (Figure S4G) adding further complexity to the combination of TCR-construct and expression system/promotor.

3. Related to the above, while most TCR gene therapy studies use retroviral vectors to express the TCR, do the authors know if transgenic overexpression of TCRs using a viral promoter behave similarly (signaling, regulation, transcriptome, exhaustion, etc) to endogenous TCRs? E.g., how would this compare to a Crispr knock-in of TCRs to the endogenous TCR locus? Are the authors' overall findings relevant only in the context of retroviral insertion of TCRs?

We agree to the relevance of these experiments to our analysis and as outlined above (comment to 2). To specifically meet these concerns, we performed orthotopic T cell receptor replacement (OTR) via CRISPR/Cas9-Knock-in^{14,15} for all four KIF2C-reactive TCRs (compared to a control TCR) side-by-side with a retroviral (RV) transduction (Figure S4F). We performed in-vitro co-cultures using A2058 expressing mutated and wildtype minigenes and measured FACS-based expression of CD137 and LAG3 as correlates for activation and inhibition. Despite small differences, both OTR engineered, and RV transduced T cells yielded similar expression results in this experimental setting, leading us to draw similar conclusions from both engineering methods (Figure S6). However, all genetic engineering strategies harbors limitations in mimicking the natural TCR functionality and are accompanied by method-related off-target influences. Thus, both engineering systems differ from the original patient-derived clonotypes due to their artificial TCR insertion, a limitation that we now additionally outlined in the discussion of the manuscript (lines 540-542).

4. Did the authors evaluate affinity of TCR against pMHC? How does functional avidity compare to affinity?

Indeed, we assessed affinity/structural avidity of the TCRs against the pMHC using multimers and measuring k_{off} rates for RV and OTR TCRs. While KIF-P2, -sc1 and -sc2 showed comparable k_{off} rates in both systems (Figure S5F-G), KIF-P1 displayed a strongly increased structural avidity/ molecular affinity compared to all other TCRs (P1 $t_{1/2}$ = 323.5s OTR, 463.3s RV; Figure S16G, H). While other recent studies identified structural avidity as major determining factor for tumor tropism of TCR clonotypes⁷, or findings on the one hand, render the picture of this patient's TCRs even more complex, on the other hand, however, shows that the differences in our rechallenge model between KIF-P2 and -sc1 (and -sc2) cannot be explained by affinity alone. Considering on the one hand only very slight differences in functional avidity (see also comment to the next question 5) and on the other hand structural avidity between the neoTCRs central for our comparison – KIF-P2 and -sc1 – both factors do not sufficiently explain the phenotype we detect. Overall, our findings suggest stimulation patterns of an individual TCR as a likely complex equation of different variables which might compensate for each other and in their diversity be fittest for different settings.

5. How do the authors define moderate avidity? All of the KIF-reactive TCRs are within ~3 fold EC50. What would be low or high?

It is correct, that the functional avidities of this set of neoTCRs reflect a narrow range. It is difficult to understand, whether and to which extent the range between “high” and “low” functional avidity across analyses of different groups can be compared and how larger differences in functional avidity impact on prolonged tumor control in a rechallenge model as ours. The approximate EC₅₀ of the neoTCRs in our analysis ranked between $10^{-6.5}$ - $10^{-7.5}$, whereas Purcarea et al. reported EC₅₀-values between $10^{-9.5}$ - $10^{-11.5}$

in an OT-I mouse model or 10^{-8} - 10^{-9} or $10^{-6.5}$ - 10^{-7} in two different melanoma patients ⁸. Also, in comparison to another recent study, the neoTCRs of Mel15 rank at the lower end of functional avidity scales (neoTCRs: EC_{50} 10^{-6} to 10^{-9} , viral TCRs: 10^{-7} to 10^{-11}) ⁷. The neoTCRs from our patient in fact differ only within a narrow range of functional avidity, despite significant differences in functionality, thus emphasizing the importance of different functional activation patterns rather than functional avidity alone. We adapted the discussion of our manuscript accordingly to include a better comparison with current literature (lines 610-619, 625-629).

Focusing on Mel15 as a clinically very interesting case (outlined in response to question 1 of reviewer 2), we demonstrate, that within such a narrow range of functional avidity, significant functional differences upon rechallenge of neoTCR-tg T cell products arise in different cancer models with engineered antigen expression (Figure 5B, G). Our data suggest that beyond avidity and affinity (as explained in the comment to the previous question), other structural aspects of TCR-peptide-MHC-interaction (e. g. binding pattern, angle of the TCR-peptide-MHC complex, etc. ^{16,17}) may influence TCR activation patterns and therefore anti-tumor functionality. Thus, we decided to categorize TCRs rather descriptively based on their activation pattern in scRNA-/TCR-seq as well as in vitro stimulation experiments rather than based on functional or structural avidity. This led us to differentiate between moderate versus strong activation. Further analyses on the decisive factors distinguishing the functionality of these neoTCRs will be subject to future investigations.

6. While the single-cell RNA-seq experiments are interesting, there are some limitations:
a. Given the above points, it is unclear that the RNA-seq signatures after in vitro stimulation will be helpful in predicting the optimal TCRs to use for therapy, which was a major goal of the study. Prospective use of the signatures to identify TCRs and show they provide better antitumor immunity than TCRs derived from other signatures would be needed to validate this.

We completely agree with the reviewer on certain limitations of this single cell approach. However, we want to highlight certain valuable strengths of our setup. Firstly, we are convinced that the in vitro stimulation is a necessary and valuable part of our assay. Due to the highly different frequencies of neoTCRs in the patient, an in vitro stimulation allowed for the expansion and reliable detection of additional neoTCRs, especially also the rare clonotypes. Moreover, while prospective use of neoTCR transcriptomic sequences might be of great value for the identification of new neoTCRs²⁻⁶, they cannot recapitulate the initial activation of these T cells after the very first tumor contact. Rare peripheral or intra-tumoral neoTCR-T cell populations might appear either more homogenous in a non-activated state post tumor encounter or more heterogenous due to many different timepoints of tumor encounter than immediately after stimulation. The in vitro stimulation in our setting “overwrites” previous states (especially since the cells necessarily experiences in vitro expansion beforehand) and therefore rather illustrates TCR- and T cell-inherent activation capacity at one, known timepoint of T cell activation. We are, of course, aware that the previous fate of each T cell influences its functional capacity and surely also its transcriptome upon restimulation as we also state in this manuscript. This is why we based the categorization of these neoTCRs only partially on the RNAseq-data and augmented them with functional in vitro data for a more integrative synthesis of a TCR-activation spectrum.

Secondly, we are aware that our study is centered around one patient with very long monitoring time and very good response to CPI (see also response to comment 1 from reviewer #2). To our knowledge not many other studies illuminate features of several TCRs from one patient in such depth. We believe that such detailed assessment of single TCRs and case studies are valuable to unravel slight functional differences between different receptors within the TCR repertoires of therapy responders supporting the understanding of the broad responsiveness and qualities impacting sustainable anti-tumor responses. We agree with the reviewer, that it would be of tremendous interest to perform this pipeline in other CPI responders like Mel15 in

comparison to non-responders to further sharpen not only the general applicability of our model, but also potential additional differences between “successful” and “non-successful” TCRs for potent tumor eradication. However, this remains beyond the scope of this manuscript and we hope to contribute to future studies by providing these data and hypotheses to the field.

b. Are there peptide concentration effects on the transcriptome of the T cells?

Similar to the effects we demonstrated by FACS staining on a protein level (Figure 3A-H, S7, S8), we are convinced that the strength of stimulus, either set by peptide dose or effector-to-target ratio, influences the transcriptome of the T cells. For our setting, we chose a comparably low peptide dose for the scRNA-/TCR-seq experiments to avoid overstimulation and activation-induced cell death and enable subsequent potent proliferation. On the other hand, as our titrations on protein levels confirm (Figure 3A-D), certain differences between neoTCRs are more prominent upon stimuli below the maximum of T cell activation (e. g. LAG-3 expression upon 0.01 and 1 μ M peptide (Figure 3G-H)). However, the FACS-based analysis of the protein level also demonstrated the consistency of the spectrum of activation strength we describe for this set of neoTCRs across titration of peptide (Figure 3A-H, S8) or target cells (Figure S7). Similar effects are expected for the transcriptome of patient T cells. While testing different peptide doses in the scRNAseq setting surely would have been interesting for further confirmation of this activation spectrum, limited patient material of this timepoint of blood withdrawal rendered these analyses beyond the scope of this project.

c. Results, line 162: the negative control are “freshly thawed CD8+ T cells”, wouldn't the best negative control be the same T cells, just unstimulated?

We agree with the reviewer that further controls for this sequencing experiment would have been interesting. However, we think that the chosen controls are reliable for our findings and statements. First, and most importantly, one central aspect was to capture the change within clonotype frequency upon in vitro stimulation, CD137⁺ enrichment and subsequent non-specific expansion to detect previously unknown neoTCRs (KIF-sc1 and -sc2, Figure 1B, C). This dynamic could only be addressed upon comparison to the unaltered, non-expanded TCR repertoire within peripheral blood of patient Mel15. Secondly, to avoid batch-effects within RNA analyses, all comparisons assessing the activation profile of neoTCRs were made only within neoTCRs from the restimulated sample and independently of the unstimulated, non-expanded, freshly thawed T cells, making a control sample unnecessary for this aspect. Thirdly, expansion of T cells by stimulation with an irrelevant/wildtype peptide (on day 0) from the batch thawed for the initial stimulation (day -1; Figure 1A) bears the risk of unintended selection of unspecific clonotypes. This includes EBV-specific TCRs due to in vitro stimulation with EBV-immortalized LCL target cells which could shift the TCR repertoire into a certain direction.

7. Introduction, line 68: reference 7 and 8 both refer to TIL studies. Leidner et al., PMID: 35648703 refers the TCR gene therapy targeting KRAS.

We thank the reviewer for this comment and adapted the manuscript accordingly.

8. Results, line 317: it would be helpful to put details related to the transduction efficiency of the different TIL used for infusion in the Results or figure legend (this is in the methods, but it would help the readers for it to be in Results or legend).

We added details on transduction efficiency to the figure legends.

Reviewer #2 (Remarks to the Author):

The manuscript by Fuchs et al describes a study where the investigators isolate neoantigen T cells from a disease-free melanoma patient undergoing checkpoint blockade using pembrolizumab. PBL-derived T cells were neoantigen stimulated and the cultures were enriched for CD137+ cells, then rapidly expanded prior to use. Neoantigen reactive T cell cultures were subjected to scRNAseq and scTCRseq to examine the relationship between the TCR expressed and biologic function. New neoantigen reactive TCR's were cloned and their function was in vitro and in vivo was evaluated against neoantigen expressing targets.

The authors conclude that T cells expressing different neoantigen TCR's have different transcriptional signatures and different antigen recognition potentials, even when targeting the same neoantigen peptide. Furthermore, T cells expressing neoantigen T cells with moderate signaling have superior control of tumor growth in their xenograft models. The authors argue that their analysis will help guide the field in improving adoptive T cell transfer for patients with advanced cancer.

While the authors data generally supports their conclusions, there are concerns with this study.

1) Isolating T cells from a patient undergoing checkpoint blockade may not be representative of the T cells in a typical cancer patient. Since the patient was disease free, it is also not clear that T cells are representative of a cancer patient.

We thank the reviewer for these thorough thoughts on the choice of patient for our study. We understand that observations made within a single patient with favorable disease course do not reflect all possible response patterns. However, we want to stress the value of such a detailed characterization of an oligoclonal tumor specific TCR repertoire directed towards tailored engineering strategies for immunotherapy. We are convinced investigations like ours are crucial to elucidate the interplay of factors behind TCR activation patterns.

Mel15 serves as a role model for the dissection of the most potent T-cell responses for patient-centered, individualized immunotherapeutic approaches. Thereby the herein reported findings cover a broad variety of aspects and though the analysis of further patients will complement the picture, this does not compromise our findings.

Moreover, the plethora of all identified neoTCRs – particularly targeting shared and divergent, mass spectrometry-validated neoantigens – prompted us to investigate in detail differences in functional behavior of single TCR clonotypes potentially also impacting their frequency. Since our group is interested to elucidate properties of potent neoTCRs for immunotherapy, we are convinced, that it is essential to analyze the TCR-repertoire of immunotherapy treatment responders which exhibit a TCR repertoire fit to target and eradicate the tumor e. g. after reinvigoration by CPI. In the light of another recent study in our own lab on a heavily pretreated patient cohort with overall low survival (ImmuNEO MASTER cohort), where TCR identification remained more difficult ¹⁸, a larger comparison on the TCR-repertoire of responders (particularly in full remission) as compared to non-responders might help to understand determinants responsible for a preferential outcome.

We therefore consider the choice of Mel15 and the comparison of his/her TCRs valuable to understand potent TCRs and TCR clonotypes. Overall, we do not intend to make a general conclusion about all cancer patients or TCR-repertoires using Mel15 alone. We are aware, that additional patients will be necessary to strengthen our hypotheses and assess further factors affecting TCR reactivities as well as engineering strategies for successful immunotherapy. To clarify the goal of our analysis we included a statement in the discussion of our manuscript (lines 622-637).

2) Since the authors used in vitro priming, CD137 selection, and a rapid expansion prior to scRNAseq, the repertoire is not likely to be reflective of the TCR repertoire and the T cell physiology is not likely to be reflective of the native T cells.

We agree that in vitro stimulation with the neoantigens KIF2C^{P13L} and SYTL4^{S363F} followed by CD137⁺ enrichment and rapid expansion alters the TCR repertoire of patient Mel15. However, using this stimulation approach it was not the intention to assess the native TCR repertoire but rather to understand if there are different TCR clonotype-dependent transcriptomic patterns after recent neoantigen-specific stimulation potentially explaining diverse native clonotype frequencies. In contrast, Mel15's native repertoire is more represented by the control sample of freshly thawed, non-expanded CD8⁺ T cells sequenced alongside with the in vitro treated/ stimulated sample (see Figure 1A). As elaborated earlier in reply to a question of reviewer #1, we prioritized this entirely unstimulated, non-expanded sample as the most important control within the limited patient material of Mel15 to reflect the more native TCR-repertoire of Mel15 at the time of blood withdrawal. Due to very low precursor frequencies of several neoTCRs, reasonable comparison of the transcriptomes of neoTCRs was only sufficiently possible for KIF-P1 and -P2 in comparison to all other non-assigned T cells (Figure S3D). The stimulation and enrichment process chosen in our setting, thus, was necessary to identify further, previously unknown neoTCR-clonotypes from Mel15 PBMCs based on a frequency change in comparison to the endogenous TCR-repertoire. The enrichment step relying on CD137 was based on thorough literature research ¹⁹.

3) Most of the analysis was using peptide loaded targets. Furthermore, the tumor cells used we transfected with a mini-gene. None of these are physiologically normal. There should be at least some experiments to show that the T cells, whether native or TCR Tg, recognize physiologically real levels of antigen. Otherwise, the message is overall relevance less convincing.

Please also consider our response to reviewer 1 (comment 1a). We are convinced that natural neoantigen expression in tumors can currently not be modelled well in artificial systems. The reason is, that neoantigen expression in tumors is highly heterogenous, depending not only on the allele frequency but many other factors influencing neoantigen processing and presentation. This heterogeneity is a consequence of continuous mutagenesis within the tumor which cannot mirrored in humanized models, so far. Therefore, we include a broad panel of antigen densities in the analysis of our TCR activation patterns by employing peptide pulsing with different amounts versus minigene-transduction (mg-tg) as well as different tumor entities with varying levels of HLA-surface expression (particularly low HLA-surface expression of lymphoma cell line U698M versus the melanoma cell line A2058 (Figure S10), see also comment 1a of reviewer #1).

Nevertheless, our pipeline for neoTCR identification includes mass spectrometry-based measurement of neoantigen surface presentation on the HLA-complexes of patient tumor material as published earlier for patient Mel15 ²⁰⁻²². Both neoepitopes, KIF2C^{P13L} and SYTL4^{S363F}, were detected via immunoprecipitation from HLA-class-I validating their presence on the tumor surface. Antigen-reactive clonotypes were detected in patient PBMCs as well as TILs indicating the presence of tumor-reactive T cells in response to the original levels of neoepitopes on tumor cells of Mel15 ²⁰. In addition, our MS-based approach of natural neoantigen identification has the advantage of validating the presentation of neoantigens in contrast to current clinically employed prediction-based pipelines ^{23,24}. Moreover, the high frequency of reactive T cell clonotypes in the patient's T cell repertoire ²¹ (Figure 1) further supports the physiological relevance of both neoepitopes in this individual patient.

However, the differences in TCR-tg T cell resilience were observed in an in vivo model with genetically engineered neoepitope expression as pointed out by the reviewer, which might have an impact on the stimulation level of our T cells ⁹. Still, antigen

presentation in our model is closer to the broadly applied overexpression of chicken OVA under a CMV-promotor in many murine models ²⁵. We added more emphasis on this fact in our manuscript (lines 583-586) to do justice to this valuable feedback.

4) With the TCR Tg T cells, the authors should consider how false pairing and culture conditions impact their results. Also, how is the level of TCR transgene expression impact the results?

All three aspects are important to consider for our model. We included data of an OTR setting, in which we inserted our TCR-chains into the TRAC locus employing CRISPR/Cas9 ^{14,15}. For KIF-P2 and KIF-sc1, we again detected comparable differences in surface expression in the RV (with endogenous TCR) and the OTR (disrupted TCR- α chain; Figure S4G). This finding rules out mispairing with the endogenous alpha chain as the sole cause of different surface expression for at least those two TCRs and support that comparisons made between these two TCRs may be independent on the mode of genetic transfer.

Concerning culture conditions, we are not entirely sure, which period of in vitro culture the reviewer is referring to. We are convinced, that any change of protocol either in TCR-transfer, in vitro T cell expansion or the TIL-P generation could impact on T cell fitness. Exemplarily, concerning TCR expression, we compared retroviral transduction to orthotopic TCR replacement (Figure S4F, G, S6) and found general accordance of both settings, yet, for any sort of clinical application, more detailed testing of different culture conditions – potentially depending on the construct and patient ²⁶ – would be essential. However, since the neoTCRs in any comparison undergo the exact same treatment, culture condition per se does not explain the differences we see.

Likely, the most important aspect is indeed TCR-surface expression. As outlined above in response to reviewer #1, the TCRs differ stably and construct-inherently in the level of surface expression, influencing T cell activation patterns (Figures S4A-E). KIF-P2 (and even more so KIF-P1) shows lower surface expression than KIF-sc1. We now added more emphasis on this aspect in the discussion (lines 540-546, 629-635).

Reviewer #3 (Remarks to the Author):

The authors in this manuscript investigated the different properties of neoantigen-specific TCRs and claimed that moderate stimulation of TCR is linked to resilience and sustained tumor control. In this manuscript however many controls and additional data are missing to conclude this.

Major:

This is a follow up to previous publication; maybe better to transform it to letter format? Figure 1, 2, 3: I miss what is the point of putting this data in this manuscript. Why 3 figures? It is difficult to extract the distinct messages of figures 1, 2 and 3.

We thank the reviewer for this feedback on the structure of our manuscript. Although we conducted a follow-up on a patient, who was previously included into our analyses, we are convinced that the current manuscript focuses on numerous new aspects and hypotheses that clearly exceed a letter format. We believe the complete set of significant data cannot be provided in a letter format and justifies a full manuscript.

First, we include scRNA- and TCR-seq as a new element into our neoTCR-identification pipeline – as a proof of concept on patient Mel15 in which we have identified several neoTCRs before^{20,21} – and moreover characterized transcriptomic differences between neoTCRs after in vitro restimulation. Figure 1 reflects on the pipeline and its use to identify new neoTCRs – which is still an important bottleneck of our own studies¹⁸ and also generally in the field²⁻⁶. To condense relevant information, we fused the original Figures 2 and 3 into now Figure 2 focusing on the overall transcriptomic signature in the dataset as well as the transcriptome of the neoTCRs. Consequently, we shifted parts of Figure 2 (now Figure S2A-F) to the Supplementary part of this manuscript.

The complexity of the second part of our manuscript further exceeds a follow up of the former works. Not only do we propose a new model for the evaluation of TCR-resilience and patterns of dysfunction by repeated stimulation compared to other recent works⁷, but moreover, we detected an unexpected advantage for TCRs with moderate stimulation patterns in this rechallenge model which may be associated with the higher frequency of these TCRs we observed in this patient. To our knowledge, this aspect has not been discussed for neoTCRs and T cell resilience is only insufficiently considered for TCR-tg T cell therapy.

Figure 4: Controls missing in figure 4a, 4b, 4c, 4h: no target and with target but no peptide.

For reasons of simplification, we initially did not show all controls performed in our assays. Yet, we thank the reviewer for this comment and are happy to include all controls we consider relevant. We added a negative control to each of these datasets plotted in former Figure 4a, b, c and h. Since it represents the most suited control for this setting from our point of view, we now depict IFN- γ (Figure 3A) or gMFI values (Figures 3B, C, D) in response to LCL targets presenting the wildtype KIF2C fragment without the mutation relevant for antigen immunogenicity. Frequencies are now depicted in Figure S5B-D. To complete the dataset, we also show gMFI of LAG-3 for the antigen titration. We also added all wildtype controls for the Annexin staining (Figure S9B-F). The cells of the wildtype condition were either pulsed with the highest concentration (100 μ M) of the KIF2C wildtype peptide or transgenic for the wildtype minigene. Overall, all these controls confirm the high specificity of our neoTCRs.

What about endogenous antigen presentation? This last point is highly relevant

We agree with the reviewer in this regard – as well as with reviewers #1 and #2. As explained above (question 1a, reviewer #1) in detail, we are aware of the artificial character of the antigen presentation in our model and further reflected on this point in our discussion to clarify for our readers (line 583-586).

As outlined above, we are convinced that even expression of the KIF2C^{P13L} mutation under its natural promotor does not entirely reflect the original heterogenous tumor due to the diversity of allele frequencies within the tumor. Yet, to cover diverse antigen densities, we performed analysis on peptide pulsed target cells confirming a stable reactivity pattern of different neoTCRs within a wide range of neoantigen density (Figure S5B-D). In addition, the analyzed cell lines harboring log-fold differences in their MHC expression (Figure S10) cover an additional variable of antigen presentation.

What is the expression of retrovirally introduced TCRs, are they all efficiently and comparably expressed?

We thank the reviewer for this very important comment, which reviewers #1 and #2 have also referred to (see also response to reviewer #1, question 2). TCR expression after retroviral transduction measured by TCRmu-staining rises concordant with increasing functional avidity in the retroviral system (Figure S4). These differences appeared as construct-intrinsic and remained visible at least for KIF-P2 and -sc1 even after orthotopic TCR replacement (OTR) as investigated now for the revision of the manuscript and integrated into our data set (Figure S4G). Hence, we regard these differences TCR-construct-specific and acknowledge that TCR-surface expression dynamics can significantly impact TCR-functionality and potentially resilience (lines 540-546, 629-635).

Please show LAG3 MFI in addition to percentage

To achieve more consistency in this regard, we replaced the frequencies for CD137+ and LAG-3+ over the course of 48h in the main manuscript with their geometric MFIs in all TCRmu+ cells (Figure 3E-H) and moved the frequencies to the supplementary parts of the manuscript (Figure S8). All gMFI values show the same spectrum of T cell activation as described before for the frequencies.

The conclusion of figure 4 is that sc1 is more activated and more inhibited; however, I do not see evidence for higher level of inhibition. (no functional inhibition shown, only more activation)

We chose the term inhibited relating to the higher expression of inhibitory receptors such as LAG-3 and PD-1 (in the RNAseq data as well as in the in vitro analysis). We understand the reviewer and to which extent this term might be misleading without functional data. We thus suggest referring to the detected activation patterns as “moderate” versus “stronger” without special emphasis of the inhibitory aspect.

Only one E:T ratio shown; choosing more E:T ratios would have provided more insight

We agree with the reviewer and thank for this valuable feedback. To complement the data shown, we performed further in vitro co culture stimulations with several E:T-ratios from 1:0.25 to 1:10 and included the results in Figure S7. With these analyses we demonstrate that higher numbers of target cells increase the production of the effector cytokine IFN- γ , as expected. However, a maximum of activation marker expression (e.g. CD137, LAG-3) is already reached at an E:T of 1:1 or 1:2 depending on marker and peptide concentration. Overall, all these analyses reflect the identical spectrum of neoTCR-activation as demonstrated in Figure 3 which is maintained throughout E:T-titration. Therefore, the E:T-ratio (1:1) chosen for several analyses in our manuscript reflects well on the range of maximal activation marker expression upon in vitro stimulation.

Figure 5: It is impossible to draw any conclusions without showing a detailed analysis of the TIL product before infusion.

What is the phenotype of TIL products? Major thing not considered: differentiation. What is the differentiation state, as in what is the frequency of memory/effector cells in the TILs before expansion as well as after ex vivo expansion?

We thank the reviewer for this question and added a more in-depth analysis of the TILs upon initial tumor encounter by splitting the original Figure 5 into two new figures: Figure 4 and Figure 5. In Figure 4 we focus on the equal in vivo tumor rejection upon first antigen encounter across different effector numbers and entities. In Supplementary Figure S11 and Rebuttal Figure R2 we depict several aspects of phenotyping the TILs in the tumor tissue upon sacrifice at day 5 after T cell injection, thus after first tumor encounter in vivo. In this respect we added information on tumor weight, T cell quantification, TCRmu+ frequency (Figure S11B-E and L-O), several activation markers (Figure S11F-I and P-S), phenotype (Figure S11J, K and T, U) and secretion of effector cytokines (Figure R2). These data demonstrate that the only difference between the TILs on day 5 could be detected in a higher TCRmu+ frequency for KIF-P2 than KIF-sc1 in the tumor (Figure S11D, N). This is particularly interesting, since a higher clonotype frequency of KIF-P2 (and -P1) was detected in the metastatic tissue and PBMCs of Me15. Our data might reflect improved proliferation or persistence tendencies.

As suggested by the reviewer we performed analyses of traditional T cell differentiation markers such as CD45RA or CD45RO on the day of sacrifice (Figure S11J, K and T, U) as well as throughout TIL-P expansion (Figure R3). Despite a lack of significant differences between different TCRs we saw a clear change in CD45RA expression comparing spleen and tumor tissue. While T cells on the day of sacrifice in the spleen (no cognate antigen present at the organ site) were predominantly CD45RA+CD45RO- (more naïve-like), more CD45RA-CD45RO+ (more memory-like) and RA-RO- T cells could be detected at the tumor site (Figure S11J, T). Regarding the differentiation state throughout in vitro expansion under IL-2 supplementation, the phenotype significantly changed to a mixture of CD45RA-CD45RO+ as well as CD45RA+CD45RO+ cells maintained over time (Figure R3)²⁷.

Overall, we could not detect phenotypical significant differences between different KIF2C^{P13L}-reactive TCRs in this regard.

To complete this dataset, we also performed further functional characterization of the TIL-P of donors A, B and C (three biological replicates of the initial in vivo restimulation experiment) after in vitro expansion on the day of reinjection. Thereby we revealed that the advantage in activation marker/cytokine expression discovered for TIL-P KIF-P2 in the first donor A (FACS data moved from original Figure 5 to Rebuttal Figure R4), was less pronounced or even absent in FACS data from the other donors B and C despite the same in vivo effect (Figure R4). Analysis for absolute secretion of a large panel of CD8/natural killer (NK)-effector cytokines of in vitro restimulated TIL-P on the day of reinjection in the lymphoma as well as in the melanoma model revealed a level of donor-dependent variation in classical T cell effector cytokines such as IFN- γ (confirming differences in FACS data (Figure R4)), IL-2, TNF or GzmB. Meanwhile we found increased IL-10 secretion for TIL-P KIF-P2 compared to KIF-sc1 – only slightly detected in the NEW conditions (Figure 5D, E, S15D, E). For donor B we could confirm these data on the RNA level in CD8-enriched samples narrowing the source down to the CD8+ T cells themselves (Figure 5F, S14D). Compared to our FACS analyses, the absolute quantification of cytokine secretion in the 13-plex panel allowed for insight into a broader panel of effector cytokines and a more cumulative assessment of T cell functionality within the first 20h after tumor encounter. Therefore, we put focus on these functional data for the TIL-P in our main manuscript (Figure 5D, E), but would of course be open to add the FACS data as presented to the reviewers in Figure R4.

Taken together, the donor-dependent heterogeneity for e. g. IFN- γ , IL-2 and TNF- α shows that classical CD8 T cell effector cytokines may not be causative for the detected in vivo phenotype. Since the anti-inflammatory cytokine IL-10 is known to possess a protective role for CD8+ T cells in infection models²⁸ and IL10-receptor (IL10R) signaling in CD8+ T cells is known to prevent exhaustion in a tumor context²⁹, we hypothesize a similarly protective role for this cytokine for TIL-P KIF-P2 compared to KIF-sc1. However, this hypothesis and the detailed pathway of IL-10 induction following moderate, but not strong T cell stimulation remains to be determined.

What is the percentage of mTCR+ cells after the expansion of the TILs?

To answer this question, we added the TCRmu+ frequency for each TIL-P from each individual mouse for donors A-B (Figure S13A-C). We furthermore show data on the absolute quantification of T cells in the TIL-P of KIF-P2 and -sc1 (Figure S13D-I). The TIL-P showed variations concerning the expansion of TCRmu+ T cells between animals while overall CD8+ cells mostly expanded at very similar rates for each mouse as well as comparing TCRs. Taken together, TCRmu+ TIL-P KIF-P2 expanded better than TCRmu+ TIL-P KIF-sc1. These data again indicate a proliferative benefit for TIL-P KIF-P2, which was also reflected in the more stable TCRmu-frequency of KIF-P2 in all three donors A, B and C (Figure S13A-C). However, since we injected the same total amount of TCRmu+ T cells into the second round of tumor-bearing hosts per TCR for the restimulation round, these quantitative differences throughout TIL-P expansion cannot account for the differences in tumor rejection afterwards.

What is the HLA typing of the donor T cells and the tumor cells.

Concerning HLA-Typing we can provide the following information (we underlined the HLA-types on which our neoantigens are presented: KIF2C^{P13L} on HLA-A03:01 and SYTL4^{S363F} on HLA-B27:05):

melanoma patient Mel15 (and thus Mel15 LCLs): A* 03:01:01, A* 68:01:01, B* 27:05:02, B* 35:03:01, C* 02:02:02, C* 04:01:01²⁰

tumor cell lines for in vivo experiments:

- U698M (B cell lymphoma): HLA-A*02:01; 03:01; HLA-B*27:05; 07:02
- A2058 (melanoma): HLA-A*03:01; 25:01, HLA-B*07:02; 18:01

[Redacted]

Why did the authors choose for this specific TIL expansion protocol (OKT3)? Would the authors expect different results when choosing for a CD3CD28 based expansion protocol?

Based on published rapid T cell expansion protocols³⁰ and thereof previously established protocols for TIL generation within our research group²⁰, we decided to continue with OKT3-based stimulation within our setting. It is an interesting question whether this protocol influences T cell functionality and therefore we directly compared our protocol with aCD3aCD28-bead based expansion. We used the same amount of IL-

2 (1000 U/ml) and irradiated feeder cells but exchanged OKT3 (30 ng/ml) with CD3aCD28 beads (thermofisher, 25µl per 2 ml, as recommended by the manufacturer), which we removed from culture after 6 days of expansion. Afterwards we only continued to supplement IL-2 in parallel to the OKT3-based protocol.

Concerning overall expansion, OKT3-cultured TIL-P showed slightly higher T cell expansion (CD8+ and TCRmu+) and higher TCRmu+ frequencies (Figure R5A-F) compared to the aCD3aCD28-protocol (Figure R5G-L). Notably, we could not detect significant differences in tumor cell killing between both protocols (Figure R5M), while CD3aCD28-expanded TCRmu+ TIL-P expressed lower frequencies of CD137 on their surface upon in vitro stimulation (Figure R5N).

Due to the overall slightly lower TCRmu+ frequency, we decided to continue with the OKT3-based protocol for our in vivo studies. Nevertheless, for any form of clinical application of T cell products, we are convinced that closer investigation of T cell expansion and culture conditions is highly relevant and might have to be adapted to individual TCRs/TCR activation patterns similarly to a recently published study even suggesting a patient-tailored T cell expansion for CAR-T cells ²⁶.

What would have been interesting: scRNA of TIL T cells...

We completely agree, that RNAseq of our TIL T cells on the day of sacrifice, but even more so upon rechallenge can provide extremely valuable insights into signaling differences. While scRNAseq from the rechallenge setting is beyond the scope of this manuscript, we performed bulk RNAseq from in vitro stimulated TIL-P and NEW cells derived from donor B to compare the transcriptome of KIF-P2 versus -sc1 and are happy to include parts of these new data in our manuscript (Figure 5F, S14D). Performing descriptive analyses on these data, we could confirm e. g. higher expression of IL10-transcripts for KIF-P2 compared to -sc1 (Figure 5F). It will be part of future analyses to further investigate immunosuppressive/inhibitory signatures and IL-10 signaling as potentially protective factor in T cell resilience.

In figure 4 the authors chose to show percentage rather than MFI; in figure 5 the authors showed MFI rather than percentage. For both the data of figure 4 and figure 5, please show either percentage or MFI in the main figure and show the other respective type in the supplement.

We replaced the frequencies for CD137+ and LAG-3+ in Figure 4 with their geometric MFIs of all TCRmu+ cells in the main figure and moved the frequencies to the supplementary parts of the manuscript (Figure S8). We hope to make our manuscript more consistent in this regard and thank the reviewer for this remark.

Positive control is missing for figure 5E 5F and 5G; (for example anti-CD3/CD28 beads, OKT3).

While, for reasons of simplification, we did not include the positive control here, we now added a PMA/Iono stimulated control to these data (Figure R4A-C). As described above in detail, we decided to depict data of a 13-plex cytokine secretion panel instead of these FACS analyses in our manuscript (Figure 5D, E).

Why do the authors chose for a B-cell lymphoma cell line rather than a melanoma cell line?

As explained in response to reviewer #1 (see comment to 1b), the U698M lymphoma cell line was selected to investigate patient-specific neoantigen-directed immune responses head-to-head and since the two Mel15-derived epitopes are restricted to diverse HLA, respectively HLA-A*03:01 and -B*27:05, U698M was the only available cell line inherently possessing both desired alleles. Consequently, our entity-agnostic approach mainly focused on efficient neoantigen-presentation leaving aside tumor-specific aspects. Since we are aware, that tumor-specific factors may influence our results, we

now also included data on reactivity against the neoantigen-expressing melanoma cell line A2058, which carries the A*03:01 allele and therefore serves as a suitable model for all KIF2C-specific TCRs. However, in parallel to the lymphoma model, the differences in tumor rejection and T cell cytokine pattern between KIF-P2 and -sc1 neoTCRs were similarly reflected upon restimulation in this melanoma model underscoring an entity-independent effect (Figure 5G, H, S15D, E).

If I understood correctly from materials and methods, the authors generated TIL products for all 5 mice per group and pooled TIL products from those two mice per group that showed highest mTCR frequency. Why no mTCR enrichment step? What were the frequencies of all TIL products generated from all mice? What were the expansion curves of TIL products? All these data would be important to make an informed judgement about what is happening between tumor explantation and TIL generation until reinfusion of the product.

We thank the reviewer for pointing to these additional aspects. Currently, enrichment steps for the transgenic population are rarely included into the clinical manufacturing of cell products³¹. In addition, TCRmu-based FACS-sorting may add not only bias due to potential differences of antibody binding, but also lead to cell stress (flow sort and possible cell activation due to TCR-binding). Therefore, we decided to forego an enrichment step already before the first T cell injection into mice.

Referring to the questions about frequencies in and expansion curves of TIL products, we included data on the TCRmu+ frequencies of all mice to Supplementary Figure 14A-C and marked those mice selected for the TIL-P per donor. In this way we excluded conditions in which CD8+ T cells did not or insufficiently expand. As explained above in response to the question about TCRmu+ frequencies, we already elaborated on data for the absolute quantification of T cells for all TIL-P conditions (Figure S13D-I).

The conclusions made in this manuscript are based on one single in vivo experiment with one T cell donor and two TCRs (figure 5). To make such a strong conclusions these experiments have to be repeated with additional T cell donors and with other TCRs also exhibiting differences in stimulation levels.

As proposed, we included two further biological replicates of the restimulation setting (in total three human donors A, B and C; Figure S12). Altogether, the three biological replicates of this initial setting all demonstrate the same trends in dynamics of rejection during the first two weeks after T cell injection: TIL-P KIF-P2 has an advantage in tumor rejection upon restimulation compared to KIF-sc1. Of all three donors, TIL-P KIF-P2 of donor A exhibited the strongest survival benefit. This alludes to certain biological variability and the numerous other factors influencing fitness of in vitro expanded T cell products which corresponds well with high inter-patient variability in real-world data from the clinic^{32,33}.

The reproducibility of the advantage of KIF-P2 over KIF-sc1 upon restimulation is further supported by a repetition of the setting with lower effector T cell numbers including all four KIF2C^{P13L}-reactive TCRs (Figure S16A, B). While no neoTCR-TIL-P is able to reach complete remission at this low effector cell number per mouse (5×10^5 TCRmu+) anymore, the before observed differences between all neoTCRs translate into tumor rejection patterns (Figure S16A). While KIF-sc1 and KIF-sc2 cannot significantly prolong survival compared to the control TCR 2.5D6, KIF-P2 and KIF-P1 are able to do so (Figure S16B). Further evidence for the reproducibility of our setting is given by transferring the initial setting into a melanoma cell line model (Figure 5G, H, S15) also confirming the same trend as mentioned above.

Minor:

Some sentences should be simplified to improve readability, for example line 100: Thereby, we are not only able to show that diverse activation patterns detected in scRNA- and

scTCR-seq of primary T cells are mirrored by in vitro and in vivo data of T cells transgenic for defined neoTCRs indicating significant stability in structural functionality.

We have adapted the manuscript accordingly.

Material and methods:

What do the authors mean by 2×10^7 transduced T cells (3.2×10^7 659 absolute T cells including non-transduced cells) were administered to 6 mice per group ($n = 6$) in two injections on two subsequent days.

A total of 2×10^7 neoTCR-tg T cells (3.2×10^7 absolute T cells including non-transduced cells) were administered to each individual of 6 mice per group ($n=6$). Injection was split to two subsequent days. We clarified this passage in the main text as well.

References

- 1 Lu, Y. C. *et al.* An Efficient Single-Cell RNA-Seq Approach to Identify Neoantigen-Specific T Cell Receptors. *Mol Ther* **26**, 379-389, doi:10.1016/j.ymthe.2017.10.018 (2018).
- 2 Oliveira, G. *et al.* Phenotype, specificity and avidity of antitumour CD8(+) T cells in melanoma. *Nature* **596**, 119-125, doi:10.1038/s41586-021-03704-y (2021).
- 3 Caushi, J. X. *et al.* Transcriptional programs of neoantigen-specific TIL in anti-PD-1-treated lung cancers. *Nature* **596**, 126-132, doi:10.1038/s41586-021-03752-4 (2021).
- 4 Lowery, F. J. *et al.* Molecular signatures of antitumor neoantigen-reactive T cells from metastatic human cancers. *Science*, eabl5447, doi:10.1126/science.abl5447 (2022).
- 5 Hanada, K. I. *et al.* A phenotypic signature that identifies neoantigen-reactive T cells in fresh human lung cancers. *Cancer Cell* **40**, 479-493.e476, doi:10.1016/j.ccell.2022.03.012 (2022).
- 6 Zheng, C. *et al.* Transcriptomic profiles of neoantigen-reactive T cells in human gastrointestinal cancers. *Cancer Cell* **40**, 410-423.e417, doi:10.1016/j.ccell.2022.03.005 (2022).
- 7 Schmidt, J. *et al.* Neoantigen-specific CD8 T cells with high structural avidity preferentially reside in and eliminate tumors. *Nat Commun* **14**, 3188, doi:10.1038/s41467-023-38946-z (2023).
- 8 Purcarea, A. *et al.* Signatures of recent activation identify a circulating T cell compartment containing tumor-specific antigen receptors with high avidity. *Sci Immunol* **7**, eabm2077, doi:10.1126/sciimmunol.abm2077 (2022).
- 9 Segal, G., Prato, S., Zehn, D., Mintern, J. D. & Villadangos, J. A. Target Density, Not Affinity or Avidity of Antigen Recognition, Determines Adoptive T Cell Therapy Outcomes in a Mouse Lymphoma Model. *J Immunol* **196**, 3935-3942, doi:10.4049/jimmunol.1502187 (2016).
- 10 Leignadier, J. & Labrecque, N. Epitope density influences CD8+ memory T cell differentiation. *PLoS One* **5**, e13740, doi:10.1371/journal.pone.0013740 (2010).
- 11 Anzalone, A. V. *et al.* Search-and-replace genome editing without double-strand breaks or donor DNA. *Nature* **576**, 149-157, doi:10.1038/s41586-019-1711-4 (2019).
- 12 Petri, K. *et al.* CRISPR prime editing with ribonucleoprotein complexes in zebrafish and primary human cells. *Nat Biotechnol* **40**, 189-193, doi:10.1038/s41587-021-00901-y (2022).
- 13 Hryhorowicz, M., Lipiński, D. & Zeyland, J. Evolution of CRISPR/Cas Systems for Precise Genome Editing. *Int J Mol Sci* **24**, doi:10.3390/ijms241814233 (2023).
- 14 Schober, K. *et al.* Orthotopic replacement of T-cell receptor α - and β -chains with preservation of near-physiological T-cell function. *Nat Biomed Eng* **3**, 974-984, doi:10.1038/s41551-019-0409-0 (2019).
- 15 Müller, T. R. *et al.* Targeted T cell receptor gene editing provides predictable T cell product function for immunotherapy. *Cell Rep Med* **2**, 100374, doi:10.1016/j.xcrm.2021.100374 (2021).
- 16 Singh, N. K. *et al.* Geometrical characterization of T cell receptor binding modes reveals class-specific binding to maximize access to antigen. *Proteins* **88**, 503-513, doi:10.1002/prot.25829 (2020).
- 17 Wu, D., Gallagher, D. T., Gowthaman, R., Pierce, B. G. & Mariuzza, R. A. Structural basis for oligoclonal T cell recognition of a shared p53 cancer neoantigen. *Nat Commun* **11**, 2908, doi:10.1038/s41467-020-16755-y (2020).
- 18 Tretter, C. *et al.* Proteogenomic analysis reveals RNA as a source for tumor-agnostic neoantigen identification. *Nature Communications* **14**, 4632, doi:10.1038/s41467-023-39570-7 (2023).
- 19 Wolf, M. *et al.* Activation-induced expression of CD137 permits detection, isolation, and expansion of the full repertoire of CD8+ T cells responding to antigen without requiring knowledge of epitope specificities. *Blood* **110**, 201-210, doi:10.1182/blood-2006-11-056168 (2007).

- 20 Bassani-Sternberg, M. *et al.* Direct identification of clinically relevant neoepitopes presented on native human melanoma tissue by mass spectrometry. *Nat Commun* **7**, 13404, doi:10.1038/ncomms13404 (2016).
- 21 Bräunlein, E. *et al.* Functional analysis of peripheral and intratumoral neoantigen-specific TCRs identified in a patient with melanoma. *Journal for ImmunoTherapy of Cancer* **9**, e002754, doi:10.1136/jitc-2021-002754 (2021).
- 22 Wilhelm, M. *et al.* Deep learning boosts sensitivity of mass spectrometry-based immunopeptidomics. *Nat Commun* **12**, 3346, doi:10.1038/s41467-021-23713-9 (2021).
- 23 Foy, S. P. *et al.* Non-viral precision T cell receptor replacement for personalized cell therapy. *Nature*, doi:10.1038/s41586-022-05531-1 (2022).
- 24 Rojas, L. A. *et al.* Personalized RNA neoantigen vaccines stimulate T cells in pancreatic cancer. *Nature* **618**, 144-150, doi:10.1038/s41586-023-06063-y (2023).
- 25 Garnier, L. *et al.* IFN- γ -dependent tumor-antigen cross-presentation by lymphatic endothelial cells promotes their killing by T cells and inhibits metastasis. *Sci Adv* **8**, eabl5162, doi:10.1126/sciadv.abl5162 (2022).
- 26 Zhang, D. K. Y. *et al.* Enhancing CAR-T cell functionality in a patient-specific manner. *Nat Commun* **14**, 506, doi:10.1038/s41467-023-36126-7 (2023).
- 27 LaSalle, J. M. & Hafler, D. A. The coexpression of CD45RA and CD45RO isoforms on T cells during the S/G2/M stages of cell cycle. *Cell Immunol* **138**, 197-206, doi:10.1016/0008-8749(91)90144-z (1991).
- 28 Trandem, K., Zhao, J., Fleming, E. & Perlman, S. Highly activated cytotoxic CD8 T cells express protective IL-10 at the peak of coronavirus-induced encephalitis. *J Immunol* **186**, 3642-3652, doi:10.4049/jimmunol.1003292 (2011).
- 29 Hanna, B. S. *et al.* Interleukin-10 receptor signaling promotes the maintenance of a PD-1(int) TCF-1(+) CD8(+) T cell population that sustains anti-tumor immunity. *Immunity* **54**, 2825-2841.e2810, doi:10.1016/j.immuni.2021.11.004 (2021).
- 30 Dudley, M. E., Wunderlich, J. R., Shelton, T. E., Even, J. & Rosenberg, S. A. Generation of tumor-infiltrating lymphocyte cultures for use in adoptive transfer therapy for melanoma patients. *J Immunother* **26**, 332-342, doi:10.1097/00002371-200307000-00005 (2003).
- 31 Hiltensperger, M. & Krackhardt, A. M. Current and future concepts for the generation and application of genetically engineered CAR-T and TCR-T cells. *Frontiers in Immunology* **14**, doi:10.3389/fimmu.2023.1121030 (2023).
- 32 Monfrini, C. *et al.* Phenotypic Composition of Commercial Anti-CD19 CAR T Cells Affects In Vivo Expansion and Disease Response in Patients with Large B-cell Lymphoma. *Clin Cancer Res* **28**, 3378-3386, doi:10.1158/1078-0432.Ccr-22-0164 (2022).
- 33 Kirouac, D. C. *et al.* Deconvolution of clinical variance in CAR-T cell pharmacology and response. *Nat Biotechnol*, doi:10.1038/s41587-023-01687-x (2023).

A

editorial note: figure redacted

B

C

D

E

F

Rebuttal Figure R1. SNV substitution for endogenous epitope presentation under the natural KIF2C promotor A, Schematic overview depicting SNV substitution using a CRISPR/Cas9-based strategy of prime editing including construction of a prime editing guide RNA (pegRNA) template containing the altered nucleotide sequence of KIF2C. Target cell lines were nucleofected with respective pegRNA and the ribonucleoprotein-complex (RNP). Efficacy of mutated gene expression was assessed by co-culture with KIF2C^{P13L}-specific T cells. pegRNA scheme adapted from Anzalone et. al. ¹¹; Figure created with BioRender.com. B, Electrophoretic validation of purification of the RNP containing Cas9 and reverse transcriptase. C, Expression of KIF2C-mRNA in several cell lines and selected healthy tissues. Relative gene expression was normalized to whole brain using the $\Delta\Delta\text{CT}$ -Method. D-E, Example of sanger DNA sequencing of the KIF2C gene locus in U698M (D) and A2058 (E) after nucleofection. F, Reactivity of KIF-P2- and 2.5D6-TCR-tg T cells against modified cells lines A2058 and U698M next to control target cells (U698M pulsed with mutated (KIF2C^{P13L}) or wild type (wt) peptide) measured by IFN- γ -secretion.

Rebuttal Figure R2. Cytokine secretion within tumor lysates after first in vivo tumor encounter reveals no functional differences between KIF-P2 and -sc1. Tumor lysate containing TILs after explant, enzymatic digestion and filtering was incubated for 20h either with addition of 50,000 further U698M mut mg-expressing tumor cells (+ mut mg) or without additional tumor cell (w/o). Multiplex analysis of human CD8-/NK-cytokines via legendplex is depicted for KIF-P2 versus -sc1 normalized to the input tumor weight. Statistical significance is calculated with unpaired t test: no statistically significant difference was detected. One dot represents one tumor-bearing mouse. Mean and SD are depicted.

■ KIF-P2
■ KIF-sc1

Rebuttal Figure R3. No significant differences in CD45RA/CD45RO expression between KIF-P2 and -sc1 TIL-P over the course of IL-2-driven expansion. FACS phenotyping of CD45RA (RA) and CD45RO (RO) was performed on days 6, 10, 14 and 19 of TIL-P expansion protocol gated on the CD8⁺/TCRmu⁺ (A, C, E, G) or only CD8⁺ (B, D, F, H) TIL-P cells of the individual mice sacrificed in the same experiment as shown in S11J. Mean and SD are shown of all individual TIL-P per mouse sacrificed on day 5.

Rebuttal Figure R4. Large donor variabilities for classical activation markers and cytokines upon in vitro restimulation of TIL-P. A-H, Ex vivo restimulation of T cells from TIL-P 21 days after tumor explant (TIL-P) compared to newly transduced (NEW) TCR-tg T cells from the same human donor (donor A: A-C, donor B: D-F, donor C: G, H) stained for CD137 (EC; A, D, G), IFN- γ (IC; B, E, H) and GzmB (IC; C, F); expression was analyzed using geometric mean of all CD3⁺CD8⁺/TCR μ ⁺ cells after 20h of co-culture. For IC cytokine analysis, secretion was blocked with Brefeldin A for the entire co-culture time. Mut mg and wt mg U698M cells used as target cells in E:T = 1:1 (50000 tg T cells : 50000 tumor cells). For donor A Phorbol-12-myristat-13-acetat (PMA)/Ionomycin (Iono) served as positive control. Mean and SD shown for three experimental replicates in A-C. For D-H triplicates were pooled prior to FACS analysis, only their mean is depicted.

Rebuttal Figure R5. Overall expansion is superior in the established OKT3 protocol compared to aCD3aCD28-bead-based TIL stimulation. A-L, FACS-based absolute quantification of CD8⁺/TCRmu⁺ and CD8⁺ cells in TIL-P conditions until day 19 as already depicted in S14 for the OKT3-based protocol (A-F) in comparison to the aCD3aCD28-based protocol (G-L). Mean and SD (A, D, G, J) of single growth curves (B, C, E, F and H, I, K, L) are depicted. M, N, TIL-P conditions from both protocols (OKT3 versus aCD3aCD28) were co-cultured with different E:T-ratios (1:1 or 1:3) with U698M-mut mg or wt-mg tumor cells and leftover dsRed⁺ tumor (M) as well as frequency of CD137⁺ of TCRmu⁺ T cells (N) shown. No significant differences were detected between similarly stimulated OKT3 and aCD3aCD28 conditions by one-way ANOVA and Tukey's multiple comparison test.

REVIEWER COMMENTS

Reviewer #1 (Remarks to the Author):

I thank the authors for their effort to address my concerns. The data and text modifications and additions have strengthened the manuscript. However, there are still some technical concerns and the conclusions of the manuscript remain limited. The reality is that the question that the authors are tackling is important but dependent on each patient/sample since the neoantigens and TCR, neoantigen density, and tumor microenvironment, etc., are patient-specific. As such, the correlation between the strength of TCR stimulation and antitumor ability of T cells/TCRs observed in this study can only be made for this patient. Moreover, the correlation between the transcriptome/function of in vitro stimulated T cells expressing native neoantigen-reactive TCRs to the antitumor ability of virally transduced TCR engineered T cells is based on the assumption that signaling by an endogenous T cell expressing the native neoantigen-reactive TCR is the same as a T cell virally engineered to express the same receptor. The authors have not demonstrated this (described more below). This is not meant to say the studies are not interesting; as more and more similar studies are performed, perhaps general trends will be observed. However, the paper currently reads as if “moderate TCR signal strength is better” is a rule rather than just based on a few TCRs targeting one neoepitope in one patient; perhaps more qualifying words could be used. There are also some technical challenges as indicated below that limit what can be concluded in these studies.

There are a few key issues that would benefit from additional information. The authors justify the use of the U698M leukemia model based on HLA of the tumor cell line (since it matches with the HLA restriction of two of the neoepitope reactivities). It is nice to see that the authors added an additional (melanoma) cell line to their in vivo studies to validate one of the reactivities. However, the neoantigen is also ectopically expressed in this model, which is quite artificial as all reviewers previously noted. Testing a model with lower antigen density be insightful and I realize that the authors attempted to use Crispr/Cas9 to generate this model but with no success. Thus, it would be very helpful if immunopeptidomics could be performed on both the U698M and the A2058 melanoma cell lines (like previously done for Mel15 autologous tumor) to determine levels of targeted neopeptide/HLA in these

minigene-expressing cell lines compared to the Mel15 tumor. Is there any way to correlate levels of neopeptide/HLA on minigene-expressing cell lines compared to the level of peptide pulsed cells? For example, running immunopeptidomics on cells pulsed with titrated doses of peptide? This would at least allow the reader to understand relative levels of antigen expression and put the results in better context. Also, in the new melanoma cell line, why were only 2 TCRs tested and not the 4 as done with the lymphoma cell line?

The second issue that would benefit from additional information is how TCR expression impacts the results, which was brought up by previously by all reviewers. The authors attempted to address this by 1) showing TCR expression levels of the virally engineered T cells; and 2) performing Crispr/Cas9 knock in of the TCRs into the TRAC locus (OTR). In doing so, this has brought up additional questions. First, there was a wide range of expression levels of the TCRs upon viral engineering, with lower expressing TCRs seemingly performing better. Are these also reflective of expression levels of the TCRs in the endogenous non-engineered T cell populations? While expression of two of the TCRs appeared similar when delivered by both viral and OTR, KIF-P1 and KIF-sc2 show some differences. The authors show some comparisons of viral vs. OTR for three of the TCRs, and notably, it seems like KIF-sc2 might show some functional differences (e.g., Fig. S6, CD137 upregulation), suggesting that activation strength might be different between viral and OTR. There is no functional data comparing viral and OTR for the KIF-P1 TCR engineered T cells. How does KIF-P1 viral vs OTR TCR engineered T cells compare functionally and how does this correlate to the scRNA-seq/activation data of the endogenous peptide stimulated T cells? This is a relevant question because the authors are making correlations between signaling strength of endogenous T cells expressing native neoantigen-reactive TCRs (using their peptide stimulation assay) and linking these to potential activity of a TCR engineered T-cell product. Another complicating factor is that TCR regulation of viral vs OTR (and endogenous) can be different, in that TCRs might be more naturally downregulated upon stimulation in the OTR setting (thereby potentially decreasing/modulating signal strength). Thus, did the authors test how viral vs OTR engineered TCR T cells behave after repeated stimulation? Even more informative would be how the viral vs OTR TCR engineered T cells perform in vivo. There is precedent for better in vivo function of CAR-T (Eyquem, PMID: 28225754) and TCR-T (Roth, PMID: 29995861) that underwent insertion at the TRAC locus compared to retroviral-

mediated insertion. Another related point: if there can be differences in TCR expression for viral vs OTR, then this makes one wonder if expression of the neoantigen-reactive TCRs in endogenous populations is different than gene engineered (viral or OTR) T cells. That is, do the authors have endogenous T-cell clones from the patient that expresses these neoantigen-reactive TCRs, and do they show the same relative TCR expression level as the gene edited T cells? For example, for KIF-P1, does the expression of the endogenous TCR in non-engineered T cells look more similar to viral or OTR engineered T cells? In summary, there are three TCR expression scenarios to consider: viral vs. OTR vs. endogenous. There doesn't appear to be sufficient experimentation to understand whether function of each TCR varies or is the same depending on how the TCR is expressed (endogenous, viral, OTR). If the authors are trying to link signaling strength level (e.g., their scRNA-seq and functional data of peptide stimulated non-engineered T cells) to antitumor function of engineered T cells, then knowing this information would be important.

Reviewer #3 (Remarks to the Author):

manuscript is improved

Reviewer #4 (Remarks to the Author):

While the authors have addressed most of the issues raised by the reviewers, there are still some points that should be clarified.

Overall, it is difficult to follow several of the points being made in the manuscript, in part due to awkward phrasing but also due to unclear explanations of the data presented in the manuscript.

1. For example, Fig. S2 is labeled as 'Expression profiles of known and unknown TCR clonotypes were compared as assessed by scTCR-/scRNA-sequencing'. In this figure it is not clear which of the cells have known reactivity and which are unknown.
2. The cluster analysis shown in Fig. 2D does not appear to be reflected in Fig. 2C in that KIF-P1 appears to contain a high number of cells in clusters 7,8 and 9 in comparison to those in cluster 5. In addition, it is very difficult to draw conclusions regarding the distribution of cells in clones other than KIF-P1 and P2 given the relatively small numbers of total T cells

corresponding to these clonotypes.

3. The expression levels of the transduced TCRs, in particularly the low levels of expression of the KIF-P1 and KIF-P2 TCRs, are very concerning and the graphs shown in Fig. S4A do not appear to reflect the frequencies of TD cells shown in Fig. S4B. The frequency of KIF-P1+ T cells shown in S4A appears to be only a few percent and in addition the expression levels are very low relative to the KIF-sc1 and SYT-T1 T cells, leading to the question of whether there is an issue with the sequence or transduction methods. In addition, the % of TD cells seen when OTR was used to increase expression of the KIF-P1 TCR was not given, although the gMFI of KIF-P1-transduced T cells appeared to be comparable to those of KIF-sc1 and SYT-T1 T cells. Given the huge difference between the expression of KIF-P1 and SYT-T1 it is not clear that OTR was able to correct the deficiency seen for the KIF-P1 TCR and so this raises the issue of whether or not these results are at least in part due to an artifact resulting from an issue with the KIF-P1 and KIR-P2 TCRs, as natural TCRs should not differ this much in terms of their expression levels. It would be helpful to show the FACS plot of the OTR modified T cells.

4. The fact that T cells were analyzed after a 24-hour co-culture with peptide pulsed targets in many of the assays can potentially lead to artifacts due to the AICD induced by the stimulus. Measurement of apoptotic cells using Annexin V and PI does not necessarily capture this phenomenon as many T cells can die and will not even be seen in these cultures as they may appear in population of small cells that are normally gated out for this analysis. One way of determining the potential effects of cell death would be to evaluate total T cell numbers after the co-culture, which should be reported for these experiments.

5. The differences between the anti-tumor effects of TIL-P KIF-P2 and TIL-P KIF-sc1 T cells in the tumor re-challenge model was clear in mice receiving the U698M lymphoma shown in Fig. 5B and C; however, the differences seen in mice receiving the A2058 solid tumor shown in Fig. 5G and H and in the repeat experiment with the U698 tumor shown in Fig. S12 were relatively modest, with both populations of TCR TD cells only slowing tumor cell growth but not leading to tumor rejection. In addition, modest differences between the abilities of TIL-P KIF-P1,P2 sc1 and sc2 to slow tumor growth in the experiment shown in Fig. S16. Given the fact that dramatic differences between the 2 TCRs was only seen in 1 of 4 experiments, the conclusion that the ability of T cells transduced with these TCRs to mediate tumor rejection

is different is does not appear to be strongly supported by the data.

6. All human T cells express CD45 and so it is not clear why substantial numbers of T cells in Fig. S11 were noted as RA-RO- as they should express 1 or both products.

7. On page 20, you stated that 'Because of the inferior surface expression capacity of KIF-P1 in the RV system (3% compared to 25% upon initial injection into the mice sacrificed for TIL-P), the potent in vivo capacity of TIL-P KIF-P1 upon rechallenge (1% versus 2-8% TCRmu+; 5×10^5 TCRmu+ per mouse) was particularly surprising.' It is not clear what this refers to and was difficult to determine where this data was presented in the manuscript.

8. These are not complete sentences and should be combined in some way to conform to English usage. 'Based on inherent immunogenicity of malignant cells, these therapies make use of the immune system's ability to recognize and eradicate tumor cells. Though the exact interplay between immune recognition and tumor eradication versus escape during immunotherapy remains ill-defined to date. However, its understanding is key to develop full immunotherapeutic potential 2.'

9. On page 23, you stated that 'These findings strengthen TCR-inherence and transferability of the described heterogenous patterns from the patient and indicate a dominant, structural or mechanic component of TCR-peptide-MHC complex assembly rather than patient-imprinted differentiation.'

10. The statement on page 4 'Since one major challenge lies in attacking mutant cells with as little off-target toxicity as possible 5,6, ...' is not correct since the toxicities were due to reactivity with non-mutant targets, and there is no evidence that targeting neoepitopes leads to off-target toxicity.

11. The statement on page 17 'In parallel, we compared (with what?) with a new transduction with the same two TCRs on CD8+ T cells from the same donor (NEW) as control groups (Figure 5A D).' is unclear as noted in parentheses.

Point-to-point reply to reviewer comments (NCOMMS-23-01734A)

We want to thank the reviewers for their feedback on our revised manuscript. To improve the message of our data and answer the raised questions, we performed several additional experiments. We are convinced our latest data on antigen density in our system as well as the comparison of engineering systems of TCR-T cells strengthen our conclusions.

Reviewer #1 (Remarks to the Author):

I thank the authors for their effort to address my concerns. The data and text modifications and additions have strengthened the manuscript. However, there are still some technical concerns and the conclusions of the manuscript remain limited. The reality is that the question that the authors are tackling is important but dependent on each patient/sample since the neoantigens and TCR, neoantigen density, and tumor microenvironment, etc., are patient-specific. As such, the correlation between the strength of TCR stimulation and antitumor ability of T cells/TCRs observed in this study can only be made for this patient. Moreover, the correlation between the transcriptome/function of in vitro stimulated T cells expressing native neoantigen-reactive TCRs to the antitumor ability of virally transduced TCR engineered T cells is based on the assumption that signaling by an endogenous T cell expressing the native neoantigen-reactive TCR is the same as a T cell virally engineered to express the same receptor. The authors have not demonstrated this (described more below). This is not meant to say the studies are not interesting; as more and more similar studies are performed, perhaps general trends will be observed. However, the paper currently reads as if “moderate TCR signal strength is better” is a rule rather than just based on a few TCRs targeting one neoepitope in one patient; perhaps more qualifying words could be used. There are also some technical challenges as indicated below that limit what can be concluded in these studies.

We thank reviewer 1 for the critical review of our revised manuscript and are pleased to hear that the first round of revisions has been acknowledged as improvement of the manuscript. As stated in our manuscript, we agree with Reviewer 1’s view that beyond this work further in-depth studies of single-patient TCR-repertoires will be necessary for a broad picture of TCR-characteristics across patients and entities. However, to our knowledge, so far, no other report provides such depth of analyses for one patient’s TCR-repertoire. The strength of our study is its focus on one single neoTCR-HLA-ligand interaction completed by non-patient-related in vitro and in vivo models. Eventually, our findings in this novel study contradict the field’s current mainstream opinion on which receptor to choose for ACT (highest surface expression, highest functional avidity¹⁻³. Therefore, we are convinced that our study provides a highly relevant contribution for future engineering decisions in the TCR-T cell field and indicates that a spectrum of characteristics for TCR qualities needs to be considered and defined to exploit the full power of ACT.

Nevertheless, we agree that our conclusions focus on retrovirally engineered TCR-T cells and by changing title as well as wording of our manuscript in several places (e.g. lines 54, 114 ff, 251, 282ff, 330, 467, 546-548, 562) we aim to limit our conclusions to our current data and adequately address the justified concern of the reviewer about statements which may be too broad with respect to the data shown here. We therefore specified our conclusion and chose more qualifying words. Since particularly the second large question raised by reviewer 1 picks up on this aspect, we will discuss the comparability of engineered T cells and endogenous T cells in more detail below.

There are a few key issues that would benefit from additional information. The authors justify the use of the U698M leukemia model based on HLA of the tumor cell line (since it matches

with the HLA restriction of two of the neoepitope reactivities). It is nice to see that the authors added an additional (melanoma) cell line to their in vivo studies to validate one of the reactivities. However, the neoantigen is also ectopically expressed in this model, which is quite artificial as all reviewers previously noted. Testing a model with lower antigen density be insightful and I realize that the authors attempted to use Crispr/Cas9 to generate this model but with no success. Thus, it would be very helpful if immunopeptidomics could be performed on both the U698M and the A2058 melanoma cell lines (like previously done for Mel15 autologous tumor) to determine levels of targeted neopeptide/HLA in these minigene-expressing cell lines compared to the Mel15 tumor. Is there any way to correlate levels of neopeptide/HLA on minigene-expressing cell lines compared to the level of peptide pulsed cells? For example, running immunopeptidomics on cells pulsed with titrated doses of peptide? This would at least allow the reader to understand relative levels of antigen expression and put the results in better context.

We thank the reviewer for this important comment on the relevance of antigen density and the suggestion on using mass spectrometry for measuring neoepitope expression on our genetically engineered tumor cell lines. As proposed, using mass spectrometry we measured the abundance of KIFP2^{P13L} on the surface of mut mg-tg tumor cell lines U698M, A2058 and Mel15 LCLs in comparison to peptide-pulsed wildtype variants of each cell line to get a rough estimation of the antigen density on the tumor cell surface (Figure S10B). Due to overall equal input of tumor cells within each sample (150x10⁶ each), the abundance of neoepitope measured by MS can be compared between all three cell lines and illustrates overall much lower epitope expression on U698M compared to A2058. Thus, the two models chosen represent a model of lower antigen density on the one hand (lymphoma) and higher antigen density (melanoma) on the other hand, knowing that the tumor entity itself might influence antigen presentation per se. It was interesting to see that the antigen level of mg-engineered tumor cell lines ranked around the values of 0.1 to 1 μ M-peptide-pulsed tumor cell lines, particularly since the overall activation level of KIF-sc1-tg T cells to the mut mg-variant exceeded the stimulation by peptide-pulsed tumor cells (Figure S10C). This indicated that the mode of antigen presentation, likely related to its stability, influences T cell activation beyond sole quantity of peptide after pulsing or gene transfer.

T cell reactivity will differ in complex primary tumors due to the biology of intratumoral heterogeneity and might not be stable over the whole course of treatment. To that end, models like ours representing only one of many cell-line-based tumor models in the field cannot imitate a natural tumor environment and dynamic disease course. Lacking further tumor material of patient Mel15, it was not possible to measure the patient's tumor alongside these cell lines.

Overall, these important experiments suggested by reviewer 1 show that our findings do not only depend on epitope density, but that the tumor cell itself has a decisive impact on T cell reactivity. Yet, elucidating all aspects of epitope density and subsequent T cell reactivity exceeds the scope of this manuscript centered on the TCR-functionality.

Also, in the new melanoma cell line, why were only 2 TCRs tested and not the 4 as done with the lymphoma cell line?

Since the major differences we aimed to investigate in a different cell line were detected between KIF-P2 and KIF-sc1 we primarily focused on these two representative neoTCRs within the scope of limited time and material. In our new investigations comparing OTR and RV setting we, as suggested by the reviewer, again performed in vitro experiments including all four KIF2C-reactive neoTCRs employing the lymphoma as well as the

melanoma cell line (Figure S16A-D, F-H). Here, the same differences in activation pattern, advantage in in vitro killing and T cell resilience were observed for both cell lines again supporting the entity-independent nature of the phenotype described.

The second issue that would benefit from additional information is how TCR expression impacts the results, which was brought up by previously by all reviewers. The authors attempted to address this by 1) showing TCR expression levels of the virally engineered T cells; and 2) performing Crispr/Cas9 knock in of the TCRs into the TRAC locus (OTR). In doing so, this has brought up additional questions. First, there was a wide range of expression levels of the TCRs upon viral engineering, with lower expressing TCRs seemingly performing better. Are these also reflective of expression levels of the TCRs in the endogenous non-engineered T cell populations?

This question raised by the reviewer in this and particularly the last paragraph of the response (“Another related point: if there can be differences in TCR expression [...]”) is highly interesting, yet hard to answer. We cannot provide expression data of TCR clonotypes of Mel15 to investigate surface expression in the patient. Despite high overall frequency of some of the neoTCRs our study is focusing on, the frequencies of KIF-P1, KIF-P2 and SYT-T1 exemplarily range between 0.5%, 0.04% and 0.002% of total PBMCs on day 945 of Mel15’s treatment ⁴, thus can hardly be quantified sufficiently by flow cytometry and would require much more sophisticated methods beyond the scope of this manuscript.

In fact, limited expansion capacity of such small TCR-clonotype populations and their loss in our – published ⁵ and unpublished (ongoing work in multiple myeloma) – and other pipelines ⁶, was the reason to introduce the scTCRseq step (Figure 1) into our workflow in the first place. Only this step enabled the identification of KIF-sc1 and -sc2.

Considering that we currently cannot show these data, we hope to satisfy the reviewer’s criticism by adapting the wording and title of our manuscript to avoid too general statements beyond engineered T cells as mentioned earlier.

Nevertheless, we want to highlight again that both, KIF-P1 and KIF-P2, by far, exceeded the other TCRs in their clonotype precursor frequency in patient Mel15’s TCR repertoire. The reconstruction, cloning and expression of those two TCRs was performed using the same pipeline that was applied to all other TCRs with similar length of DNA, same backbone, same promotor – overall without any differences in the process known to us. Yet, they still differ in surface expression, even in a second expression system (OTR; described below in more detail; Figures S4, S16) and showed increased resilience which coincides with a higher frequency in the patient.

While expression of two of the TCRs appeared similar when delivered by both viral and OTR, KIF-P1 and KIF-sc2 show some differences. The authors show some comparisons of viral vs. OTR for three of the TCRs, and notably, it seems like KIF-sc2 might show some functional differences (e.g., Fig. S6, CD137 upregulation), suggesting that activation strength might be different between viral and OTR. There is no functional data comparing viral and OTR for the KIF-P1 TCR engineered T cells. How does KIF-P1 viral vs OTR TCR engineered T cells compare functionally and how does this correlate to the scRNA-seq/activation data of the endogenous peptide stimulated T cells? This is a relevant question because the authors are making correlations between signaling strength of endogenous T cells expressing native neoantigen-reactive TCRs (using their peptide stimulation assay) and linking these to potential activity of a TCR engineered T-cell product.

To further investigate the impact of TCR expression under the natural promotor in our system, we performed additional experiments comparing OTR versus RV system in

more detail. We dedicated a detailed Supplementary Figure S16 (replacing old Figure S6) at the end of our manuscript to this topic to underline the importance of this comparison between different engineering systems as evident by the reviewer's question.

We therefore analyzed three additional biological replicates of OTR- versus RV-TCR-engineered T cells (Figure S4G-K and S16) and can confirm that KIF-P1 and KIF-sc2 both seem to profit in level of surface expression from the OTR expression system (Figure S4K). Both TCRs are expressed at a higher level under the TRAC promotor compared to the CMV promotor, however, this does not translate into a functional advantage for any of the two TCRs in the OTR-engineered T cells (Figure S16F-H). We want to highlight that substantial differences in T cell expansion protocol were necessary to make such a comparison possible, since CRISPR/Cas9 knock-in-frequencies ranged between 2.7 and 7.99 % in all CD8⁺ (0.66 to 2.2 % in all cells since the protocol is performed on PBMCs^{7,8}); this was in the range of a knock-in efficiency to be expected according to our collaborators. Since our retroviral system resulted in much higher frequencies, we were able to expand TCR-tg T cells as gently as possible with IL-7 and IL-15 only^{4,9} after transduction and initial activation with aCD3/aCD28 and IL-2 for our prior experiments. The low CRISPR-knock-in efficiency now required TCRmu-based FACS-sorting followed by intensive in vitro stimulation of TCR-engineered cells by repeated use of phytohemagglutinin (PHA) and IL-2 (Figure S4G). Since any single step in this new protocol can cause large differences in T cell phenotype and thus functionality, we also illustrated these protocol differences further (Figure S4G) in our manuscript to clarify for our readers. To compare OTR and RV system, the virally transduced cells underwent the same protocol for all comparisons in Figure S16.

Within this setting we functionally tested RV versus OTR TCR-engineered T cells and included data on the surface expression of activation upon stimulation of OTR- versus RV-KIF2C-reactive TCRs with either U698M or A2058 (Figure S16A-D). Alongside we present data on killing of U698M and residual TCRmu⁺ counts after 20h and 72h (Figure S16F-H). We also added in vivo data of the first tumor encounter (Figure S16I-M). While all these data confirmed a more moderate activation level of OTR-KIF-P2, they also demonstrated its strong killing capacity linked to increased numbers of residual TCRmu⁺ T cells upon first tumor contact in vitro and even in vivo despite lower TCR surface expression. Interestingly, neoTCR KIF-P1 again showed a functional advantage after first tumor encounter in the RV-system, partly also visible upon higher TCRmu⁺ counts after 72h of co culture in the OTR system (Figure S16H). We previously discussed the special features of KIF-P1 already (Figure S15).

Concerning the reviewer's question about KIF-sc2, there is no functional difference or advantage to the RV-system detected in our analyses (Figures S16F-H) despite higher expression of this TCR under the TRAC-promotor (Figure S4K),

Another complicating factor is that TCR regulation of viral vs OTR (and endogenous) can be different, in that TCRs might be more naturally downregulated upon stimulation in the OTR setting (thereby potentially decreasing/modulating signal strength). Thus, did the authors test how viral vs OTR engineered TCR T cells behave after repeated stimulation? Even more informative would be how the viral vs OTR TCR engineered T cells perform in vivo. There is precedent for better in vivo function of CAR-T (Eyquem, PMID: 28225754) and TCR-T (Roth, PMID: 29995861) that underwent insertion at the TRAC locus compared to retroviral-mediated insertion.

Beyond the initially performed *in vitro* analysis of OTR engineered T cells, we now investigated the efficacy of the two model TCRs KIF-P2 and -sc1 within the *in vivo* rechallenge protocol as well. Of the heavily pretreated RV-cells from this experiment (Figure S4G, S16I-M) we were not able to generate sufficient amounts of TIL-P RV-KIF-P2 and -sc1 for a second *in vivo* application. As for the OTR conditions, we were able to sufficiently expand TIL-P OTR-KIF-P2 and -sc1 for *in vitro* assessment of repeated tumor recognition and reinjection into a new round of tumor-bearing hosts (Figure S16N-W). Both, the *in vitro* restimulation (Figure S16N-U) and in particular the *in vivo* model (Figure S16V, W) substantiated the significantly improved tumor control of TIL-P OTR-KIF-P2 upon rechallenge. Administration of TIL-P OTR-KIF-P2 demonstrated significantly improved survival of tumor-bearing mice (Figure S16W). Thus, independently of the promoter and engineering system (viral versus CRISPR/Cas9), the more moderate TCR KIF-P2 exhibited increased resilience upon restimulation.

So far it remains unclear why RV-engineered T cells did not further expand in comparison to the OTR-engineered T cells in this experiment though both had undergone equally heavy pretreatment (Figure S4G). However, this technical difference is not the focus of this manuscript and remains to be elucidated in future projects

Another related point: if there can be differences in TCR expression for viral vs OTR, then this makes one wonder if expression of the neoantigen-reactive TCRs in endogenous populations is different than gene engineered (viral or OTR) T cells. That is, do the authors have endogenous T-cell clones from the patient that expresses these neoantigen-reactive TCRs, and do they show the same relative TCR expression level as the gene edited T cells? For example, for KIF-P1, does the expression of the endogenous TCR in non-engineered T cells look more similar to viral or OTR engineered T cells? In summary, there are three TCR expression scenarios to consider: viral vs. OTR vs. endogenous. There doesn't appear to be sufficient experimentation to understand whether function of each TCR varies or is the same depending on how the TCR is expressed (endogenous, viral, OTR). If the authors are trying to link signaling strength level (e.g., their scRNA-seq and functional data of peptide stimulated non-engineered T cells) to antitumor function of engineered T cells, then knowing this information would be important.

We kindly refer to our answer to this question in detail above. We agree that these three modes of TCR expression must be differentiated. Given that we cannot show data of surface expression of TCRs on endogenous TCR clones lacking material and also feasibility to expand these clones from overall very low precursor frequencies – particularly for KIF-sc1 and -sc2 – we adapted our manuscript accordingly and stated more clearly that our conclusions concern TCR-engineered T cells (e.g. change in title, lines 54, 114 ff, 330, 467, 546-548, 562).

Regarding the comparison of viral versus OTR we explained in detail how both protocols are distinct from each other and which functional comparisons we made. Overall, regarding the differences in protocol but moreover the mechanistic differences between OTR and RV engineering system, our data confirms the conception that both expression systems may reveal different facets of TCR functionality¹⁰. We think it will be crucial to compare different promoters and insertion methods as well as expansion protocols per construct in the future and right now we cannot explain all differences between both systems.

Yet, despite all differences we are eventually even more convinced of the increased resilience of the more moderate TCR KIF-P2 since we detected similarly increased tumor control upon rechallenge in the OTR setting as previously in our *in vivo* experiments from virally engineered T cells.

Reviewer #3 (Remarks to the Author):

manuscript is improved

We thank reviewer 3 for assessing the revised version of our manuscript and are happy that the data and explanations we added sufficiently answered his/ her/ their questions.

Reviewer #4 (Remarks to the Author):

While the authors have addressed most of the issues raised by the reviewers, there are still some points that should be clarified.

We thank reviewer 4 for his/her/their assessment of the improvement of our manuscript during the first round of revision.

Overall, it is difficult to follow several of the points being made in the manuscript, in part due to awkward phrasing but also due to unclear explanations of the data presented in the manuscript.

1. For example, Fig. S2 is labeled as 'Expression profiles of known and unknown TCR clonotypes were compared as assessed by scTCR-/scRNA-sequencing'. In this figure it is not clear which of the cells have known reactivity and which are unknown.

We thank the reviewer for pointing to a misleading figure name and changed it to "Expression profiles of TCR clonotypes were compared as assessed by scTCR-/scRNA-sequencing."

2. The cluster analysis shown in Fig. 2D does not appear to be reflected in Fig. 2C in that KIF-P1 appears to contain a high number of cells in clusters 7,8 and 9 in comparison to those in cluster 5. In addition, it is very difficult to draw conclusions regarding the distribution of cells in clones other than KIF-P1 and P2 given the relatively small numbers of total T cells corresponding to these clonotypes.

We assume that depicting "unstimulated" versus "stimulated" cells in the same graph might cause confusion and therefore color-coded the subtitles matching Figure 2A and C for clarification which clonotypes correspond to which cluster analysis.

3. The expression levels of the transduced TCRs, in particularly the low levels of expression of the KIF-P1 and KIF-P2 TCRs, are very concerning and the graphs shown in Fig. S4A do not appear to reflect the frequencies of TD cells shown in Fig. S4B. The frequency of KIF-P1⁺ T cells shown in S4A appears to be only a few percent and in addition the expression levels are very low relative to the KIF-sc1 and SYT-T1 T cells, leading to the question of whether there is an issue with the sequence or transduction methods. In addition, the % of TD cells seen when OTR was used to increase expression of the KIF-P1 TCR was not given, although the gMFI of KIF-P1-transduced T cells appeared to be comparable to those of KIF-sc1 and SYT-T1 T cells. Given the huge difference between the expression of KIF-P1 and SYT-T1 it is not clear that OTR was able to correct the deficiency seen for the KIF-P1 TCR and so this raises the issue of whether or not these results are at least in part due to an artifact resulting from an issue with the KIF-P1 and KIF-P2 TCRs, as natural TCRs should not differ this much in terms of their expression levels. It would be helpful to show the FACS plot of the OTR modified T cells.

We agree with reviewer 4, that differences in surface expression between our constructs are significant and likely influence T cell functionality. We extended the results paragraph concerning this topic (lines 250-251, 273-280) and added further information to Figure S4. We further emphasized in figure legend S4B and C, that two different batches of transduced cells were analyzed and therefore data originate from two different experiments.

While for the other neoTCRs we adjusted the rate of TCRmu⁺ cells by adding non-transduced T cells to reach equal total numbers of T cells and TCRmu⁺ cells, the very low transduction rates of KIF-P1 had already caused us to exclude this TCR from most analysis in Figures 3-5. Despite very low frequencies of TCR-tg T cells, this TCR, however, demonstrated unexpected functional potential as discussed in our manuscript

(Figure S15) and differed from the other neoTCRs in k_{off} rate/structural avidity. In the past, comparison of non-optimized and codon-optimized sequences had already shown improvement of surface expression upon codon-optimization ruling out further potential for sequence optimization ¹¹.

To further investigate the differences in expression between OTR and RV system and to understand the promotor's influence on TCR surface expression, we repeated the previous experiment and now include data of three biological replicates of OTR versus RV engineered T cells. As requested by the reviewer we added representative FACS plots of one representative healthy donor to the rebuttal letter (Figure R1) and depicted frequency of CD8⁺ T cells as well as gMFI of TCRmu⁺ T cells in our manuscript (Figure S4H-I). Since overall knock-in frequency upon CRISPR/Cas9-based OTR was significantly lower than in our RV system, FACS-based TCRmu⁺ enrichment and aggressive expansion with PHA and IL-2 was necessary for these experiments. After sort and initial expansion, the density of transgenic TCRs on the surface per cell after retroviral transduction remained comparable to our prior protocol (Figure S4J, K). For the OTR-engineered T cells, on the other hand, KIF-P1 and KIF-sc2 profited from orthotopic expression under the TRAC promotor (Figure S4J, K) as seen during our last round of revisions already. However, as we explained in response to reviewer 1 above, this higher surface expression did not translate into functional advantages. The other two TCRs KIF-P2 and KIF-sc1 remained on the same level of TCR surface expression (OTR versus RV). Thus, at least for the latter two neoTCRs, differences appear to be construct-inherent.

We agree that the conclusions we make for TCR-engineered T cells cannot be drawn for endogenous TCR clonotypes. Therefore, we adapted the manuscript accordingly and clarified that our conclusions concern engineered cells. We also agree that the reduced surface expression of KIF-P1 and -P2 is not yet fully understood. Nevertheless, we want to point out that KIF-P1 and KIF-P2 had extraordinarily high frequencies in patient Mel15 and that increased resilience detected in our experiments corresponds to this pattern in the patient.

4. The fact that T cells were analyzed after a 24-hour co-culture with peptide pulsed targets in many of the assays can potentially lead to artifacts due to the AICD induced by the stimulus. Measurement of apoptotic cells using Annexin V and PI does not necessarily capture this phenomenon as many T cells can die and will not even be seen in these cultures as they may appear in population of small cells that are normally gated out for this analysis. One way of determining the potential effects of cell death would be to evaluate total T cell numbers after the co-culture, which should be reported for these experiments.

As suggested by the reviewer we investigated the absolute count of CD8⁺CD3⁺ and TCRmu⁺ cells in a co culture setting similar to Figures S5-8 after 20h of co culture (Figure R2). When compared to the unstimulated control TCR (2.5D6), loss of CD8⁺ as well as TCRmu⁺ T cells was detectable in all specifically stimulated conditions (KIF-P2, -sc1, -sc2 and SYT-T1). This happened to a similar extent in all conditions without major differences within these first 20h of co culture (Figure R2). Thus, these experiments do not alter the message of our data in Figure 4 with slight trends of more apoptotic cells after 24h for more strongly stimulated TCR-tg T cells, yet no major differences between neoTCRs upon initial tumor encounter.

We also included absolute quantification of remaining TCRmu⁺ T cells upon co culture with U698M in our novel experiments comparing different engineering systems (Figure S16G after 24h, H after 72h). Focusing on initial tumor encounter, there was a slight increase in residual OTR-KIF-P2 TCRmu⁺ T cells and a more pronounced increase of

residual RV-KIF-P1 TCRmu⁺ T cells after initial encounter of 24h (Figure S16G). However, these differences were not significant.

5. The differences between the anti-tumor effects of TIL-P KIF-P2 and TIL-P KIF-sc1 T cells in the tumor re-challenge model was clear in mice receiving the U698M lymphoma shown in Fig. 5B and C; however, the differences seen in mice receiving the A2058 solid tumor shown in Fig. 5G and H and in the repeat experiment with the U698 tumor shown in Fig. S12 were relatively modest, with both populations of TCR TD cells only slowing tumor cell growth but not leading to tumor rejection. In addition, modest differences between the abilities of TIL-P KIF-P1,P2 sc1 and sc2 to slow tumor growth in the experiment shown in Fig. S16. Given the fact that dramatic differences between the 2 TCRs was only seen in 1 of 4 experiments, the conclusion that the ability of T cells transduced with these TCRs to mediate tumor rejection is different does not appear to be strongly supported by the data.

It is true, that a the most striking difference between TIL-P-KIF-P2 and KIF-sc1 with high impact on the overall long-term survival was detected in donor A. Nevertheless, we detected the same significant advantage of KIF-P2 in all other experiments as well as in the newly added in vivo rechallenge comparison of OTR-engineered KIF-P2- versus KIF-sc1-T cells (new Figure S16V, W). We want to stress that we are not only working on primary human cells instead of a genetic, inbred mouse model, but also that these cells are extracted from tumors of variable size and are expanded in presence of variable remaining tumor cells during TIL-P generation. Moreover, considering that all TCRs showed equal killing capacity upon initial tumor encounter with only minor differences in functional avidity and activation strength, we are in fact convinced by the strength and consistency of the data after repeated in vivo challenge, even across different engineering modes.

6. All human T cells express CD45 and so it is not clear why substantial numbers of T cells in Fig. S11 were noted as RA-RO- as they should express 1 or both products.

We agree that we would expect the expression of either one of the two markers on the T cells. We provide control FACS plots from the U698M experiment in Figure R3 and can exclude a major staining errors since we stained the spleen-derived cells simultaneously with overall much lower frequencies of double-negative cells. As demonstrated in Figure R3 of our previous rebuttal letter (November 2023), the frequency of double-negative cells decreased over time of culture after extraction upon sacrifice on day 5, thus the observed CD45RO⁻CD45RA⁻ populations may be isolation and treatment related. Currently, we cannot provide a definitive explanation for these double negative cells on the day of tumor explant, but since all tumor samples were treated equally, any technical effect (e.g. during tumor explanation, digestion or meshing) should concern both TCRs equally.

7. On page 20, you stated that 'Because of the inferior surface expression capacity of KIF-P1 in the RV system (3% compared to 25% upon initial injection into the mice sacrificed for TIL-P), the potent in vivo capacity of TIL-P KIF-P1 upon rechallenge (1% versus 2-8% TCRmu⁺; 5x10⁵ TCRmu⁺ per mouse) was particularly surprising.' It is not clear what this refers to and was difficult to determine where this data was presented in the manuscript.

The reviewer is right that we do not show these data in the manuscript despite stating the frequencies. We simplified the statement in the text (lines 421-425), extended this information in the supplementary figure legend of S15. All neoTCRs apart from KIF-P1 started with the same frequency of TCRmu⁺ cells into the in vivo experiment. Due to this low TCRmu⁺ frequency we had to inject overall much lower numbers of KIF-P1 TCR-tg

T cells and, therefore, were particularly surprised by the potency of the TIL-P generated in the rechallenge round (Figure S15A, B).

8. These are not complete sentences and should be combined in some way to conform to English usage. 'Based on inherent immunogenicity of malignant cells, these therapies make use of the immune system's ability to recognize and eradicate tumor cells. Though the exact interplay between immune recognition and tumor eradication versus escape during immunotherapy remains ill-defined to date. However, its understanding is key to develop full immunotherapeutic potential 2.'

We thank the reviewer for highlighting phrases that need language improvement. We changed the phrases to: "ICI treatment especially bases on unleashing T cells specifically recognizing tumor cells. However, the exact cellular interplay is often multi-faceted and requires deeper understanding to improve therapeutic response." (lines 58-64).

9. On page 23, you stated that 'These findings strengthen TCR-inherence and transferability of the described heterogenous patterns from the patient and indicate a dominant, structural or mechanic component of TCR-peptide-MHC complex assembly rather than patient-imprinted differentiation.'

We understand this criticism and simplified the statement accordingly: "These in vitro findings strengthen TCR-inherence of the heterogenous activation patterns of the patient-derived neoTCRs rather than patient-imprinted differentiation." (lines 529-532)

10. The statement on page 4 'Since one major challenge lies in attacking mutant cells with as little off-target toxicity as possible 5,6, ...' is not correct since the toxicities were due to reactivity with non-mutant targets, and there is no evidence that targeting neoepitopes leads to off-target toxicity.

We do not entirely understand the criticism since our initial phrase was not intended to refer to off-target toxicity due to neoTCRs, but that their employment is a promising possibility to circumvent targeting healthy tissues. We hope that changed wording helps to improve understanding of our intended meaning: "Since a major challenge of adoptive cellular transfer lies in attacking mutant cells without targeting healthy tissues, neoantigens arising from somatic, tumor-restricted mutations promise a safe, precise and highly personalized target structure." (lines 66-70)

11. The statement on page 17 'In parallel, we compared (with what?) with a new transduction with the same two TCRs on CD8⁺ T cells from the same donor (NEW) as control groups (Figure 5A D).' is unclear as noted in parentheses.

We edited the sentence as follows: "In parallel, we compared the performance of these TIL-P with a new transduction of the same two TCRs on CD8⁺ T cells from the same donor (NEW) as control groups (Figure 5A)." (line 378f)

References

- 1 Sim, M. J. W. *et al.* High-affinity oligoclonal TCRs define effective adoptive T cell therapy targeting mutant KRAS-G12D. *Proc Natl Acad Sci U S A* **117**, 12826-12835, doi:10.1073/pnas.1921964117 (2020).
- 2 Poole, A. *et al.* Therapeutic high affinity T cell receptor targeting a KRAS(G12D) cancer neoantigen. *Nat Commun* **13**, 5333, doi:10.1038/s41467-022-32811-1 (2022).
- 3 Klebanoff, C. A., Chandran, S. S., Baker, B. M., Quezada, S. A. & Ribas, A. T cell receptor therapeutics: immunological targeting of the intracellular cancer proteome. *Nat Rev Drug Discov* **22**, 996-1017, doi:10.1038/s41573-023-00809-z (2023).
- 4 Bräunlein, E. *et al.* Functional analysis of peripheral and intratumoral neoantigen-specific TCRs identified in a patient with melanoma. *Journal for ImmunoTherapy of Cancer* **9**, e002754, doi:10.1136/jitc-2021-002754 (2021).
- 5 Tretter, C. *et al.* Proteogenomic analysis reveals RNA as a source for tumor-agnostic neoantigen identification. *Nature Communications* **14**, 4632, doi:10.1038/s41467-023-39570-7 (2023).
- 6 Chatani, P. D. *et al.* Cell surface marker-based capture of neoantigen-reactive CD8(+) T-cell receptors from metastatic tumor digests. *J Immunother Cancer* **11**, doi:10.1136/jitc-2022-006264 (2023).
- 7 Schober, K. *et al.* Orthotopic replacement of T-cell receptor α - and β -chains with preservation of near-physiological T-cell function. *Nat Biomed Eng* **3**, 974-984, doi:10.1038/s41551-019-0409-0 (2019).
- 8 Müller, T. R. *et al.* Targeted T cell receptor gene editing provides predictable T cell product function for immunotherapy. *Cell Rep Med* **2**, 100374, doi:10.1016/j.xcrm.2021.100374 (2021).
- 9 Bassani-Sternberg, M. *et al.* Direct identification of clinically relevant neoepitopes presented on native human melanoma tissue by mass spectrometry. *Nat Commun* **7**, 13404, doi:10.1038/ncomms13404 (2016).
- 10 Legut, M., Dolton, G., Mian, A. A., Ottmann, O. G. & Sewell, A. K. CRISPR-mediated TCR replacement generates superior anticancer transgenic T cells. *Blood* **131**, 311-322, doi:10.1182/blood-2017-05-787598 (2018).
- 11 Lupoli, G. *In vitro and in vivo characterization of T-cell receptors specifically recognizing human melanoma neoantigens identified by immunopeptidomics and in-silico predictions*, Technical University Munich, (2022).

Rebuttal Figure 1. Representative FACS dot plots of all four KIF2C^{P13L}-specific TCRs one day before and five days after FACS sort enrichment of CD8⁺/TCRmu⁺ T cells. Fraction of TCRmu⁺ cells of alive cells/CD8⁺ T cells is depicted for one representative donor for the retroviral (RV) as well as the orthotopic TCR replacement (OTR) setting.

A

B

C

Rebuttal Figure 2. Absolute count of CD3⁺CD8⁺ (A), CD3⁺CD8⁺/TCRmu⁺ (B) as well as the frequency of TCRmu⁺ cells of all CD3⁺CD8⁺ T cells are shown for a co culture setup with U698M (mut mg versus wt mg) after 20h (E:T = 1:2). neoTCRs are compared to unspecific TCR 2.5D6. Mean and SD of three biological replicates are depicted.

tumor

spleen

CD45RA (BV510)

CD3+CD8+/TCRmu+

CD3+CD8+

KIF-P2

KIF-P2

KIF-P2

KIF-P2

KIF-sc1

KIF-sc1

KIF-sc1

KIF-sc1

KIF-P2

KIF-P2

KIF-P2

KIF-P2

KIF-sc1

KIF-sc1

KIF-sc1

KIF-sc1

FMO CD45RO (APC)

FMO CD45RA (BV510)

CD45RO (APC)

Rebuttal Figure 3. Representative FACS pseudocolour plots of CD45RO- and CD45RA-staining upon sacrifice of tumor-bearing mice on day 5 after i.v. T cell injection. Two representative mice for either the tumor- or spleen-derived T cells of KIF-P2 or KIF-sc1 are depicted. Gating on CD3+CD8⁺/TCRmu⁺ (left) is compared to CD3⁺CD8⁺ (right). The FMO controls are shown for CD45RA as well as CD45RO.

REVIEWER COMMENTS

Reviewer #1 (Remarks to the Author):

I thank the authors again for making substantial effort to address my concerns. The additional data have strengthened the manuscript and their rewording has appropriately tempered their conclusions which now more accurately reflects their data. There are only minor points that require clarification:

1. In their new immunopeptidomics data (Fig. S9B), it's unclear what the units are for the y-axis "abundance". Related to this, while U698M expresses relatively less neopeptide/MHC than A2058, it's not clear if these model systems are considered high/low pMHC densities (e.g., how many neopeptide/MHC molecules are predicted to be on each cell for each tumor model)?

2. Fig 3A-F does not have labels to identify each TCR (which colors/symbols refer to which TCR?). Also for Fig.3, is the data from TCR enriched cells or bulk populations? This would have implications for interpretation of the cytokine secretion assays.

3. Lines 344 to 347 in the marked up version, "Subsequently, to investigate differences within our observed spectrum of activation, we focused on two TCRs with shared neoantigen-specificity, HLA-restriction as well as similar surface expression under different promoters, yet different activation patterns: moderate (KIF-P2) versus strong (KIF-sc1)". According to their data (e.g., Fig. S4) and previous statements in the manuscript (e.g., line 275) KIF-P2 does not have a similar transgenic TCR surface expression as KIF-sc1. This probably is not what the authors meant (but it is how I interpreted it) so this sentence would benefit from rewording.

Reviewer #4 (Remarks to the Author):

While the authors addressed some of the concerns raised by the reviewers there were some points that were not adequately addressed in the revised manuscript. The major points are listed below.

1. The reactivity patterns of KIF-P1 and P2 are indicative of highly cross-reactive TCRs, which is apparent in their recognition motifs and the number of human proteins containing matching recognition motifs. This indicates that the reason why these clonotypes are highly expanded may be that they are cross-reactive with many antigens and relatively non-promiscuous. These clones may also be less reactive with the KIF neoepitope since they are not a particularly good fit to this epitope. They may represent clonotypes that also recognize viral epitopes, for example, and may not be expanded due to their tumor reactivity as bystander cells are known to be present in tumor samples. Given this cross-reactivity it would be helpful to know if these clonotype do recognize these candidate self-antigens, which would be easy to test.

2. It is difficult to conclude much about the relative activities of KIF-sc1 and KIR-sc2, as the data in Fig. 3 indicate that for most of the assays there were minor differences that do not appear to be significant. Even the claim that SYT-T1 shows stronger activation than the KIF TCRs is only manifest at a few concentrations or time points and does not appear to be significantly different from the KIF TCRs.

3. It appears that the overall %s of KIF-P2 and KIF-sc1+ T cells in the pooled populations transferred to secondary recipients were similar and thus the overall number of neoepitope-reactive T cells administered to the secondary recipients were similar but was the overall growth rate of these populations similar? The data for the overall expansion of KIF-P2 and KIF-sc1+ T cells as shown in Fig. S12 D-I but this was a separate experiment according to the legend and does not represent the expansion of T cells in used in the secondary transfer experiment.

There were also some minor points that should be addressed in the revised manuscript.

1. There were some statements that were not supported by the data. On page 10 the text reads:

‘Regarding unbiased analysis of unstimulated neoTCRs, again only KIF-P1 and -P2 transcriptomes could be analyzed comprising sufficient cell counts. Comparing both TCR clonotypes with all other unstimulated T cell clones, cytotoxic markers including FGFBP2,

GZMB, GZMH, GNLY and NKG7 were predominantly upregulated (Figure S3D).’ It appears from the data that these genes were down-regulated in the KIF-P1 and P2 clonotypes. If this is correct, then it difficult to understand why effector genes appear to be up-regulated relative to the SYT TCRs. It is also counterintuitive that the KIF-P1 and P2 clonotypes that represented the most highly expanded populations in the unstimulated population would represent the population expressed lower levels of inhibitory markers that generally are associated with terminal differentiation than the additional clonotypes that are less expanded in the unstimulated population of cells.

2. It is not clear what is being referred to in the statement on page 15 ‘In fact, the level of T cell activation after co culture with mg-expressing or peptide-pulsed targets correlated with the level of antigen, however, was also dependent on other determinants of the tumor entity (Figure S9C-E).’ – what is ‘mg-expressing’

?

3. In this statement ‘During in vitro expansion of TIL-P of the individual mice we again detected differences in TCRmu+-frequencies (Figure 132A-C)’ the figure should be referred to as S12A-3.

Point-to-point reply (NCOMMS-23-01734B)

Reviewer #1 (Remarks to the Author):

I thank the authors again for making substantial effort to address my concerns. The additional data have strengthened the manuscript and their rewording has appropriately tempered their conclusions which now more accurately reflects their data.

We thank reviewer 1 for the critical review of our manuscript. We agree that the additional data included, and textual changes made, improved the manuscript and the precision of our message. We are happy to address all remaining concerns to further improve readability.

There are only minor points that require clarification:

1. In their new immunopeptidomics data (Fig. S9B), it's unclear what the units are for the y-axis "abundance". Related to this, while U698M expresses relatively less neopeptide/MHC than A2058, it's not clear if these model systems are considered high/low pMHC densities (e.g., how many neopeptide/MHC molecules are predicted to be on each cell for each tumor model)?

The abundance measured by MS is the peak intensity or area under the curve (from the LC-MS trace) of the detected peptide and is depicted in arbitrary units. We clarified this in the legend of Figure S9 as "abundance (in arbitrary units) showing the peak intensity or area under the curve (from the LC-MS trace) of the detected peptide KIF2C^{P13L}".

Furthermore, the data presented for the cell lines aimed at a relative comparison between the different cell lines regarding the expressed minigenes but also to peptide-pulsed cells. These quantification methods for either HLA density (FACS) or peptide expression (MS) only work relatively in themselves. We want to avoid a too general statement about overall classification of these cell lines as high or low pMHC densities outside the methods we can comment on and treat them as model systems which are per se different to highly heterogenous patient material as discussed in our previous rebuttal.

2. Fig 3A-F does not have labels to identify each TCR (which colors/symbols refer to which TCR?). Also for Fig.3, is the data from TCR enriched cells or bulk populations? This would have implications for interpretation of the cytokine secretion assays.

We thank the reviewer for the attentive review of our figures. The labels of Figure 3A-F follow the color code as indicated next to 3H. To clarify, we added this information to the beginning of the legend of Figure 3.

3. Lines 344 to 347 in the marked up version, "Subsequently, to investigate differences within our observed spectrum of activation, we focused on two TCRs with shared neoantigen-specificity, HLA-restriction as well as similar surface expression under different promoters, yet different activation patterns: moderate (KIF-P2) versus strong (KIF-sc1)". According to their data (e.g., Fig. S4) and previous statements in the manuscript (e.g., line 275) KIF-P2 does not have a similar transgenic TCR surface expression as KIF-sc1. This probably is not what the authors meant (but it is how I interpreted it) so this sentence would benefit from rewording.

We agree with the reviewer that this phrasing might cause misunderstanding. Our intended meaning was concerning their similarity of surface expression upon either RV or OTR engineering. However, we adapted the phrase accordingly and rephrased parts of the sentence: "Subsequently, to investigate differences within our observed spectrum of activation, we focused on two TCRs with shared neoantigen-specificity, HLA-restriction as well as similar behavior of surface expression in different engineering systems, yet different activation patterns: moderate (KIF-P2) versus strong (KIF-sc1)".

Reviewer #4 (Remarks to the Author):

While the authors addressed some of the concerns raised by the reviewers there were some points that were not adequately addressed in the revised manuscript. The major points are listed below.

We thank the reviewer for his/her/their feedback on our manuscript. We hope to convince the reviewer that some of the information required is either already included into the manuscript or that additional questions can be answered in the scope of this rebuttal letter.

1. The reactivity patterns of KIF-P1 and P2 are indicative of highly cross-reactive TCRs, which is apparent in their recognition motifs and the number of human proteins containing matching recognition motifs. This indicates that the reason why these clonotypes are highly expanded may be that they are cross-reactive with many antigens and relatively non-promiscuous. These clones may also be less reactive with the KIF neoepitope since they are not a particularly good fit to this epitope. They may represent clonotypes that also recognize viral epitopes, for example, and may not be expanded due to their tumor reactivity as bystander cells are known to be present in tumor samples. Given this cross-reactivity it would be helpful to know if these clonotype do recognize these candidate self-antigens, which would be easy to test.

We demonstrate a high specificity of both neoTCRs in the in vitro and in vivo system chosen for our experiments outside Mel15 given the absent reactivity against the wt-peptide or -minigene control shown in Figures 1C, 3A-D and S5B-D, S6C, F, I, S7C, F, I, L, O, S8B, D, F as for KIF-P1 S15C. Challenging those TCRs repeatedly in vivo using our xenogeneic mouse model, we could confirm that KIF-P1 and -P2 show an impressively strong and specific reactivity against KIF2C^{P13L}-bearing tumors. We do not expect that any relevant cross reactivity does play a role in our mouse model with exclusive HLA-A3 expression on the tumor cells.

Nonetheless, as published before, KIF-P1 and KIF-P2 bear limited potential cross-reactivity in the human repertoire and a selection of these potentially cross-reactive peptides identified by ScanProsite was tested previously. Of note, many of these peptides were derived from spliced protein variants inheriting the same sequence and being therefore redundant. Relevant peptides were selected based on their potential HLA binding capacity, but did not elicit any reactivity by either TCR KIF-P1 or KIF-P2 (1, Figure S7 in this previous publication). Of course, we cannot definitively exclude potential recognition of any other not predicted, structurally similar epitopes, which might indeed influence clonotype abundance in the repertoire of patient Mel15 1. However, we consider this not as a focus of our current manuscript.

2. It is difficult to conclude much about the relative activities of KIF-sc1 and KIR-sc2, as the data in Fig. 3 indicate that for most of the assays there were minor differences that do not appear to be significant. Even the claim that SYT-T1 shows stronger activation than the KIF TCRs is only manifest at a few concentrations or time points and does not appear to be significantly different from the KIF TCRs.

We agree with the reviewer's notion that the differences between our neoTCRs assessed in this specific setting are minor and mostly not significant. However, we don't suggest significant differences for the experiments made in Figure 3 or its corresponding supplementary figures, but rather confirm distinct patterns which we repeatedly observed upon specific T cell activation (e.g. summarized in lines 301-309). Notably, these differences in moderate versus stronger activation, mirrored by surface activation markers, cytokine secretion and apoptotic cells, were observed throughout all our in vitro experiments. Therefore, it was even more striking that these rather minor differences were opposed by significant in vivo differences upon rechallenge representing continuous T cell activation which is difficult to mimic in in vitro studies. We also state this in our manuscript lines 575-579. Particularly because of this we are convinced that our data and proposed model are highly relevant for the field.

3. It appears that the overall %s of KIF-P2 and KIF-sc1+ T cells in the pooled populations transferred to secondary recipients were similar and thus the overall number of neoepitope-reactive T cells administered to the secondary recipients were similar but was the overall growth rate of these populations similar? The data for the overall expansion of KIF-P2 and KIF-sc1+ T cells as shown in

Fig. S12 D-I but this was a separate experiment according to the legend and does not represent the expansion of T cells in used in the secondary transfer experiment.

The data presented in Figure S12D-I were acquired during the initial round of revisions of this manuscript in response to reviewer #3 and represent relative quantification for the initial rechallenge experiment in the form of frequency of TCRmu⁺ of CD8⁺ T cells as presented in Figure S12A-C. As stated by the reviewer, we tried to keep the reinjected TIL-P as consistent as possible. Therefore, the amount of TCRmu⁺ T cells was comparable for both neoTCRs, concerning absolute counts of TCRmu⁺ as well as frequency of TCRmu⁺ within all CD8⁺ T cells for all three donors A, B and C. The data in Figure S12D-I was acquired at a later time point, in the same experimental setup comprising all KIF-neoTCRs resulting in the rechallenge data presented in Figure S15A-B. Therefore, this setting relied on the exact same tumor cell line (U698M-mut mg), under the same conditions as the previous experiments with donors A, B and C. Even donor A was recruited again for the experiment in S12D-I to keep these experiments as consistent as possible. Therefore, the data shown in Figure S12D-I are representative data of our system.

While the data in S12 already show the TCR-specific expansion during the 21 days of generating the TIL-P in our system, we can furthermore demonstrate for donor A (Figure 5B,C), that both TCR-tg TIL-P-T cell populations show similar, non-existent background growth in response to the wildtype-minigene (wt mg)-tumor cell line (U698M-wt mg) or on themselves (T cell only) after in vitro restimulation of TIL-P on the day of reinjection into the second round of tumor-bearing hosts. Please find below the CTV-based quantification of proliferation of the pooled TIL-P of KIF-P2 and KIF-sc1 to which we now added a “T cell only” control (Figure R1).

Analysis of proliferation of single cells during the 21-day period of TIL expansion would be indeed an interesting aspect but remains beyond the scope of our manuscript.

We are convinced that with the information in Figure S12D-I and the data added to this rebuttal, the provided data supports our message sufficiently and hope that answers the reviewer's question.

Figure R1. A-C, Pooled TIL-P for either KIF-P2 or KIF-sc1 from the experiment on donor A (Figure 5B, C and S11E, H, K) were either cultured without addition of tumor cells (A) or restimulated with U698M-wt mg (B) or -mut mg (C) as target cells in E:T = 1:1 (50,000 tg T cells:50,000 tumor cells) on the day of reinjection into the second round of tumor bearing hosts (day 21 of TIL-P expansion protocol, Figure 5A). T cells were CTV-labelled prior to restimulation co culture setup and triplicates were pooled prior to FACS analysis. On day 4 CTV-labelled T cells (alive/CD3⁺CD8⁺/TCRmu⁺ cells) were analyzed.

There were also some minor points that should be addressed in the revised manuscript.

1. There were some statements that were not supported by the data. On page 10 the text reads: 'Regarding unbiased analysis of unstimulated neoTCRs, again only KIF-P1 and -P2 transcriptomes

could be analyzed comprising sufficient cell counts. Comparing both TCR clonotypes with all other unstimulated T cell clones, cytotoxic markers including FGF2, GZMB, GZMH, GNLY and NKG7 were predominantly upregulated (Figure S3D).¹ It appears from the data that these genes were down-regulated in the KIF-P1 and P2 clonotypes. If this is correct, then it difficult to understand why effector genes appear to be up-regulated relative to the SYT TCRs. It is also counterintuitive that the KIF-P1 and P2 clonotypes that represented the most highly expanded populations in the unstimulated population would represent the population expressed lower levels of inhibitory markers that generally are associated with terminal differentiation than the additional clonotypes that are less expanded in the unstimulated population of cells.

The reviewer is referring to two different comparisons in this newly raised concern that has not undergone change in the last round of revisions. On the one hand, we focused on the specifically restimulated populations of neoTCRs (Figure 2E-G) by comparing KIF- towards SYT-TCR-clonotypes based on sufficient cell counts. This comparison showed stronger activation on SYT-specific TCRs compared to KIF-TCRs. On the other hand, we analyzed the unstimulated fraction referred to in the phrase cited above by the reviewer (only unstimulated, blue cells of Figure 2A). Here we could only compare KIF-TCR clonotypes KIF-P1 and KIF-P2 to all other unassigned TCR-clonotypes since we did not detect enough SYT-specific TCRs in the unstimulated fraction of MEL15's PBMCs. Thus, the phrase cited by the reviewer is not a contradiction, but a mere additional information on top of the central comparison made in Figure 2.

Referring to Figure S3D, the non-stimulated KIF-P1 and -P2 clonotypes indeed show higher expression of the described cytotoxic markers than all non-assigned unstimulated cells and are therefore plotted on the very left part of the volcano plot. In general, all genes in the left part are higher expressed in KIF-P1/-P2 and all genes on the right side are higher expressed in non-assigned unstimulated cells. Regarding the observation of low inhibitory gene expression in KIF-P1/-P2 cells, this is actually a very interesting observation and might rely on the fact that the patient was already disease-free for a couple of months at the time point of this analysis.

2. It is not clear what is being referred to in the statement on page 15 'In fact, the level of T cell activation after co culture with mg-expressing or peptide-pulsed targets correlated with the level of antigen, however, was also dependent on other determinants of the tumor entity (Figure S9C-E).'¹ – what is 'mg-expressing'?

We clarified this phrase in the text as follows: "In fact, the level of T cell activation after co culture with minigene-expressing or peptide-pulsed targets correlated with the level of antigen since much higher concentrations of peptide were needed to achieve comparable activation between the mut mg and pulsed conditions for U698M compared to A2058. However, regarding Mel15 LCL in comparison to the other cell lines, it becomes evident, that this response seemed also dependent on other determinants of the tumor entity (Figure S9C-E)."

3. In this statement 'During in vitro expansion of TIL-P of the individual mice we again detected differences in TCRmu⁺-frequencies (Figure 132A-C)' the figure should be referred to as S12A-3.

We thank the reviewer for this attentive review of our manuscript. The crossing out of 3 in the marked version of our manuscript was barely visible and the wrong number is now deleted together with all other deleted passages of the old manuscript version.

References

- 1 Bräunlein, E. *et al.* Functional analysis of peripheral and intratumoral neoantigen-specific TCRs identified in a patient with melanoma. *Journal for ImmunoTherapy of Cancer* **9**, e002754 (2021). <https://doi.org/10.1136/jitc-2021-002754>

REVIEWERS' COMMENTS

Reviewer #1 (Remarks to the Author):

The authors missed addressing one of my comments (the second part of comment 2),
"...Also for Fig.3, is the data from TCR enriched cells or bulk populations? This would have implications for interpretation of the cytokine secretion assays."

All of my other comments/concerns have been sufficiently addressed.

Reviewer #4 (Remarks to the Author):

I believe that all of the the issues raised by the reviewers have been adequately addressed in the revised manuscript.

Point-to-point reply for final revisions of NCOMMS-23-01734C

We want to thank all reviewers for their thorough feedback on our manuscript which has substantially improved and strengthened the message of our work.

Reviewer #1 (Remarks to the Author):

The authors missed addressing one of my comments (the second part of comment 2), "...Also for Fig.3, is the data from TCR enriched cells or bulk populations? This would have implications for interpretation of the cytokine secretion assays."

All of my other comments/concerns have been sufficiently addressed.

We apologize for this oversight on our side. All TCR-populations were bulk populations, whose TCRmu⁺ TCR-tg frequencies we adjusted by addition of non-transduced T cells to generate equal transduction rates for all conditions in each co culture experiment. We chose the condition with lowest transduction rate of each transduction and adjusted all higher ones to it. We had already stated this in our method section, however, to clarify for our readers we added a statement to Figure legend 3 accordingly: "For all co cultures, TCRmu⁺ rates were adjusted by addition of non-transduced T cells to equalize TCRmu⁺ cell frequencies for all neoTCRs."

We thank reviewer #1 for his/her/their comments to improve our manuscript.

Reviewer #4 (Remarks to the Author):

I believe that all of the the issues raised by the reviewers have been adequately addressed in the revised manuscript.

We thank reviewer #4 for his/her/their feedback on our work.